# Erythropoietin regulates energy metabolism through EPO-EpoR-RUNX1 axis

Weiqin Yin [1], Praveen Kumar Rajvanshi[1], Heather M. Rogers[1],
Teruhiko Yoshida [2], Jeffrey B. Kopp [2], Xiuli An[3], Max Gassmann [4] &
Constance T. Noguchi [1] ✉

Erythropoietin (EPO) plays a key role in energy metabolism, with EPO receptor (EpoR) expression in white adipose tissue (WAT) mediating its metabolic activity. Here, we show that male mice lacking EpoR in adipose tissue exhibit increased fat mass and susceptibility to diet-induced obesity. Our findings indicate that EpoR is present in WAT, brown adipose tissue, and skeletal muscle. Elevated EPO in male mice improves glucose tolerance and insulin sensitivity while reducing expression of lipogenic-associated genes in WAT, which is linked to an increase in transcription factor RUNX1 that directly inhibits lipogenic genes expression. EPO treatment in wild-type male mice decreases fat mass and lipogenic gene expression and increase in RUNX1 protein in adipose tissue which is not observed in adipose tissue EpoR ablation mice. EPO treatment decreases WAT ubiquitin ligase FBXW7 expression and increases RUNX1 stability, providing evidence that EPO regulates energy metabolism in male mice through the EPO-EpoR-RUNX1 axis.

Erythropoietin (EPO) regulates red blood cell production by binding to EPO receptor (EpoR) on erythroid progenitor cells in the bone marrow to promote survival, proliferation, and differentiation[1]. EpoR expression extends beyond erythroid tissue and provides for potential EPO activity in non-erythroid tissues including EPO-protective response to injury or ischemia in brain, the cardiovascular system or skeletal muscle[2–4]. In addition to increased hematocrit, EPO treatment in mice protects against diet-induced obesity[5,6]. Mice with EpoR restricted to erythroid tissue (ΔEpoR$_E$-mice) become obese, glucose intolerant, and insulin resistant[7]. EPO treatment in mice during high-fat diet (HFD) feeding reduces fat mass, increases metabolic activity, increases white adipose tissue (WAT) cellular respiration, mitochondrial biogenesis, and shifts WAT gene expression toward a brown adipose tissue (BAT) program increasing uncoupling protein UCP1, and increases BAT expression of transcription coregulator PRDM16, and UCP1, and STAT3 activation[8,9]. EPO metabolic regulation in human is suggested by the negative association between endogenous EPO level and percent weight change per year in male Pima Indians from the Gila River Indian Community with high prevalence of obesity and type 2 diabetes[10,11]. Furthermore, EPO induction at high altitude may account for the inverse association between altitude and obesity observed among US military recruits and among Peruvian adults[12,13]. For over 3 decades, recombinant human EPO has been used to treat anemia in chronic kidney disease resulting in decreased transfusion requirement[14]. EPO treatment has been associated with increased insulin sensitivity in hemodialysis patients independent of EPO stimulated increase in hematocrit[15]. Hemodialysis patients with EPO resistance are associated with a higher frequency of metabolic syndrome[16]. Healthy young men with acute EPO treatment exhibited increased resting energy expenditure with a tendency to increase fat oxidation[17]. EPO regulation of fat metabolism in animal models may shed light on human metabolic response to acute and chronic EPO treatment.

In mammals, WAT is critical for excess energy storage and energy mobilization as fatty acids into the blood stream[18]. Dysfunction of lipid metabolism causes a variety of metabolic-related diseases such as obesity, type 2 diabetes mellitus, lipodystrophy, and cachexia. Increase

[1]Molecular Medicine Branch, National Institute of Diabetes and Digestive and Kidney Diseases, NIH, Bethesda, MD, USA. [2]Kidney Diseases Branch, National Institute of Diabetes and Digestive and Kidney Diseases, NIH, Bethesda, MD, USA. [3]Laboratory of Membrane Biology, New York Blood Center, New York, NY, USA. [4]Institute of Veterinary Physiology and Zurich Center for Integrative Human Physiology, University of Zurich, Zurich, Switzerland. ✉e-mail: connien@niddk.nih.gov

in fat cell mass leads to hormone imbalance and metabolic effects including impaired glucose tolerance and insulin resistance[19]. Here we used mouse models to assess EPO metabolic regulation of fat mass and lipid metabolism. EpoR-tdTomato-Cre-mice containing insertion of tdTomato-Cre downstream of the EpoR coding region showed EpoR in adult non-erythroid tissue is high in WAT compared with BAT, skeletal muscle, and liver. While EPO treatment in male mice can reduce body fat and weight gain, especially during HFD feeding, this is not observed in female mice that are not ovariectomized due to the protective effect of estrogen to diet-induced obesity[20].

Here, we show that EPO treatment in male mice and transgenic Tg6-mice with constitutive overexpression of human EPO[21] demonstrated metabolic response to increased EPO and that EPO regulation of lipid metabolism in WAT, BAT, skeletal muscle, and liver required EpoR expression in adipose tissue. EPO activity in subcutaneous WAT (scWAT) increased protein stability of transcription factor RUNX1, increased expression of lipolytic genes that contain RUNX1 dependent enhancer motifs, and decreased expression of lipogenic genes that contain RUNX1 dependent silencer motifs. Deletion of EpoR in adipose tissue decreased stability of RUNX1 in scWAT and abrogated EPO stimulated increase of lipolytic genes and decreased lipogenic gene expression in WAT as well as BAT, skeletal muscle, and liver.

## Results

### High transgenic EPO expression decreased fat mass and increased glucose tolerance and insulin sensitivity

Elevated EPO level in mice induces red blood cell production and modulates fat–glucose metabolism. EPO regulation of fat mass is sex dimorphic, apparent in males and less evident in females related to estrogen protective activity against obesity[20,22]. Therefore, we focused on the effect of high transgenic EPO expression in male Tg6-mice as a model system for chronic EPO treatment. Compared with wild-type (WT) littermate controls, Tg6-mice exhibit lower body weight and increased hematocrit beginning 2 weeks after birth and increased adipose tissue mitochondrial oxygen consumption rate (OCR) (Supplementary Fig. 1a–c). Tg6 hematocrit increased after 2 weeks of age to 0.52 (Tg6) vs 0.42 (WT) at postnatal day 21, and to 80% at 35 days (Supplementary Fig. 1b). Tg6-mice at 100 days exhibited no inflammation or liver necrosis (Supplementary Fig. 1d–f), indicating that overexpression of transgenic EPO in young Tg6-mice does not induce tissue necrosis. The increase in hematocrit due to high transgenic EPO expression was concomitant with an increase in spleen size and by 2 months, spleen weights of Tg6-mice were three times that of WT-mice (Supplementary Fig. 2). Five months Tg6-mice show no signs of degeneration in liver, kidney, skeletal muscle and sciatic nerve[23], suggesting that Tg6-mice may be useful to assess response associated with chronic EPO treatment. Tg6-mice show EPO-protective activity in light induced retinal degeneration[24], cardiac-protection in permanent coronary artery ligation[25], skeletal muscle repair[4], and metabolic regulation of blood glucose and body mass[5]. Of note, EPO treatment in sick lean mice can decrease inflammation and disease manifestation, and improve locomotor activity as exemplified in a model for sepsis, without significant change in body weight[26]. However, after 7 months, Tg6-mice exhibit multiple organ degeneration, liver inflammation, and, in about half the mice, hindlimb paralysis and myofiber degeneration in hindlimb skeletal muscle[23]. Tg6-mice mean survival is reduced (7.4 months)[27]. Therefore, young Tg6-mice prior to 4 months were used in the current study.

Tg6-mice from age 3 weeks had lower body weight and reduced fat mass and no difference in lean mass compared with WT littermate controls (Fig. 1a–d), indicating that high EPO affects body weight and fat mass by reducing fat accumulation or inhibiting fat formation. High serum EPO in Tg6-mice also significantly increased blood glucose tolerance and insulin sensitivity (Fig. 1e–h). Adipocyte cell size contributes importantly to cellular function and health[28]. Adipocytes in WAT and BAT from Tg6-mice were smaller compared to WT-mice

(Fig. 1i–k), consistent with lower body weight and fat mass of Tg6-mice. Compared with littermate controls, young Tg6-mice exhibited reduced serum levels of leptin, insulin, and glucose, increased levels of adiponectin, and comparable levels of inflammatory cytokines TNF, INF-γ, IL-1α, IL-1β, and IL-6, and AST and ALT as an indication of liver function (Supplementary Fig. 3a–k). Tg6-mice fed HFD for 3 weeks showed lower body weight and reduced fat mass and no significant difference in lean mass compared with WT-mice fed on HFD (Supplementary Fig. 4a–c). Glucose tolerance was also significantly increased in Tg6-mice compared with WT littermate control mice on HFD mice (Supplementary Fig. 4d, e). Three weeks EPO treatment in WT-mice on HFD also showed analogous reduction in body weight and fat mass and no difference in lean mass, and increased glucose tolerance compared with saline treatment (Supplementary Fig. 4f–j). In addition, metabolic measurements by indirect calorimetry did not identify significant differences between WT-mice and Tg6-mice or any difference in lean mass, although Tg6-mice appeared to exhibit reduced activity at night (Supplementary Fig. 5a–j).

### EpoR was mainly expressed in white adipose tissue

EPO stimulates red blood cell production by binding to EpoR on erythroid progenitor cells and early stage erythroblasts that express the highest level of EpoR[21]. Determination of non-erythroid EpoR expression in EpoR-tdTomato-Cre mice showed EpoR at high level in spleen, scWAT and epididymal WAT (eWAT) compared with BAT, cerebral cortex, hypothalamus, skeletal muscle, and liver that overlapped with tdTomato expression (Fig. 2a and Supplementary Fig. 5k–m). The relative level of EpoR expression was reflected in the number of EpoR positive cells (Fig. 2b) and mRNA expression (Fig. 2c) and was validated by western blotting (Fig. 2d, e). The high levels of EpoR expression in WAT suggest that EPO may target adipose tissue mainly via EpoR expression and affect the synthesis and decomposition of fat in adipose tissue. The effect of EPO on other organs may be directly mediated by EpoR expression, although at a markedly reduced level compared with WAT, or may be an indirect consequence of EpoR mediated EPO response of adipose tissue affecting fat storage in other organs. To test this hypothesis, we determined the effect of high EPO on expression of genes associated with fat synthesis and lipid metabolism.

### EPO increased the expression of lipolytic genes and inhibited expression of lipogenic genes in subcutaneous WAT of Tg6-mice

Mitochondria uncoupling protein 1 (UCP1) is associated with non-shivering thermogenesis in BAT. UCP1 was increased in Tg6 scWAT compared with WT-mice (Fig. 3a, b). Tg6 scWAT also showed elevated expression of BAT-associated genes, *Ucp1*, *Pgc1α*, *Cidea*, and *Prdm16* compared with WT scWAT in male mice (Fig. 3c). However, expression of BAT-associated genes was not different from WT in corresponding eWAT (Fig. 3d). These results suggest that the EPO–EpoR axis may affect metabolism of fat in various organs by promoting the browning of scWAT. In scWAT, high EPO significantly increased expression of lipolytic genes, *Pparγ* and *Lpl*, and decreased expression of lipogenic genes, *Acc1/2*, *Fas*, *Lipin1*, *Srebf1*, and *Scd1* in Tg6-mice compared with WT-mice (Fig. 3e). While expression of UCP1 and BAT-associated genes was not elevated in Tg6 eWAT, increases in expression of lipolytic genes and decreases in lipogenic genes were observed in eWAT as well as BAT, skeletal muscle, and liver in the Tg6-mice compared with WT-mice (Fig. 3f–i) albeit at reduced level compared with changes in Tg6 scWAT (Fig. 3e). Browning of white fat in scWAT may be related to decreased fat synthesis[29].

### EPO increased lipolytic gene expression and decreased lipogenic gene expression in WT-mice but had no effect on ΔEpoR$_E$ mice with erythroid restricted EpoR

In Tg6-mice with chronic overexpression of transgenic EPO, the EPO–EpoR axis played an important role in regulating lipolysis and

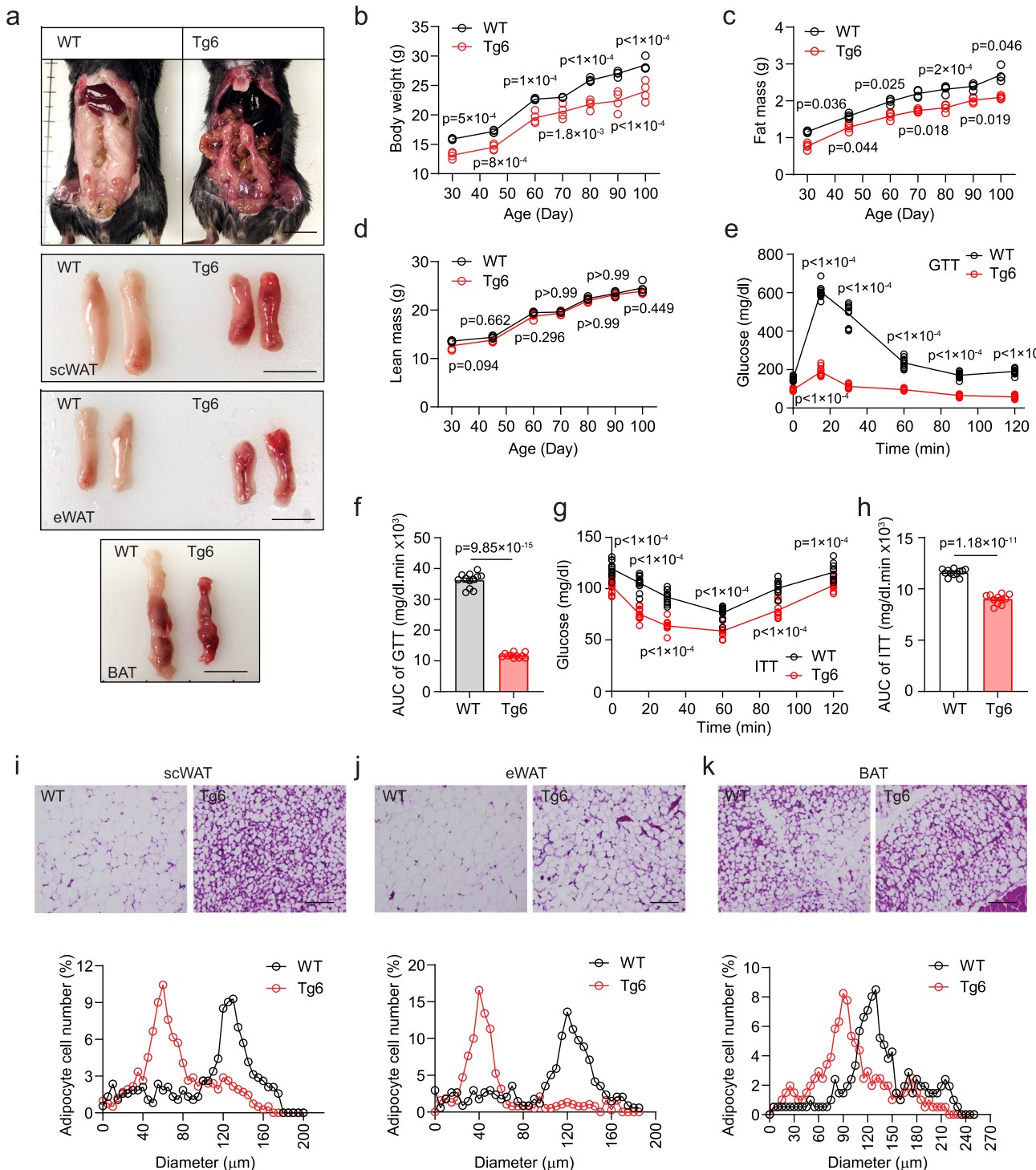

**Fig. 1 | Chronic high EPO transgenic expression in young mice decreased fat mass, increased glucose and insulin tolerance, and reduced adipocyte size in adipose tissue.** Metabolic parameters were determined for young, male Tg6-mice (constitutive transgenic EPO expression) compared with wild-type (WT) mice on normal chow diet from postnatal 21 days to postnatal 100 days. **a** Tg6-mice exhibited reduced body size and smaller fat pads from subcutaneous WAT (scWAT), epididymal WAT (eWAT) and BAT compared with WT-mice. Scale bar: 1.0 cm. **b–h** Body weight (**b** $F_{(6, 18)} = 2.82$), fat mass (**c** $F_{(2.220, 6.660)} = 3.209$), glucose tolerance test (GTT) (**e** $F_{(5, 49)} = 323.1$) and area under the curve (AUC) for GTT (**f** $t = 33.26$, df = 21), and insulin tolerance test (ITT) (**g** $F_{(5, 49)} = 6.470$) and

AUC for ITT (**h** $t = 15.07$, df = 21) were reduced, but lean mass (**d** $F_{(6, 18)} = 0.5754$) was similar in Tg6-mice (red) compared with WT-mice (gray). **i–k** H&E-stained sections of adipose tissue from scWAT (**i**), eWAT (**j**), and BAT (**k**) showed reduced adipocyte size in Tg6-mice (red) compared with WT-mice (gray). $p$ values are indicated. **f, h**: $t$-test; **b–e, g, i–k**: two-way ANOVA with Bonferroni's multiple comparisons test. Scale bar: scWAT: 50 μm; eWAT and BAT: 25 μm. **b–d** WT-mice and Tg6-mice: $n = 4$/group; **e–h** WT-mice: $n = 12$, Tg6-mice: $n = 11$; **i–k** $n = 200$ cells/group. **f, h** Data shown as mean ± SEM. **f, h**: two-sided unpaired $t$-test. $p$ values indicated in figure. Source data are provided as a Source Data file.

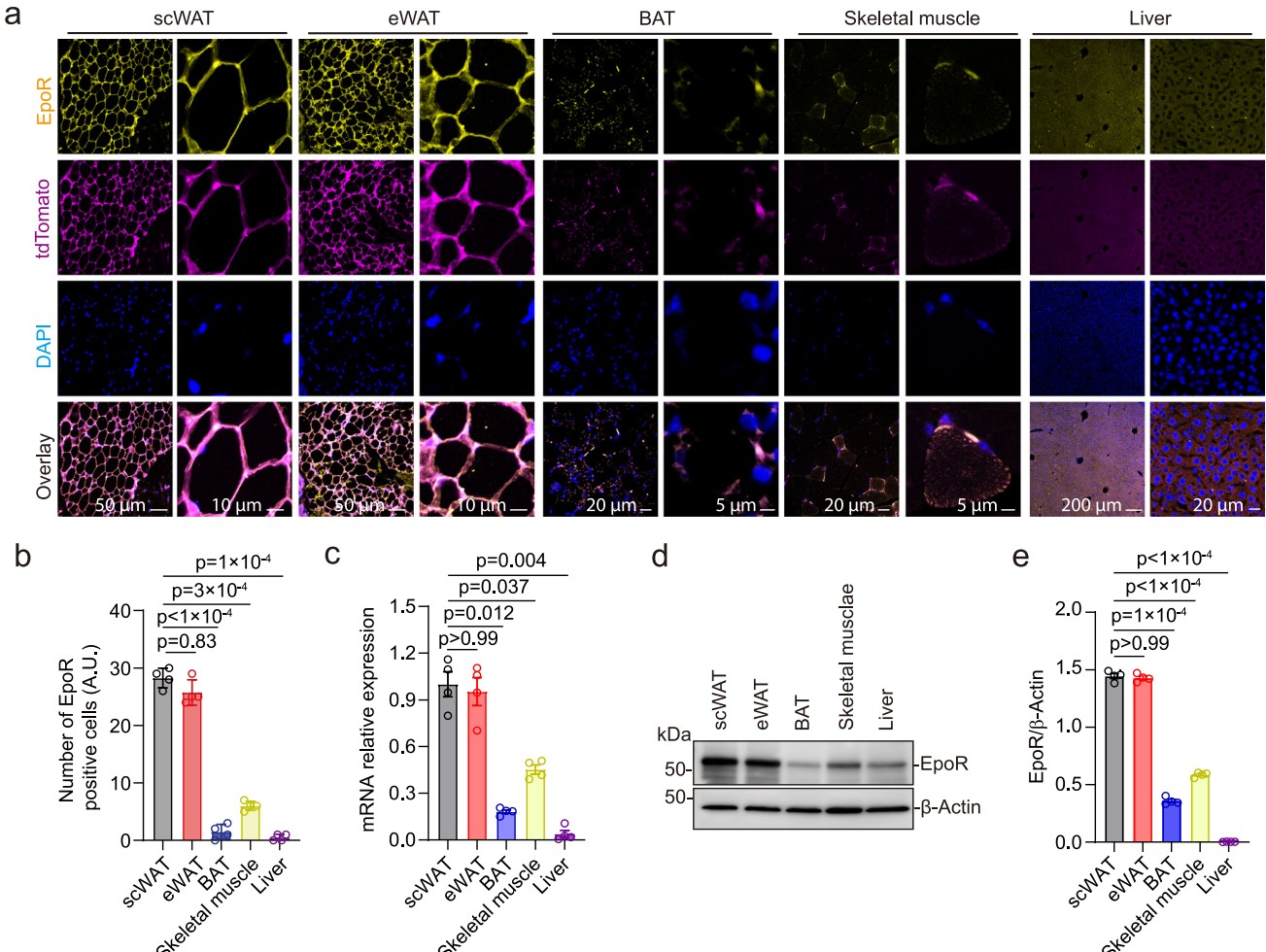

**Fig. 2 | Tissue-specific expression of EpoR was determined in EpoR-tdTomato-Cre mice. a** Images shown are staining of scWAT, eWAT, BAT, skeletal muscle, and liver from male mice (postnatal 100 days) for EpoR (yellow), tdTomato (purple), and DAPI (blue) at low and high (scale bar: as indicated) magnification. Scale bar: scWAT and eWAT, 50 μm (left), 10 μm (right); BAT and Skeletal muscle, 20 μm (left), 5 μm (right); Liver, 200 μm (left), 20 μm (right). **b, c** Number of EpoR positive cells (**b** $F(1.574, 4.721) = 366.9$) and relative mRNA expression determined by qPCR (**c** $F(1.941, 5.824) = 52.06$) were assessed for scWAT (gray), eWAT (red), BAT (blue), skeletal muscle (yellow), and liver (purple). **d, e** EpoR protein expression was determined by western blotting with β-actin as control (**d**) and was quantified by ImageJ (**e** $F(1.540, 4.619) = 2194$) for scWAT (gray), eWAT (red), BAT (blue), skeletal muscle (yellow), and liver (purple). Data are shown as mean ± SEM. **b, c, e**: $n = 4$ mice/group. **b, c**: the result shown in figure represents one of three independent experiments, $p$ values are indicated. Scale bars indicated in the figure. One-way ANOVA with Bonferroni's multiple comparisons test. Source data are provided as a Source Data file.

lipogenesis in WAT, BAT, skeletal muscle, and liver (Fig. 3). Tg6 liver sections showed increase presence of red cells and blood vessels, and no signs of necrosis (Supplementary Fig. 1d–f), or of increased inflammatory cytokines (Supplementary Fig. 3e–i), providing evidence that the observed catabolic lipid changes are not a consequence of whole-body necrosis. To determine if activation of the EPO–EpoR axis in erythroid cells or by transient EPO treatment affects fat metabolism gene expression, EPO was administered to WT-mice and ΔEpoR$_E$-mice at postnatal 21 days, three times a week for 3 weeks. EPO treatment significantly decreased body weight and fat mass in WT-mice (Fig. 4a, b) and had no effect on lean mass (Supplementary Fig. 6a). ΔEpoR$_E$-mice showed increased hematocrit with EPO treatment[7] but not EPO regulation of body weight, fat mass or lean mass (Fig. 4a, b and Supplementary Fig. 6a), providing evidence that EPO effects on body weight and fat mass are mediated via non-erythroid EpoR. EPO treatment in WT-mice significantly changed lipid metabolism genes with increased expression of lipolysis genes, *Pparγ*, and *Lpl*, in scWAT compared with PBS treatment, but not in ΔEpoR$_E$ scWAT (Fig. 4c). Increase in *Pparγ* and *Lpl* expression with EPO treatment was also observed in eWAT, BAT, skeletal muscle, and liver of WT-mice and the

increases were lower compared with scWAT (Fig. 4c–e and Supplementary Fig. 6b, c). Minimal differences in corresponding expression of these lipolysis genes were observed with EPO treatment in ΔEpoR$_E$-mice (Fig. 4c–e and Supplementary Fig. 6b, c). EPO treatment also decreased expression of lipogenic genes, *Acc1, Acc2, Fas, Lipin1, Srebf1*, and *Scd1* in scWAT, eWAT, BAT, skeletal muscle, and liver in WT-mice, with decreases greatest in WT scWAT (Fig. 4c–e and Supplementary Fig. 6b, c). The absence of corresponding changes in expression of lipogenic genes in ΔEpoR$_E$-mice further underscores the role of non-erythroid EpoR in mediating EPO regulation of fat mass and lipid metabolism.

EPO treatment showed the greatest changes in lipolysis and lipogenic-associated gene expression in WAT, especially scWAT in WT-mice. To show that EPO regulation of lipid metabolism could be mediated mainly via adipose tissue EpoR expression, mice with adipose tissue knockout of EpoR using the Adiponectin-Cre transgene (EpoR$^{Adiponectin-KO}$-mice) were treated with EPO (subcutaneous injection three times per week for 3 weeks) beginning at postnatal 21 days. EPO did not affect body weight or fat mass (Fig. 4f, g) and did not significantly change expression of lipolytic or lipogenic-

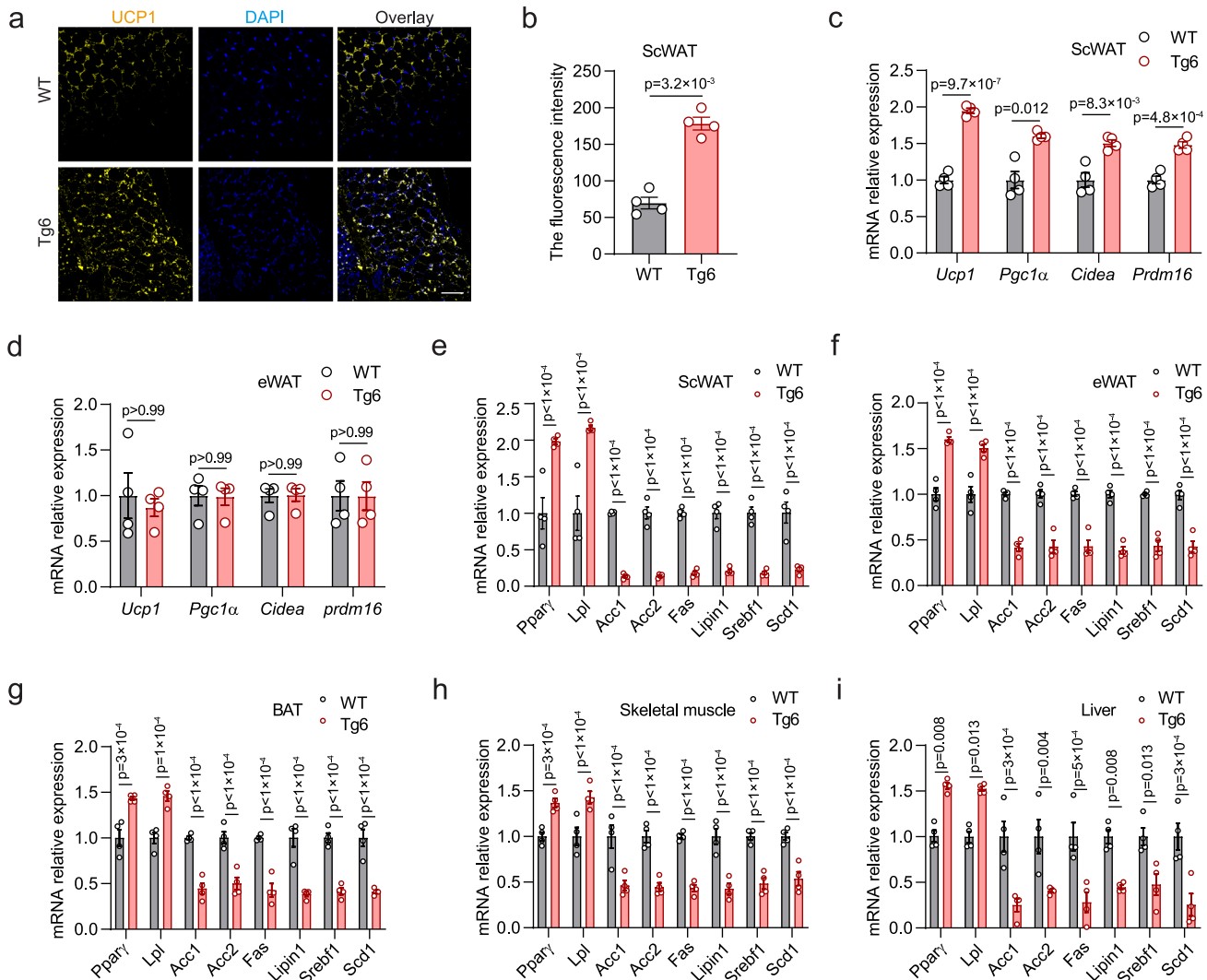

**Fig. 3 | Expression of lipid metabolism-associated genes in Tg6-mice with high EPO compared with WT-mice. a, b** Images show staining for uncoupling protein 1 (UCP1; yellow) and DAPI (blue) in scWAT from WT-mice and Tg6-mice (**a**) and quantification of fluorescence intensity by ImageJ (**b** ($t$, df = 8.656, 3)). **c, d** Two-sided unpaired $t$-test. Scale bar: 200 μm. Gene expression of BAT-associated genes Ucp1, peroxisome proliferator-activated receptor-γ coactivator 1α (Pgc1α), cell death-indicating DNA fragmentation factor a-like effector A (Cidea) and Prdm16 in scWAT (**c** $F_{(3, 9)}$ = 14.44) and eWAT (**d** $F_{(3, 9)}$ = 0.1504) from WT-mice and Tg6-mice. **e–i** Gene expression was determined by real-time quantitative PCR (qPCR) for

peroxisome proliferator-activated receptor γ (PPARγ), lipoprotein lipase (LPL), acetyl-CoA carboxylases (ACC1, ACC2), Fas, Lipin1, sterol regulatory element binding protein 1 (Srebf1), and stearoyl-CoA desaturase 1 (Scd1) for scWAT (**e** $F_{(7, 21)}$ = 41.43), eWAT (**f** $F_{(7, 21)}$ = 52.60), BAT (**g** $F_{(7, 21)}$ = 32.11), skeletal muscle (**h** $F_{(7, 21)}$ = 33.81), and liver (**i** $F_{(7, 21)}$ = 14.66) for WT-mice and Tg6-mice. Bars indicate WT-mice (gray) and Tg6-mice (red) mice. **b–i:** $n$ = 4 mice/group, the result shown in figure represent one of three independent experiments. Data shown as mean ± SEM. $p$ values are indicated. **b:** $t$-test; **c–i:** two-way ANOVA with Bonferroni's multiple comparisons test. Source data are provided as a Source Data file.

associated gene expression in scWAT, skeletal muscle, liver, eWAT, and BAT of EpoR$^{Adiponectin-KO}$-mice (Fig. 4h–j and Supplementary Fig. 7a, b). We also generated an alternate mouse model with knockout of EpoR in adipocytes using aP2-Cre (EpoR$^{aP2KO}$-mice), which also expresses in other cell types including macrophages[30]. EpoR$^{aP2KO}$-mice treated with EPO did not show change in body weight, fat or lean mass, or significantly alter expression of lipolytic or lipogenic-associated gene expression in scWAT, eWAT, BAT, skeletal muscle, and liver (Supplementary Fig. 7c–j). Therefore, EPO–EpoR axis regulation of fat metabolism was mediated mainly by EpoR on adipose tissue.

The EpoR agonist ARA290 is proposed to promote non-erythroid EpoR activity without increasing EPO stimulated erythropoiesis. As observed with EPO treatment, EpoR agonist ARA290 in WT-mice decreased fat mass and body weight and improved glucose tolerance, although, unlike EPO treatment, ARA290 did not increase hematocrit

(Fig. 5a–g). As with EPO treatment, ARA290 treatment in WT-mice changed lipid metabolism gene expression, significantly increased expression of lipolysis genes, *Pparγ*, and *Lpl*, and significantly decreased expression of lipogenic genes, *Acc1, Acc2, Fas, Lipin1, Srebf1*, and *Scd1* in scWAT, eWAT, BAT, skeletal muscle and liver compared with saline treatment (Fig. 5h–j and Supplementary Fig. 8a, b). Conversely, treatment with EpoR antagonist EMP9 decreased lipolysis gene expression and increased lipogenic gene expression compared with saline treatment in scWAT, eWAT, BAT, skeletal muscle, and liver (Fig. 5h–j and Supplementary Fig. 8a, b). Treatment with EMP9 and the combination of ARA290 and EMP9 in WT-mice did not affect hematocrit or show improvement in body weight, fat mass or glucose tolerance (Fig. 5a–g), and treatment with the combination of ARA290 and EMP9 did not affect lipid metabolism gene expression compared with saline treatment (Fig. 5h–j and Supplementary Fig. 8a, b). ARA290 treatment in WT-mice on HFD showed a greater improvement in

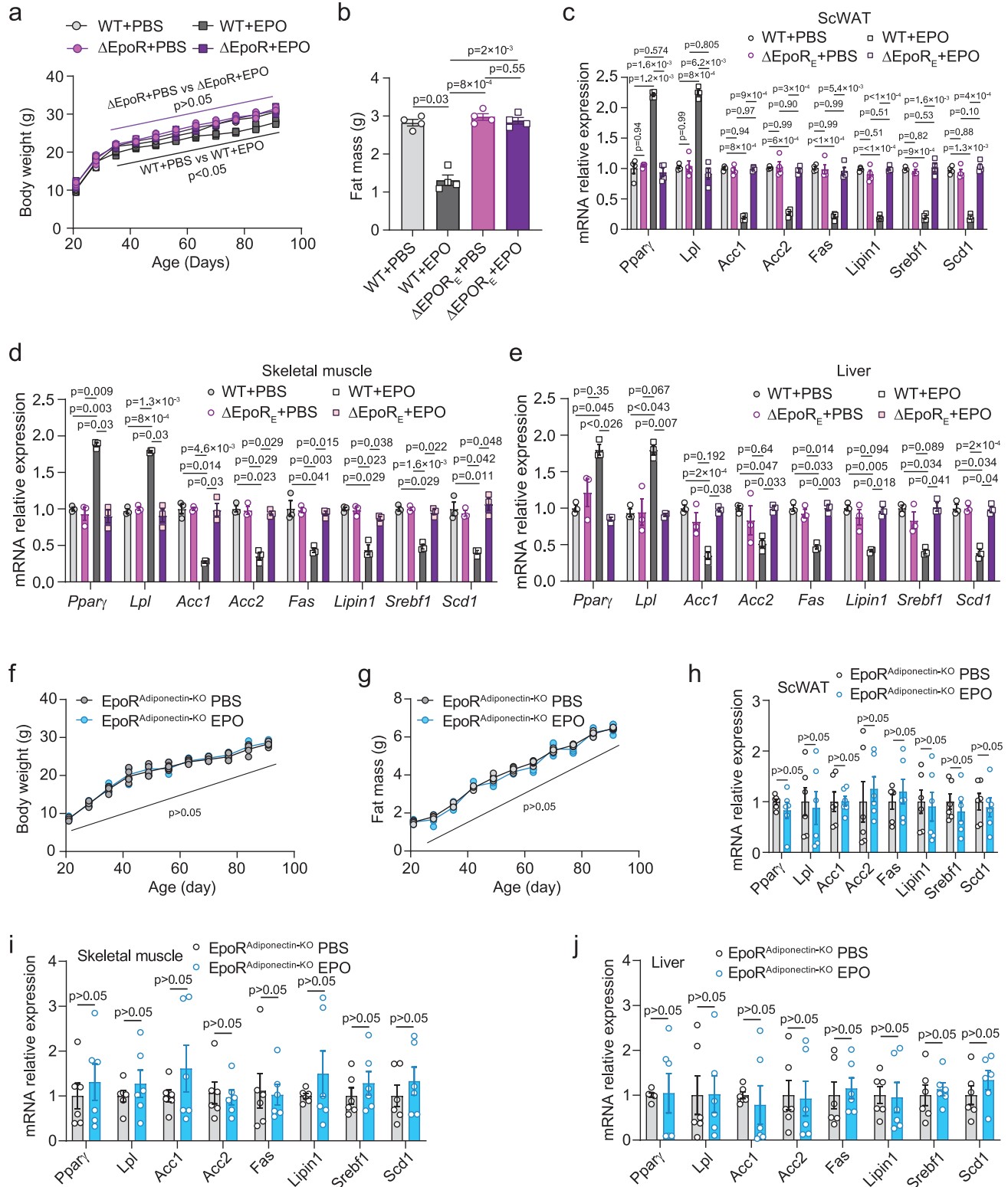

metabolism with decreased fat mass/body weight and improved glucose tolerance, and protection against diet-induced obesity, also without change in hematocrit (Supplementary Fig. 9a–f), and accentuated the differences in lipid metabolism gene expression (Supplementary Fig. 9g–k). These data provide additional evidence that EPO regulation of metabolism and glucose tolerance is not dependent on EPO stimulated erythropoiesis, result from EpoR response in non-erythroid tissue and can be simulated by activation of the EpoR heterodimer.

**EPO activation of EpoR modulated expression of lipolytic and lipogenic gene expression in subcutaneous white adipose tissue through RUNX1**

Among non-erythroid tissues, EPO had the greatest effect on expression of lipolytic and lipogenic gene expression in scWAT in Tg6-mice and in EPO-treated WT-mice (Figs. 3, 4 and Supplementary Figs. 6, 7). EPO treatment in ΔEpoR_E, EpoR^Adiponectin-KO or EpoR^aP2KO mice increased hematocrit but did not affect lipolytic or lipogenic gene expression in scWAT, BAT, skeletal muscle, or liver (Fig. 4 and Supplementary

**Fig. 4 | EPO regulation of lipid metabolism gene expression in WT-mice was not evident in ΔEpoR$_E$ mice or mice that lack EpoR in WAT. a, b** Body weight (**a** $F_{(2.61, 7.82)} = 4.15$) and fat mass (**b** $F_{(1.225, 3.675)} = 69.91$) were determined for WT-mice treated with PBS (light gray) or EPO (dark gray) and ΔEpoR$_E$ mice treated with PBS (light purple) or EPO (purple) for 3 weeks beginning at postnatal 21 days. **c–e** WT-mice were treated with PBS (light gray) or EPO (dark gray) and ΔEpoR$_E$ mice were treated with PBS (light purple) or EPO (purple) for 3 weeks beginning at postnatal 21 days and gene expression determined for *Ppary, Lpl, Acc1, Acc2, Fas, Lipin1, Srebf1,* and *Scd1* by qPCR in the scWAT (**c** $F_{(2.397, 7.192)} = 113.7$), skeletal muscle (**d** $F_{(1.586, 3.173)} = 44.39$), and liver (**e** $F_{(1.884, 3.769)} = 15.88$). **f, g** Body

weight (**f** $F_{(10, 50)} = 0.577$) and fat mass (**g** $F_{(10, 50)} = 2.661$) were determined for EpoR$^{Adiponectin-KO}$ mice treated with PBS (gray) or EPO (blue). **h–j** EpoR$^{Adiponectin-KO}$ mice were treated with PBS (gray) or EPO (blue) for 3 weeks beginning at postnatal 21 days and gene expression for *Ppary, Lpl, Acc1, Acc2, Fas, Lipin1, Srebf1,* and *Scd1* in scWAT (**h** $F_{(7, 35)} = 0.3032$), skeletal muscle (**i** $F_{(7, 35)} = 0.4291$), and liver (**j** $F_{(7, 35)} = 0.2572$) determined by qPCR. Data shown as mean ± SEM. **a–c** $n = 4$; **d, e** $n = 3$; **f–j** $n = 6$, **b–e, h–j**: the $n$ means mice per group. The result shown in figure represents one of three independent experiments. $p$ values are indicated. One-way or Two-way ANOVA with Bonferroni's multiple comparisons test. Source data are provided as a Source Data file.

Figs. 6, 7). We searched for potential regulatory motifs by which the EPO–EpoR axis may regulate transcription of lipid metabolism genes in scWAT. In total, 3000 bp of 5′ promoter sequences of corresponding lipolytic and lipogenic genes regulated by increased EPO were analyzed by the multiple sequence alignment program Clustal Omega[31,32]. Two conserved motifs from the promoter regions of lipolytic and lipogenic genes were identified by MEME consisting of 44 bp (potential RUNX binding motif) and 29 bp (potential IRF binding motif) (Fig. 6a, b). Gene expression of IRF family members in scWAT of WT-mice and Tg6-mice was not different (Supplementary Fig. 10a). Expression of RUNX1, RUNX2, and RUNX3 genes in scWAT was also similar in WT-mice and Tg6-mice (Supplementary Fig. 10b). Since RUNX1 expression has been linked to the browning of white fat[33] we examined RUNX1 protein levels in scWAT. Western blotting using RUNX1 specific antibody showed that Tg6-mice had increased RUNX1 protein level in scWAT compared with WT-mice (Fig. 6c). RUNX1 protein levels were not different in Tg6 eWAT, BAT, skeletal muscle, and liver compared with WT-mice (Fig. 6d), indicating that high EPO specifically affects RUNX1 protein in scWAT. EPO treatment in WT-mice similarly increased RUNX1 protein in scWAT of WT-mice but not in scWAT of ΔEpoR$_E$-mice (Fig. 6e, f). Treatment with the non-erythroid EpoR agonist ARA290 in WT-mice did not affect RUNX1 mRNA expression in scWAT but increased scWAT RUNX1 and CBFβ protein compared with control saline treatment (Supplementary Fig. 11a–c). To determine if treatment with RUNX1-inhibitor could block the increase in scWAT RUNX1 in Tg6-mice, we treated mice (P60) with RUNX1-inhibitor Ro5-3335 at 5 mg/kg by subcutaneous injection every other day for six times and tissues were harvested after 2 weeks treatment. RUNX1 mRNA in scWAT was not affected by RUNX1-inhibitor treatment (Supplementary Fig. 10c). However, RUNX1-inhibitor attenuated the increase in RUNX1 protein in scWAT of Tg6-mice compared with WT-mice (Fig. 6g). Potential for RUNX1 protein to regulate lipolytic and lipogenic gene expression by binding to the RUNX1 binding motif in the respective gene promoters (Fig. 6a, b) was assessed by ChIP analysis (Fig. 6h). ChIP analysis showed RUNX1 protein binding in the promoters of lipolytic genes, *Ppary* and *Lpl*, and lipogenic genes, *Acc1/2, Fas, Lipin1, Srebf1,* and *Scd1*, in scWAT, eWAT, BAT, liver, and skeletal muscle in WT-mice (Fig. 6h). However, RUNX1 protein levels were elevated in scWAT of Tg6-mice, but were not different in eWAT, BAT, liver, and skeletal muscle of WT-mice and Tg6-mice (Fig. 6c, d). These results suggest that EPO regulation of RUNX1 may affect the fat metabolism of other tissues mainly by affecting the metabolism of scWAT.

We used luciferase reporter gene constructs to demonstrate that RUNX1 could regulate transcription via the RUNX1 binding motifs from the promoter regions of lipid metabolism genes and that activation of the EPO–EpoR axis further increased RUNX1 regulatory activity. RUNX1 expression plasmid was co-transfected with luciferase constructs containing RUNX1 binding motifs identified from lipolytic or lipogenic genes into HEK293T cells and altered luciferase activity by twofold or more (Fig. 6a, i and Supplementary Table 3). RUNX1 expression plasmid co-transfected with reporter gene constructs containing RUNX1 binding motif from lipolytic genes (PPARγ-Luc and LPL-Luc) more than doubled the luciferase activity (Fig. 6i). Activation

of the EPO–EpoR axis by adding co-transfection of EpoR expression plasmid followed by EPO treatment further increased RUNX1 regulation of luciferase activity (Fig. 6i). In marked contrast, luciferase activity for constructs containing RUNX1 binding motifs of lipogenic genes (ACC1-Luc, ACC2-Luc, FAS-Luc, Lipin1-Luc, Srebf1-Luc, and Scd1-Luc) decreased by threefold with co-transfection of RUNX1 expression plasmid (Fig. 6i). Activation of the EPO–EpoR axis further decreased RUNX1 suppression of luciferase activity (Fig. 6i and Supplementary Table 3). Without RUNX1 expression plasmid, transfection with EpoR expression plasmid with and without EPO treatment did not affect luciferase activity of the reporter gene constructs containing the RUNX1 binding motifs (Supplementary Fig. 12a and Supplementary Table 3). Mutation of the RUNX1 binding site in the luciferase reporter gene constructs abrogated RUNX1 regulated activity (Supplementary Fig. 12b). These data provide evidence for EPO–EpoR–RUNX1 dependent transcription activity of lipid metabolism genes, and for the associated enhancer activity of the RUNX1 binding motif in the promoter region of lipolytic genes and silencer activity of the RUNX1 binding motif in the promoter region of lipogenic genes. The greatest change in luciferase activity of the reporter gene constructs containing the RUNX1 binding motifs were observed with co-transfection of RUNX1 and EpoR expression plasmids followed by EPO treatment.

To demonstrate RUNX1 regulation of lipolytic and lipogenic gene expression in scWAT, WT and Tg6-mice were treated with RUNX1-inhibitor Ro5-3335 diluted in saline containing DMSO. Ro5-3335 administration attenuated the changes observed in Tg6-mice associated with high EPO on expression of lipid metabolism genes (Fig. 6j). There were no significant differences in lipolytic gene expression and reduced or no significant differences in lipogenic gene expression between WT and Tg6 after Ro5-3335 treatment in scWAT (Fig. 6j), or in eWAT, BAT, skeletal muscle or liver (Supplementary Fig. 13a–d). RUNX1-inhibitor treatment did not significantly alter glucose tolerance test (GTT) in WT-mice and decreased glucose tolerance in Tg6-mice (Fig. 6k, l). No differences in GTT or in expression of lipid metabolism genes in scWAT, eWAT, BAT, skeletal muscle or liver in WT-mice treated with saline or DMSO vehicle control were observed (Supplementary Fig. 14a–g). Together, these results suggest that increase activation of the EPO–EpoR axis affects expression of RUNX1 protein level that, in turn, regulates expression of lipolytic and lipogenic gene expression in scWAT in mice, especially in Tg6-mice. Metabolic activity of scWAT can affect fat metabolism of other organs, and scWAT response to high EPO associated with Tg6-mice or with WT-mice treated with EPO likely contributes to eWAT, BAT, skeletal muscle, and liver EPO response.

## EPO–EpoR axis inhibited FBW7 ubiquitylation and subsequent RUNX1 degradation

Ubiquitylation, enzymatic posttranslational modification that adds the small molecule ubiquitin to a substrate protein, contributes importantly to protein homeostasis, regulates cellular activity of many proteins and, with polyubiquitylation, targets proteins for proteasome degradation. RUNX1 is a critical transcription factor for definitive hematopoietic stem cells and hematopoietic differentiation to myeloid and lymphoid lineages[34]. RUNX1 is rapidly degraded

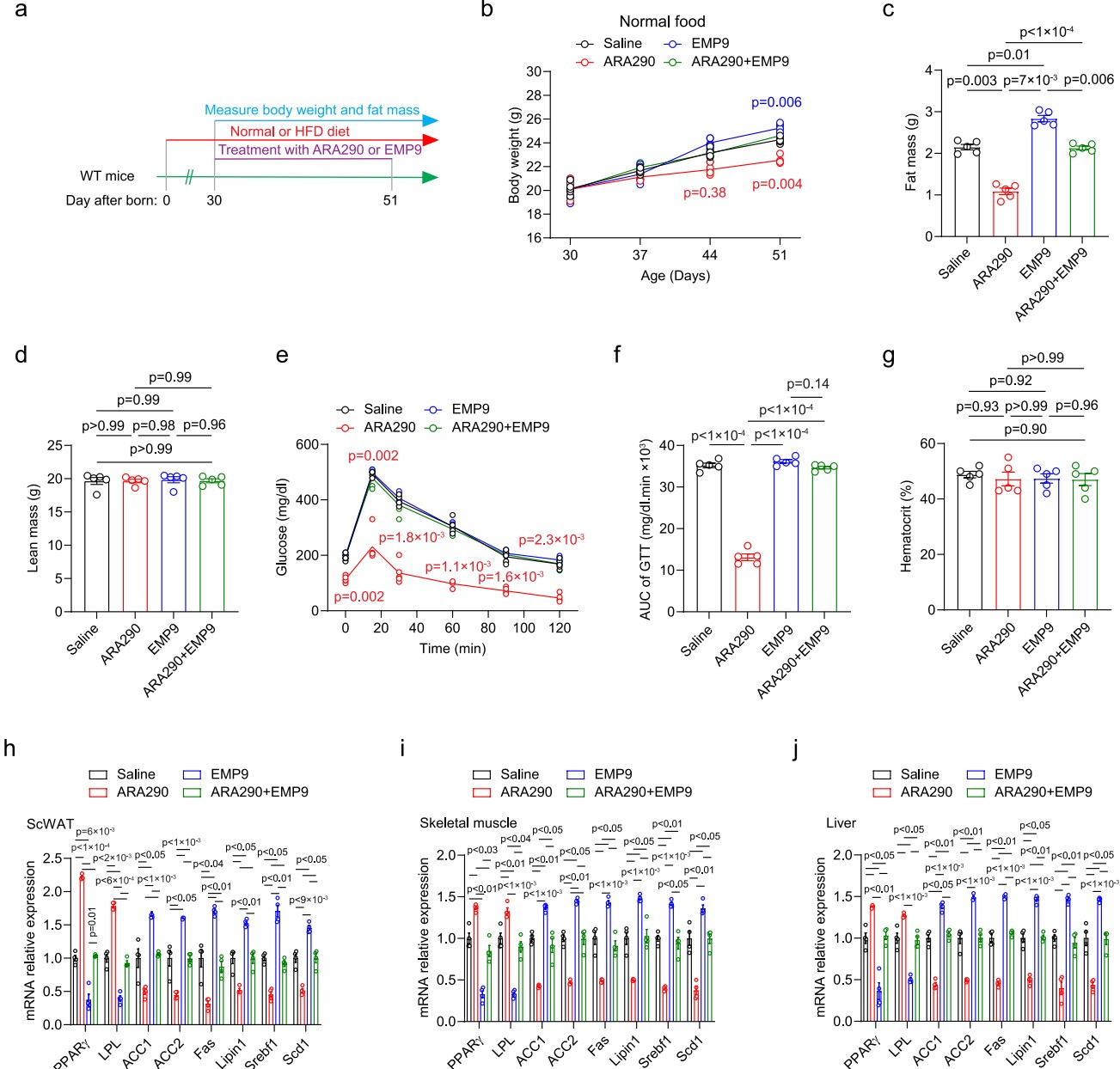

**Fig. 5 | EpoR agonist ARA290 treatment in WT-mice decreased fat mass and increased glucose tolerance, without increasing EPO stimulated erythropoiesis.** Non-erythropoietic EpoR agonist ARA290, but not EpoR antagonist EMP9 mimics the EPO effect on fat accumulation in the WT-mice. **a** Experiment set up chart. **b–g** WT-mice were fed normal chow diet and treated with Saline (black), ARA290 (red), EMP9 (blue), or ARA290 + EMP9 (green) for 3 weeks and body weight (**b** F(3.187, 12.75) = 7.169), fat mass (**c** F(1.907, 7.627) = 100.2), lean mass (**d** F(1.956, 7.823) = 0.0441), glucose tolerance (GTT) (**e** F(2.632, 10.53) = 15.13) and area under the curve (AUC for GTT) (**f** F(2.255, 9.019) = 347.2), and hematocrit

(**g** F(1.932, 7.726) = 0.2888) were determined. **h–j** Lipid metabolism genes expression in the scWAT (**h** F (2.568, 7.703) = 161.1), skeletal muscle (**i** F(2.762, 8.285) = 74.57), and liver (**j** F(2.748, 8.244) = 69.77) was assayed with qRT-PCR in WT-mice feed normal chow diet and treated with ARA290, EMP9, or ARA290 + EMP9; saline treated group was used as control. **b–j:** n = 5 mice/group. The result shown in figure represents one of three independent experiments. Data shown as mean ± SEM. One-way or Two-way ANOVA with Bonferroni's multiple comparisons test. Source data are provided as a Source Data file.

by posttranslational ubiquitylation through the ubiquitin-proteasome pathway[34]. Since RUNX1 protein was higher but mRNA level not different in scWAT of Tg6-mice compared with WT-mice (Fig. 6c and Supplementary Fig. 10b), increased RUNX1 protein may reflect EPO inhibition of RUNX1 ubiquitylation and degradation in scWAT of Tg6-mice. E3 ubiquitin ligases are one of three enzymes required for ubiquitylation. We determined the potential for high EPO in Tg6-mice to alter scWAT expression of several E3 ubiquitin ligases (Fig. 7a). E3 ubiquitin ligases, MARCH1[35], MARCH5[36], TRIM family members (11[37], 21[38], 23[39], 26[40], 27[41], 32[42]), MYLIP[43], MKRN1[44],

COP1[45], and PELI3[46] were expressed similarly in scWAT of Tg6-mice and WT-mice. However, FBXW7 E3 ubiquitin ligase mRNA expression was significantly reduced (Fig. 7a) and FBW7 protein level encoded by the FBXW7 gene[47] was decreased (Fig. 7b) in scWAT of Tg6-mice compared with WT-mice. Corresponding level of K48-linked poly-ubiquitin signal was also decreased in scWAT of Tg6-mice compared with WT-mice (Fig. 7b). These data suggest that FBW7 may control adipocyte metabolism by targeting RUNX1 for degradation by the ubiquitylation signaling pathway and that EPO disrupts this activity by decreasing FBXW7 mRNA and FBW7 protein in scWAT. Thus,

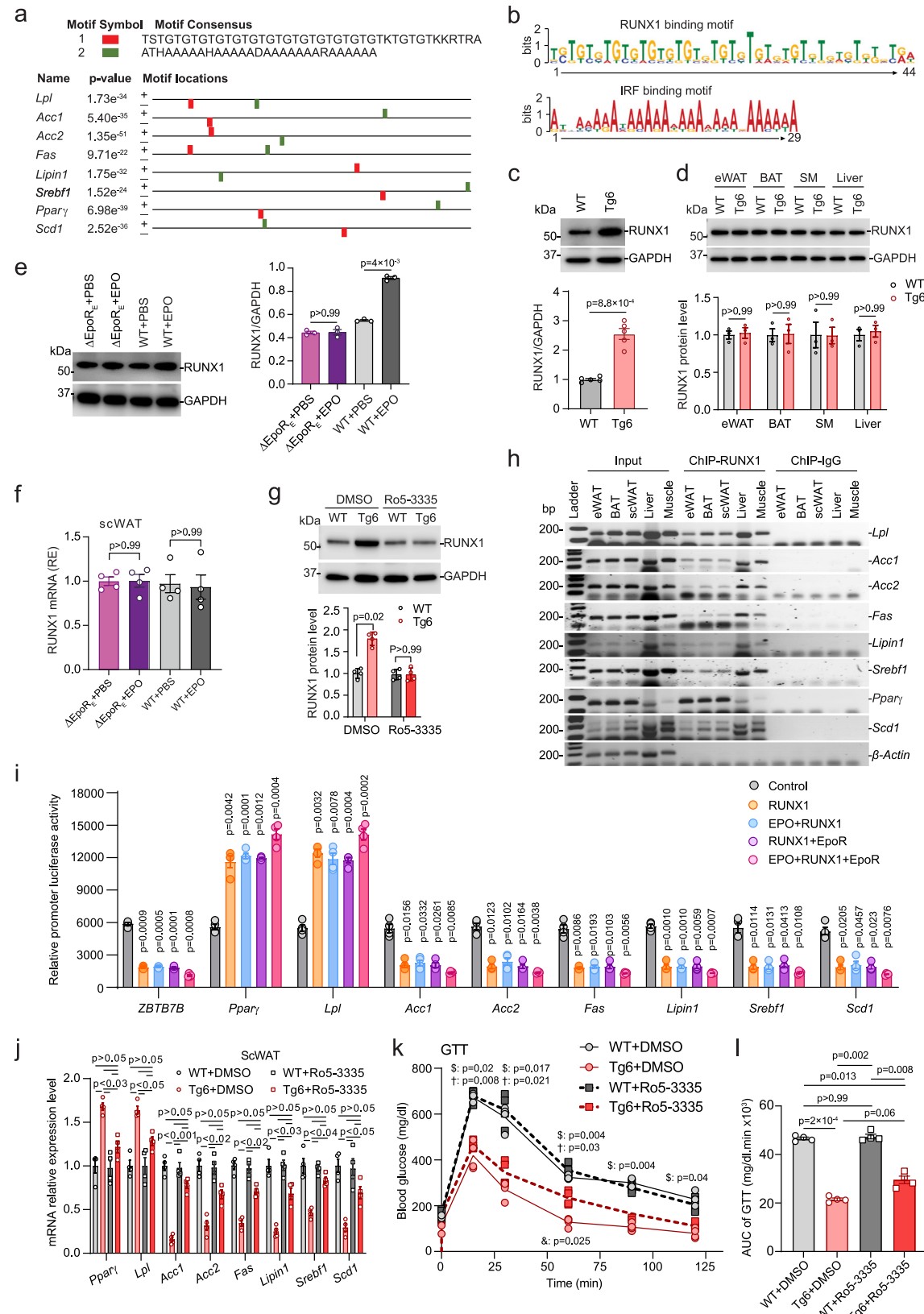

FBW7 would be an important regulator of energy and lipid metabolism.

To demonstrate that EPO can interfere with FBXW7 driven ubiquitylation and inhibit RUNX1 degradation mediated by EpoR, we used HEK293T cells as a model system to assess relative protein stability of RUNX1 by activation of the EPO–EpoR axis. Combinations of EpoR,

FBW7, and RUNX1 expression plasmids were transfected into HEK293T cells and treated with EPO or PBS. Proteins targeted for degradation by the ubiquitin proteasome system are tagged by poly-ubiquitylation at via enzymatic covalent attachment of small 76 amino acid ubiquitin monomers linked at lysine position 48 (K48-Ub) that require E2 ubiquitin-conjugating enzyme and E3 ubiquitin ligase[48]. We

**Fig. 6 | Regulation of lipid metabolism-associated gene expression in scWAT by EPO assessed by conserved motif identification. a–d** Promoter sequences of lipid metabolism genes in Tg6 scWAT were analyzed using Clustal Omega and MEME. RUNX motif (**a** red; **b** upper); IRF motif (**a** green; **b** lower). Western blotting for RUNX1 protein in WT (gray) and Tg6 (red) scWAT (**c**, $n = 5$, $t = 8.151$, df = 8, two-sided unpaired $t$-test), eWAT, BAT, skeletal muscle, and liver (**d**, $n = 3$, F(3, 6) = 0.06043), data shown as mean ± SEM. **e, f** Western blotting for RUNX1 protein (**e**, $n = 3$, F(1.13, 2.25) = 230) and qPCR for relative gene expression (**f**, $n = 4$, F(1.216, 3.649) = 0.2208) in $\Delta$EpoR$_E$ and WT-mice treated with EPO for 3 weeks ($\Delta$EpoR$_E$, purple; WT, grey), quantified with GAPDH control by ImageJ. **g** Western blotting for scWAT RUNX1 protein in WT (dark gray) and Tg6 (red) mice treated with RUNX1 inhibitor Ro5-3335 (5 mg/kg) or DMSO control (WT, light gray; Tg6, light red), $n = 4$, F(1, 3) = 16.15, data shown as mean ± SD. **h** RUNX1 protein binding to proposed promoter regions in lipid metabolism genes by ChIP analysis with β-actin control for WT scWAT (Input DNA, ChIP-RUNX1, ChIP-IgG), original images in Supplementary Fig. 16.

**i** Transcription activity for RUNX1-binding regions from promoters inserted into luciferase-reporter plasmids, with ZBTB7B control for RUNX1-dependent silencer activity in HEK293T cells, co-transfected with various plasmids, F(2.792, 8.376) = 91.56. **j** Lipid metabolism gene expression (qPCR) in scWAT of WT and Tg6-mice treated with RUNX1 inhibitor Ro5-3335 (Ro5-3335 treatment group, squares; WT, dark gray; Tg6, red) or DMSO control (circles; WT, light gray; Tg6, light red), F(2.709, 8.128) = 31.27. **k, l** Glucose tolerance test (GTT, **k** F(2.489, 7.468) = 10.55) and area under the curve (AUC, **l** F(2.045, 6.136) = 170.1) for mice treated with RUNX1 inhibitor (Ro5-3335) (WT, dark gray; Tg6, red) or DMSO control (WT, light gray; Tg6, light red). Comparisons indicated: $WT + DMSO vs Tg6 + DMSO; †WT + Ro5-3335 vs Tg6 + Ro5-3335; &Tg6 + DMSO vs Tg6 + Ro5-3335; mean ± SEM. **i–l** $n = 4$/group, the result shown in figure represent one of three independent experiments. One-way or two-way ANOVA with Tukey's (**i**) or Bonferroni's (**d–g, j–l**) multiple comparisons test. Source data are provided as a Source Data file.

assessed K48-Ub and K63-Ub (proteasome-independent ubiquitylation by K63-polyubiquitin), and plasmid derived expression of Flag-RUNX1, FBW7-Myc, and EpoR (Fig. 7c and Supplementary Fig. 15a). K48-polyubiquitin was detected with transfection of the FBW7 expression plasmid but not when FBW7 expression plasmid was co-transfected with activation of the EPO–EpoR axis (co-transfection of EpoR expression plasmid followed by EPO treatment) (Fig. 7c). Hence, activation of the EPO–EpoR axis markedly decreased FBW7 ubiquitin ligase activity and formation of K48-Ub polyubiquitin chain. K63-polyubiquitin was not affected with or without transfection of expression plasmids for FBW7, RUNX1 or EpoR. Flag-RUNX1 was readily detected with transfection of the RUNX1 expression plasmid. Co-transfection of RUNX1 expression plasmid with FBW7 expression plasmid greatly diminished RUNX1 protein level indicating decreased RUNX1 stability and increased RUNX1 degradation, consistent with activation of the ubiquitin proteasome system by increased production of FBW7. Co-transfection of expression plasmid for RUNX1 and FBW7 combined with activation of the EPO–EpoR axis (EpoR co-transfection and EPO treatment) abrogated the decrease in RUNX1 protein level by increased FBW7 protein (Fig. 7c and Supplementary Fig. 15a). EpoR protein levels from the EpoR expression plasmid were not affected by co-transfection with the other expression plasmids or by EPO treatment. Activation of the EPO–EpoR axis decreased FBW7 protein level expressed from the FBW7 plasmid. These results indicate that increased FBW7 E3 ubiquitin ligase that promotes the ubiquitin proteasome system increases RUNX1 degradation, and that activation of the EPO–EpoR axis decreases K48-polyubiquitin and increases RUNX1 protein stability. This provides evidence that the EPO–EpoR axis in scWAT regulates RUNX1 stability and identifies a mechanism for EPO regulation of fat metabolism mediated by inhibition of RUNX1 degradation by FBW7 ubiquitylation and by increased RUNX1 transcription regulation of associated lipolytic and lipogenic gene expression. These data together with EPO response in EpoR$^{Adiponectin-KO}$-mice and EpoR$^{aP2KO}$-mice showed that EPO regulation of fat metabolism in scWAT was mediated by EpoR and that RUNX1-stimulated activity in WAT by EPO could affect fat metabolism in other tissues and contribute to body weight and fat mass regulation independent of EPO stimulated erythropoietic response.

RUNX1 partners with CBFβ to enhance DNA binding and regulate gene expression[49]. RUNX1 protein is not detectable in embryonic extracts from CBFβ knockout mice while RUNX1 transcripts remain unchanged[50]. Although whole-body knockout of CBFβ is embryonic lethal, fat specific knockout of CBFβ generates mice with similar body weight as WT-mice with low body adiposity and adipokines, and a progressive loss of body fat with hyperglycemia and hyperinsulinemia and worsening of lipodystrophy with increasing age[51]. HEK293T cells express endogenous CBFβ (Supplementary Fig. 15b) that provides for RUNX1 stability and detection of Flag-RUNX1 with transfection of RUNX1 expression plasmid (Fig. 7c). Transfection of Flag-RUNX1

facilitates detection of endogenous CBFβ and co-transfection with CBFβ expression plasmid further increases Flag-RUNX1 detection (Fig. 7c). Co-transfection of RUNX1 expression plasmid with FBW7 expression plasmid activated the degradation pathway for RUNX1 and Flag-RUNX1 detection was decreased and ubiquitylation increased indicated by increased detection of K48-Ub even with co-transfection of CBFβ expression plasmid (Fig. 7c and Supplementary Fig. 15a). This suggests that RUNX1 was likely to be interacting with CBFβ and degraded by FBW7 more easily than protection by heterodimer formation with CBFβ. Activity of the degradation pathway for RUNX1 by FBW7 expression and increased ubiquitylation and intensity of K48-Ub were also evident in the presence of EPO signaling (EPO treatment with EpoR expression) with only endogenous CBFβ expression. However, the combination of increased CBFβ by co-transfection with CBFβ expression plasmid and EPO signaling (EPO treatment with EpoR expression) increased Flag-RUNX1 stability and decreased ubiquitylation and K48-Ub even with increased expression of FBW7 (Fig. 7c), suggesting that EPO signaling promotes RUNX1 interaction and protective heterodimerization with CBFβ and a decrease in RUNX1 degradation by FBW7.

## Discussion
Endogenous EPO contributes to regulation of metabolic function by interacting with EpoR. EpoR restricted to erythroid tissue ($\Delta$EpoR$_E$-mice) and deletion of EpoR in fat tissue (EpoR$^{aP2KO}$-mice) induce metabolic disturbances due to an imbalance between fat storage and energy expenditure resulting in the body storing more fat and burning less energy[7,8]. EpoR-tdTomato-Cre-mice revealed that among non-erythroid tissues, EpoR is expressed at high level in WAT and spleen, and to a lesser extent in BAT, skeletal muscle, liver and brain. Metabolic behavior of EpoR$^{Adiponectin-KO}$-mice and EpoR$^{aP2KO}$-mice illustrate the importance of WAT in metabolic response to endogenous EPO and to EPO treatment in expression regulation of lipolytic genes, *Ppary* and *Lpl*, and lipogenic genes, *Acc1/2*, *Fas*, *Lipin1*, *Srebf1* and *Scd1*. In other studies, EPO administration reduced blood glucose level and improved glucose tolerance in mouse models that include C57BL/6, ob/ob susceptible to diabetes and obesity, BALB/c, and *PTP1B*$^{-/-}$ associated with resistance to diabetes, and Tg6-mice at 3 months exhibited lower insulin levels and increased insulin sensitivity[5]. Alternatively, generation of EpoR adipocyte knockdown on a mixed background using aP2-Cre-mice did not show altered glucose or energy homeostasis in diet-induced obese mice at 8 months[52], suggesting that genetic background influences metabolic response to EPO. In a mouse model of kidney disease, increasing circulating EPO decreased serum triglyceride, and enhanced lipid catabolism and increased JAK2-STAT5 signaling in adipose tissue[53]. In hematologic patients, EPO treatment was also associated with decreased blood glucose levels[54].

PPARγ activation has been reported to reduce the delivery of fatty acids to liver and muscle, reduce fat synthesis, and inhibit lipid

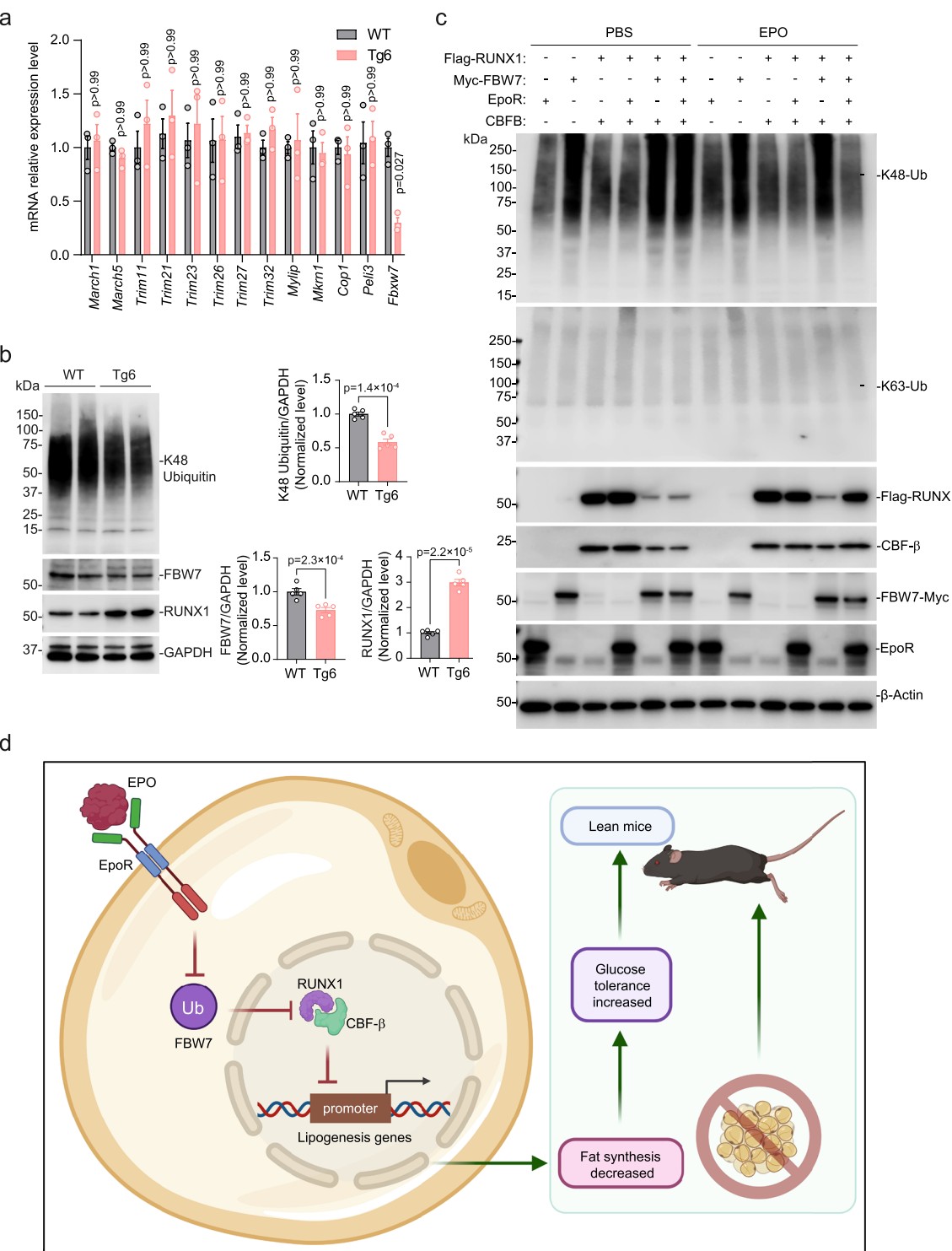

metabolism[55]. PPARγ can induce expression of LPL in adipocytes, promoting lipid metabolism and lowering blood lipid levels[55]. ACC1/2, FAS, and SCD affect lipid metabolism and transcriptional regulation of lipogenic gene Srebf1 mainly regulates the expression of fatty acid synthesis genes, Lipin1 and LDLR[56,57]. Adipocyte knockout of ACC1 or FAS in mice inhibits fatty acid synthesis, promotes browning and increases UCP1 in WAT[58]. Here, elevated EPO in mice on normal chow diet significantly increased lipolysis gene (*Ppary* and *Lpl*) expression in WAT, BAT, skeletal muscle, and liver in EPO-treated WT-mice and in Tg6-mice. This was accompanied by a corresponding decrease in lipogenic genes (*Acc1/2, Fas, Lipin1, Srebf1,* and *Scd1*) in WAT, BAT, skeletal muscle, and liver. Changes in genes

associated with lipid metabolism by elevated EPO were greatest in scWAT, and relatively high EpoR in adipose tissue among non-erythroid tissues suggests a specialized role of WAT for EPO-regulated energy storage. We found that in young Tg6-mice, high transgenic EPO levels promoted browning of WAT and increased brown fat associated gene expression in WAT. Hence, elevated EPO could regulate body weight and fat mass by increasing lipolytic *Ppary* and *Lpl* expression and decreasing lipogenic *Acc1/2, Fas, Lipin1, Srebf1,* and *Scd1* expression in WAT and promoting browning of white fat. Furthermore, EPO response in WAT contributed to regulation of lipid metabolism genes in BAT, skeletal muscle, and liver where EpoR expression is low.

**Fig. 7 | EPO activation of EpoR signaling in scWAT inhibited ubiquitin ligase FBW7, thereby increasing RUNX1 protein stability. a** E3 ubiquitin ligase-related proteins MARCH1, MARCH5, TRIM11, TRIM21, TRIM23, TRIM26, TRIM27, TRIM32, MYLIP, MKRN1, COP1, PELI3, and FBW7 relative gene expression were determined by qPCR for scWAT of WT-mice (gray) and Tg6-mice (red) ($n = 3$/genotype; the result shown in figure represents one of three independent experiments. $t$-test, $F_{(12, 24)} = 1.313$). **b** K48 Ubiquitin and FBW7 and RUNX1 protein expression were determined by western blotting with GAPDH as control (left) were determined for scWAT WT-mice (gray) and Tg6-mice (red) mice and were quantified by ImageJ (right). High transgenic EPO expression in Tg6-mice significantly inhibited FBW7 protein expression with a decrease in K48 Ubiquitin, and increased RUNX1 protein in scWAT. ($n = 5$/genotype; $t$-test). K48: $t = 7.564$, df = 8; FBW7: $t = 4.407$, df = 8; RUNX1: $t = 14.27$, df = 8. **c** HEK293T cells were transfected with expression plasmids for FBW7 (FBW7-Myc), and/or RUNX1 (Flag-RUNX1), and/or CBFβ, and/or EpoR and were treated with EPO (3 U/ml) or PBS for 24 h. The ubiquitylation of K48-Ub and K63-Ub, and protein levels of Flag-RUNX1, FBW7-Myc, CBFβ, and EpoR were determined by western blotting with β-actin as control. **d** Proposed model for EPO−EpoR axis protects against fat store by regulating lipogenic and lipolysis signaling pathway in the scWAT. Elevated EPO increases EPO binding to and activation of EpoR in scWAT resulting in reduction of ubiquitin ligase FBW7 and promotes RUNX1 stability and activity to regulate lipid metabolism and fat storage and to trigger lipolysis and promotes a lean phenotype in mice. Created with BioRender.com. Figure 7d Created with BioRender.com released under a Creative Commons Attribution-NonCommercial-NoDerivs 4.0 International license. Data shown as mean ± SEM. $p$ values are indicated. Two-way ANOVA with Bonferroni's multiple comparisons test (**a**), two-sided unpaired $t$-test (**b**). Source data are provided as a Source Data file.

Tg6-mice have elevated EPO in plasma and brain. Compared with plasma EPO levels in WT-mice, Tg6 EPO levels determined by radio-immunoassay were 12X in plasma, 5X in brain (about 17 times endogenous WT brain EPO), 2X to 3X in lung and not elevated in heart, kidney or liver[23,59]. Interestingly, Tg21 mice that overexpress transgenic EPO only in brain show no change in hematocrit and the metabolic effect of increased brain EPO on fat mass and body weight in male mice was apparent only after HFD feeding and was analogous to response to EPO intracerebroventricular pump in WT-mice on HFD[6]. Differences in expression of lipid metabolism genes observed between Tg6-mice and WT littermate control mice were comparable to differences in WT-mice between EPO and PBS treatment, suggesting that changes in expression of lipid metabolism genes between Tg6-mice and WT-mice may be largely related to the increased circulating EPO levels.

The mammalian genome contains three related RUNX genes, RUNX1, RUNX2, and RUNX3[60]. Transcription factor RUNX1 is required for embryonic endothelial-to-hematopoietic transition and generation of hematopoietic stem and progenitor cells[61,62]. RUNX1 plays an important role during leukemogenesis and RUNX1 is required for leukemia development by both RUNX1-RUNX1T1 and CBFB-MYH1[60]. In WAT, CDK6 negative regulation of white-to-beige-fat transition is mediated in part by suppression of RUNX1 that promotes expression of brown fat associated genes, *Ucp1* and *Pgc1α* that contain RUNX1 binding sites in their respective promoters[33].

Here we found that the EPO−EpoR axis regulated scWAT RUNX1 protein level and affected lipid metabolism in Tg6-mice and in EPO-treated WT-mice. Moreover, inhibition of RUNX1 in Tg6-mice significantly reduced lipolytic gene expression in WAT, BAT, skeletal muscle, and liver with a minimal effect on glucose tolerance. While the coordinated role of EPO−EpoR axis targeted RUNX1 in regulating WAT browning remains unclear, our results partially explain the role of EPO−EpoR−RUNX1 in lipid metabolism (Fig. 7d). We previously found that body weight gain and fat mass accumulation increased in ΔEpoR$_{E}$-mice that lack EpoR in non-hematopoietic tissue and in mice with loss of EpoR in adipose tissue (EpoR$^{aP2KO}$-mice)[8]. This suggests that EPO can affect lipid metabolism and weight gain primarily by regulating lipid metabolism in adipose tissue mediated by EpoR in WAT. Accordingly, EPO-treated male mice that lack of EpoR expression in adipocytes (EpoR$^{aP2KO}$) and adipocyte specific EpoR KO (EpoR$^{Adiponectin-KO}$) did not show the increase in *Pparγ* and *Lpl* or the decrease in *Acc1/2*, *Fas*, *Lipin1*, *Srebf1*, and *Scd1*expression in WAT, BAT, skeletal muscle, and liver that was observed in EPO-treated WT-mice. Additional work is required to clarify the broader contributions of EPO−EpoR and RUNX1 crosstalk to WAT metabolism including regulation of the brown fat program in WAT.

EPO decreased expression of FBXW7 which encodes FBW7 ubiquitin ligase that targets proteins including select transcription factors for ubiquitylation and proteasome degradation[63]. We observed that EPO−EpoR axis regulation of FBXW7 expression and FBW7 protein in scWAT was concomitant with increased RUNX1 protein that can affect lipid metabolism gene expression mediated by RUNX1 interaction. Specifically, RUNX1 binding motifs in lipolytic gene promoters acted as enhancers for RUNX1 stimulated transcription activation while RUNX1 binding motifs in lipogenic gene promoters acted as silencers for RUNX1 suppression of transcription. EpoR expression was required for this EPO stimulated activity in WAT which then affected expression of lipolytic and lipogenic genes expression in BAT, skeletal muscle, and liver resulting from decreased WAT FBXW7 expression and increased WAT RUNX1 stability (Fig. 7d). RUNX1 partners with CBFβ to enhance DNA binding and regulation of gene expression[49]. Co-transfection assays in HEK293T cells that express endogenous CBFβ demonstrated that increased expression and stability of RUNX1 were affected by expression of FBW7, increased expression of CBFβ and activation of the EPO−EpoR axis (Fig. 7c).

Limitations of the current study include the use of Tg6-mice with high chronic expression of transgenic human EPO in brain, lung and other tissues and the abnormally high chronic hematocrit that increases from 52% at 2 weeks to >80% by 5 weeks of age[23]. Tg6-mice have enhanced endothelial nitric oxide synthase activity to regulate the resultant elevated blood viscosity[21] that may provide an indirect EPO activity to improve metabolism as mice with overexpression of endothelial nitric oxide synthase exhibit resistance to diet-induced obesity and adipose tissue with increased metabolic activity and altered lipid metabolism[64]. To validate results linked to high EPO for lipid metabolism gene expression, both Tg6-mice and WT-mice treated with EPO were used. Adipose tissue expression of EpoR and adipose tissue response to EPO are important determinants of EPO metabolic regulation[8,9,22] and we focused on EPO stimulated regulation of fat mass, especially lipid metabolism in WAT. How EPO stimulation affects WAT transcriptome and proteome including other RUNX1 target genes, secreted adipokines and metabolites that may also affect metabolic response in other tissues have yet to be identified. While we considered EPO metabolic activity and EpoR expression in WAT, BAT, skeletal muscle and liver, the relationship between the EPO−EpoR axis and E3 ubiquitin ligase signaling and the tissue-specific mechanism of FBXW7 and RUNX1 in regulation of lipid metabolism in BAT, skeletal muscle and liver are unknown. We did not include other EpoR expressing tissues such as brain and central nervous system, immune cells/macrophages, cardiovascular system/heart/endothelium, and gastrointestinal tract[65] that are likely to contribute to EPO metabolic response. EPO treatment improves glucose metabolism in male and female mice and the current study was restricted to male mice because of the overlapping protective effect of estrogen against diet-induced obesity in female mice[6,20,22]. How these animal studies related to EPO stimulation in human to regulation of fat mass and lipid metabolism and the potential sex-dimorphic response[11,66] warrants further study. Clinical treatment of anemia in chronic kidney disease with oral hypoxia-inducible factor prolyl hydroxylase inhibitors that stimulate endogenous EPO production to increase hemoglobin may be as effective as erythropoiesis-stimulating agents[67,68], and the metabolic

responses to this new class of drugs remain to be determined. EPO mimetic peptides were designed to stimulate non-erythroid EPO-like activity without increasing hematocrit such as ARA290 that has been suggested to benefit human metabolic control and neuropathy[69–71], raising the potential for EPO mimetics to treat obesity and metabolic syndromes. Despite these limitations, our study provides new insights into how the EPO–EpoR axis regulates lipid metabolism and improves obesity. Our results suggest that the EPO–EpoR axis in non-hematopoietic tissue effects may simultaneously alleviate obesity associated metabolic derangements.

## Methods

### Mouse models
All the animal studies comply with all relevant ethical regulations, the animal experiment procedures were approved by the NIDDK Animal Care (ASP: K048-MMB-22) and Use Committee and followed NIH guidelines. Tg6-mice, overexpressing PDGFβ promoter-driven human *EPO* transgene[21] and WT-mice (Strain No.:000664, Jackson Laboratory, Bar Harbor, Maine) were on a C57BL/6 background[23]. Heterozygous male Tg6-mice were crossed with female WT-mice to obtain heterozygous male Tg6-mice and male WT littermates as controls. EpoR-tdTomato-Cre mice were generated with tdTomato and Cre knocked in at the end of the EpoR coding region, allowing independent expression of EpoR, tdTomato and Cre under control of the EpoR promoter[72]. Mice with EpoR restricted to erythroid tissue ($\Delta$EpoR$_E$) contain the *TgEpoR$_E$* transgene (mouse EpoR cDNA driven by GATA-1 hematopoietic regulatory domain) on a EpoR$^{-/-}$ background[73]. EpoR$^{Adiponectin-KO}$ mice were generated by crossing Adiponectin-Cre mice (Strain No.: 028020 C57BL/6 background) with EpoR$^{floxp/floxp}$ mice[74] that were backcrossed onto a C57BL/6 background for at least seven generations[8]. EpoR$^{aP2KO}$ mice were generated by crossing aP2-Cre mice (Strain No.: 005069, C57BL/6 background) with EpoR$^{floxp/floxp}$ mice. Because female mice produce estrogen, which has a protective effect against EPO-regulated fat accumulation, they were not included in this study[20]. Mice were maintained under a 12-h light/dark cycle with free access to food and water. The animal room was kept at a constant temperature of 23 °C with 40% humidity. Body weight was measured using the same scale/balance by the same person. WT male mice were divided into two groups, normal diet (Bio-Serv, Cat. F4031) group and a HFD (Bio-Serv, Cat. F3282) group. The mice in the normal diet group were further divided into four groups (5/group). Mice in the HFD group were also divided into four groups (5/group). Mice in the normal diet group and the HFD group were intraperitoneally injected with saline, EpoR agonist ARA290 (100 µg/kg) (Peptide Sciences, Henderson, NV), EpoR blocking peptide EMP9 (2.5 mg/kg) (catalog # MBS8243539; MyBioSource, Inc., San Diego, CA) and ARA290 + EMP9 every other day per week for 3 weeks. All mice had free access to diet 24 h a day throughout the experiments. If the mice were tested from birth to weaning, their parents were provided with either a normal or HFD. Once the mice were weaned, they continued to have free access to their designated diet all day. The mice were monitored weekly for body weight, fat mass, and lean body mass. Three weeks after injection, mice were tested for GTT and hematocrit. The mice were euthanized using a vaporizer (Model: V3000PS, PARKLAND SCIENTIFIC, Coral Spring, FL) equipped with an induction chamber and waste gas scavenger. Iso-flurane was gradually administered at concentrations exceeding 5% in oxygen until respiratory arrest persisted for more than 2 min. Subsequently, the chamber was flushed with oxygen only, and the mice were removed and cervical dislocation used to ensure euthanasia. Mice tissues harvested to detect expression levels of genes associated with fat metabolism in ScWAT, eWAT, BAT, skeletal muscle and liver. Body composition was measured using the EchoMRI 3-in-1 (Echo Medical Systems, Houston, TX) in conscious, immobilized mice. Male mice were used to study the EPO effects on the lipid metabolism because EPO treatment can reduce body fat and weight gain in male mice,

especially during HFD feeding. This is observed in female mice only after ovariectomy due to the protective effect of estrogen to diet-induced obesity[6,20,22].

### Glucose and insulin tolerance tests
GTT and insulin tolerance test (ITT) were carried out on mice fasted overnight, injected intraperitoneally with glucose (Cat.G7528, Millipore-Sigma; 2.0 g/kg body weight) or insulin (Humulin R, 0.75 U/kg), respectively, and blood glucose measured before (0 min) and up to 120 min after injection of glucose from tail vein bleeds (Alphatrak 2 glucometer, Zoetis, Parsippany, NJ). Blood insulin secretion was measured in 30 µl of blood collected before (0 min) and up to 120 min after glucose injection; serum was prepared by $1000 \times g$ centrifuge 15 min and frozen on dry ice immediately for later assessment[7].

### Histology and immunofluorescence staining
Mouse tissues were fixed in 10% formalin (Cat. HT501128, Millipore-Sigma, Burlington, MA) and then cryosectioned at 20 µm thickness using the Cryostat Leica CM1950. Hematoxylin and eosin (H&E)-stained sections were performed with H&E Staining Kit (Hematoxylin and Eosin) (Cat. ab245880, Abcam) and imaged to measure fat cell sizes. H&E-sectioned slides were scanned by a NonoZoomer digital scanner and analyzed with NDP viewing software and ImageJ. At least 2000 cells were counted for each group. Oil Red O staining was performed with Oil Red O Stain Kit (Lipid Stain) (Cat. ab150678, Abcam). For paraffin sections, antigen retrieval was performed by boiling in sodium citrate buffer (Cat. ab93678, Abcam, Boston, MA) for 30 min after paraffin sections were deparaffinized with xylene 3 × 5 min and 100% ethanol, 95% ethanol, 70% ethanol, 50% ethanol for 10 min each, and rehydrated with deionized water for 2 × 5 min. Mouse tissue sections were washed with 1× PBS/0.1% Triton X-100 for 3 × 5 min, blocked with 10% normal goat serum (Cat. 50197Z; Thermo Fisher Scientific, Waltham, MA), and incubated at 4 °C overnight. Sections were incubated with Goat Anti-Mouse Erythropoietin R Antigen Affinity-purified Polyclonal Antibody (Cat. AF1390, R&D Systems, Minneapolis, MN) at 15 µg/mL incubated at 4 °C overnight. Tissues were stained using the Alexa Fluor™ 488 and/or Alexa Fluor™ 568-conjugated Donkey Anti-Goat IgG Secondary Antibody (Alexa Fluor™ 488: Cat. A-11055; Alexa Fluor™ 568: Cat. A-11057, Thermo Fisher Scientific) and counterstained with DAPI (blue) (Cat. 62248, Thermo Fisher Scientific). UCP1 Antibody (Cat. MAB6158, Novus Biologicals) was used at 10 µg/mL and incubated at 4 °C overnight. Tissues were stained using the Alexa Fluor™ 488 and/or Alexa Fluor™ 568-conjugated Goat anti-mouse IgG Secondary Antibody (Alexa Fluor™ 488: Cat. A-11001; Alexa Fluor™ 568: Cat. A-11004; Thermo Fisher Scientific) and counterstained with DAPI (blue). Primary antibodies and secondary antibodies used in this study are indicated in Supplementary Table 1.

### Western blotting
Mouse tissues or cells were lysed in RIPA Lysis and Extraction Buffer (Cat. 89900, Thermo Fisher Scientific) supplemented with 1× protease inhibitor cocktail (Cat. P8340; Sigma-Aldrich, St. Louis, MO). The total protein concentration was determined by a Pierce™ BCA Protein Assay Kit (Cat. 23225; Thermo Fisher Scientific). About 25 µg proteins were loaded onto 4–20% tris-glycine SDS/PAGE gel. Proteins were separated using 120 V/400 mA for 65 min, transferred to PVDF membranes using the Trans-Blot Turbo Transfer System (Cat. 1704150, Bio-Rad, Hercules, CA) and visualized using protein-specific antibodies (Supplementary Table 1). The membranes were washed with 1× PBST for 3 × 5 min, blocked with 5% fat-free milk for 1 h, and incubated with primary antibodies overnight at 4 °C. Next day, the membrane was washed with 1× PBST for 3 × 5 min, incubated with secondary antibodies for 2 h at room temperature, and then developed using Clarity or Clarity Max ECL western blotting substrates (Cat. 1705062, Bio-Rad), and analyzed using an Odyssey XF Imaging System (LI-COR

Biosciences, Lincoln, NE). Quantitative analysis was carried out using NIH ImageJ system and normalized to β-actin. Original images for western blotting are in Supplementary Fig. 16.

## ChIP-PCR

ChIP assays were conducted on various mouse tissues utilizing the High-Sensitivity ChIP Kit (Cat. ab185913, Abcam). Approximately 20 mg of tissue was employed for ChIP analysis of the RUNX1 binding to lipid metabolism genes across different tissues. The tissues were dissected into small pieces (1–2 mm$^3$) using scissors. To initiate cross-linking, a solution was prepared by adding 37% formaldehyde to cell culture medium, resulting in a final concentration of 1% formaldehyde. Subsequently, 1 ml of the cross-linking solution was added to each 20 mg tissue sample, which was then incubated for 20 min on a rocking platform. Following this, 1.25 M glycine was added to the tissue tubes to achieve a final concentration of 0.125 M. The tubes were then mixed and centrifuged at $200 \times g$ for 5 min to pellet the tissues. The supernatant was carefully removed, and the tissues were transferred to a Dounce homogenizer. Next, 0.5 mL of working lysis buffer was added, and the tissues were homogenized for 40 strokes. The homogenized mixture was then transferred to a 2 ml tube and centrifuged at $1000 \times g$ for 5 min at 4 °C. After removing the supernatant, the chromatin pellet was resuspended in ChIP buffer and transferred to a 1.5 ml tube. The chromatin lysates were incubated on ice for 10 min, with occasional vertexing.

Chromatin shearing was achieved using water bath sonication with 20 cycles of shearing, each cycle consisting of 15 s of sonication followed by 30 s of rest, at 170–190 watts under cooling conditions. The resulting chromatin solution was either utilized immediately or stored at −80 °C for future use. PCR amplification was carried out using High Fidelity PCR Master Mix (Cat. 12140314001, Millipore-Sigma) with the ProFlex PCR system (Applied Biosystems). The PCR products were analyzed using 1.5% agarose gel and imaged using the Syngene™ NuGenius Gel system (Syngene™ NUGKIT). Primer sequences are provided in Supplementary Table 2, with β-actin serving as the control.

## Oxygen consumption rate (OCR) assay

For assessing adipose tissue mitochondrial OCR, the detailed method was modified as described previously[75]. Mice ~1 month old were used to perform the OCR assay. The mice were euthanized with isoflurane following the NIH/NIDDK AUCU guidelines. scWAT was then isolated and pooled from three mice. The tissue was immediately minced to a fine consistency and digested for 60 min at 37 °C with shaking ($20 \times g$) in 3 ml of Krebs-Ringer bicarbonate buffer (Cat. K4002, Millipore-Sigma) supplemented with 2% BSA (Cat. A7030, Millipore-Sigma) and 1600U type I collagenase (Cat. SCR103, Millipore-Sigma). The mixture was vortexed vigorously for a few seconds every 5–10 min to lyse the mature adipocytes. After digestion, 9 ml of Krebs-Ringer bicarbonate buffer with 2% BSA was added, and the mixture was centrifuged for 5 min at $350 \times g$. The supernatant was removed, and the cell pellet was gently resuspended in 4 ml of high-glucose Dulbecco's Modified Eagle Medium (DMEM) (Cat. 11965092, Thermo Fisher) supplemented with 10% fetal bovine serum (FBS) (Cat. A5670701, Thermo Fisher) and 100 U/ml penicillin/streptomycin (Cat. 15140122, Thermo Fisher). The suspension was passed through a 40 μm cell strainer (Cat. 352340, Life Sciences) into a 6 cm diameter cell culture dish (Cat. 430166, Corning). The following day, the media were replenished with fresh media and changed every 2nd day until the cells reached 90% confluence, which took ~4–5 days. Approximately 350,000 cells were used for distribution into 20 wells of an XF24 plate, leaving four wells per plate empty to correct for positional temperature variations. Mitochondrial activity was measured by the Seahorse XF Cell Mito Stress Test Kit (Agilent, Cat #103015-100) using Seahorse XFe24 Extracellular Flux Analyzer (Agilent) according to the manufacturer's instructions.

## Indirect calorimetry measurement of mice

Before each experimental study, mice were acclimated to the chambers or cages at 22 °C for a duration of 2 days. Analysis was conducted using data obtained on the 3rd day. The duration of measure lasted 400 min. Measurements of total energy expenditure, respiratory exchange ratio (RER), food intake (via floor feeder), and physical activity (measured using infrared beam breaks with a spacing of 0.5 inches) were carried out using a 12-chamber indirect calorimetry system (CLAMS with Oxymax software v5.52, Columbus Instruments, Columbus, OH). In this system, mice were housed individually with ad libitum access to food (provided via hanging feeder) and water in plastic cages containing ~95 g of wood chip bedding (7090 Teklad sani-chips bedding, Envigo, Indianapolis, IN). The testing cage parameter as follows: 2.5 L volume, flow rate 0.5 L/min, sampling flow 0.4 L/min, settle time 55 s, measure time 5 s. Ambient temperature was continuously monitored using a U12-012 data logger (Onset, Bourne, MA) placed in an empty calorimetry chamber. All 12 calorimetry chambers were located within a single temperature-controlled environmental chamber. The system was calibrated using a defined mixture of $O_2$ and $CO_2$. Physical activity was measured using infrared beam breaks with a spacing of 1 inch, and ambient temperature was continuously monitored in each cage. Food intake and physical activity were assessed at 13-min intervals. Resting metabolic rate was calculated with ambulation equal zero. The RER was calculated as the ratio between $VCO_2$ and $VO_2$.

## Real time-PCR

Total RNA was extracted using the RNeasy Lipid Tissue Mini Kit (Cat. 74804, Qiagen, Germantown, MD) and RNeasy Fibrous Tissue Mini Kit (Cat. 74704, Qiagen) according to manufacturer's instructions and reverse-transcribed with SuperScript™ IV First-Strand Synthesis System (Cat. 18091050, Thermo Fisher Scientific). The expression of specific genes in each tissue was detected by relative quantification using gene-specific primers (Supplementary Table 2) and PowerUp™ SYBR™ Green Master Mix for qPCR (Cat. A25742, Applied Biosystem) or HiScript III All-in-one RT SuperMix Perfect for qPCR (Cat. R333-01, Vazyme) for real-time RT-PCR with normalization to housekeeping genes GAPDH and 18S using the Delta-Delta CT method using a 7900 Sequence Detector (Applied Biosystems, Foster City, CA, USA). For absolute quantification of mouse EpoR mRNA in mouse tissues, probe-based Taqman PCR and mouse EpoR and S16 cDNA plasmids were used (16S, internal control)[7]. Primers and probes are listed in Supplementary Table 2.

## RUNX1-inhibitor treatment

Mice beginning at day P21 were treated with 100 μl of 75 μM Ro5-3335 RUNX1-inhibitor (Cat. HY-10847, MedChemExpress, Monmouth Junction, NJ). Ro5-3335 was initially diluted in DMSO to a final concentration of 10 mM, then further diluted in saline to achieve a final concentration of 75 μM, resulting in 0.75 μl of DMSO/100 μl of 75 μM Ro5-3335 solution. The drug or DMSO in saline was administered via subcutaneous injection three times per week for 2 weeks. Each experiment group consisted of five mice.

## EPO treatment

EPO treatment (3000 U/kg; Epoetin alpha, Amgen, Thousand Oaks, CA) in mice was administered by subcutaneous injection beginning at day P21 in WT-mice, ΔEpoR$_E$-mice, EpoR$^{Adiponectin-KO}$-mice and EpoR$^{aP2KO}$-mice for three times per week and continued for 3 weeks.

## Plasmids

To investigate the binding sites of RUNX1 in vitro, we constructed luciferase plasmids by cloning the potential RUNX1 binding regions identified in the promoters of lipid metabolism-related genes listed in Supplementary Table 3. These genes included the mouse *Lpl*, *Acc1*,

*Acc2*, *Lipin1*, *Fas*, *Srebf1*, *Scd1*, *Ppary*, as well as the human *ZBTB7B* gene (as a control for RUNX1 silencer activity), cloned into the pNL3.1[Nluc/minP] vector from Promega (Cat. N1031). The cloning was performed using XhoI and HindIII restriction enzymes. The sequences of the cloned plasmids were confirmed by Quintarabio company and were found to be 100% identical to the potential RUNX1 binding sequences identified in the promoter regions of the corresponding genes. These cloned plasmids are listed in Supplementary Table 3.

## Cells culture and transient plasmid transfection
HEK293T cells were purchased from ATCC company (Cat. CRL-3216™) and cultured in DMEM with GlutaMAX supplement (Cat. 1564011, Thermo Fisher Scientific), 10% FBS (Cat. A4766801, Thermo Fisher Scientific) and 100 U/ml penicillin–streptomycin (10,000 U/mL; Cat. 15140122, Thermo Fisher Scientific). Plasmid DNA for transfection was prepared and purified as previously described[76]. Plasmids purchased from Addgene, including pCMV-Myc FBXW7 WT (Cat. 16652), pLX304_zeo_mmFBXW7a (Cat. 160102), pcDNA3.1-Flag Runx1 FL (Cat. 14585), Banshee Runx1 FL mCherry (Cat. 80157), pCWXPGR-pTF-EpoR (Cat. 114289), and pCMV5-CBFbeta (Cat. 12427) were transfected into HEK293T cells using Lipofectamine™ 3000 Transfection Reagent (Cat. L3000015, Thermo Fisher Scientific) with Opti-MEM™ I Reduced Serum Medium (Cat. 31985070, Thermo Fisher Scientific). For a 10 cm dish transfection, each plasmid about 5 μg was mixed with Lipofectamine™ 3000 at a ratio of 1:2.5 in Opti-MEM medium and incubated at room temperature for 10–15 min before being added to HEK293T cells that had been subcultured for 16–20 h to a density of 70%–80%. After 48 h of transient plasmid transfection, cells were treated with EPO (3 U/ml) or PBS for 24 h. Cells were then harvested, and total proteins extracted with RIPA buffer including 1× protease inhibitor cocktail (Cat. 8340, Millipore-Sigma) and separated using western blotting.

## NanoBiT luciferase studies
The Nano-Glo® Live Cell Assay System (Cat. N2011, Promega, Madison, WI) is a single-addition, non-lytic detection reagent used to measure NanoBiT® or NanoLuc® luminescence from living cells[77]. We used the potential RUNX1 binding regions identified in the promoter regions of the lipogenic genes *Acc1/2*, *Fas*, *Lipin1*, *Srebf1*, and *Scd1* and the lipolytic genes *Ppary* and *Lpl*. RUNX1 binding site sequences (centered on the predicted RUNX1 binding site) were amplified by PCR (Supplementary Table 3) and separately subcloned into pNL3.1 Minimal Promoter-Driven NanoLuc® Genetic Reporter Vectors (Promega). To test if RUNX1 expression, EPO signaling or the combination of RUNX1 and EPO signaling could activate a luciferase reporter gene containing a RUNX1 binding motif from lipolytic or lipogenic genes, luciferase reporter gene activity was assessed in HEK293T cells. Expression plasmids for RUNX1 or EpoR or EpoR and RUNX1 were co-transfected with one of the reporter gene constructs containing the RUNX1 binding site from lipogenic genes *Acc1/2*, *Fas*, *Lipin1*, *Srebf1* or *Scd1*-luc, or the RUNX1 binding site from lipolytic genes *Ppary* and *Lpl*, or the *ZBTB7B*-luc control construct. Luciferase activity was determined following the manufacture's instruction for the Nano-Glo® Live Cell Assay System and using the GloMax Discover System microplate reader (Supplier No.:GM3000, Promega).

## ELISA assay
Leptin, adiponectin, insulin, IFN-γ, IL-1α, IL-1 β, IL-6, and TNF concentration in the serum samples were quantified by using the Mouse/Rat Leptin Quantikine ELISA Kit (Cat. MOB00B, R&D Systems), HMW & Total Adiponectin ELISA (Cat. 80-ADPHU-E01, ALPCO), Mouse Insulin Standard 2 ng (Cat. 90020, Crystal Chem Inc.), ELISA MAX™ Standard Set Mouse IFN-γ (Cat. 430801, Biolegend), ELISA MAX™ Standard Set Mouse IL-1α (Cat. 433404, Biolegend), ELISA MAX™ Standard Set Mouse IL-1β (Cat. 432601, Biolegend), Mouse IL-6 Quantikine ELISA Kit

(Cat. M6000B-1, R&D Systems), Mouse TNF-alpha Quantikine ELISA Kit (Cat. MTA00B, R&D Systems), Mouse Alanine Aminotransferase (ALT) ELISA Kit (Cat. MBS264717, MyBioSource) and Mouse Aspartate Aminotransferase (AST) ELISA Kit (Cat. MBS450720, MyBioSource) according to the manufacturer's instructions.

## Statistical analysis
All experiments were conducted at least three times independently, with the results from one of these experiments shown in the figures (Figs. 2a, 6h, and 7c). Values are expressed as mean + SEM or SD. Results between two groups of mice were compared by unpaired *t*-test; for more than two groups we used one or two-way ANOVA with Bonferroni multiple comparisons tests. *P* values of <0.05 were considered statistically significant.

## Reporting summary
Further information on research design is available in the Nature Portfolio Reporting Summary linked to this article.

## Data availability
The data are available within the article include Supplementary Information or Source data file. The original data were uploaded in the Figshare database. Figshare: https://doi.org/10.6084/m9.figshare.26042866. Source data are provided with this paper.

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

## Acknowledgements
The authors thank Oksana Gavrilova and the National Institute of Diabetes and Digestive and Kidney Diseases Mouse Metabolism Core. The authors also thank the National Institute of Diabetes and Digestive and Kidney Diseases Advanced Light Microscopy & Image Analysis Core (ALMIAC). This work was supported by the Intramural Research Program of the National Institute of Diabetes and Digestive and Kidney Diseases and by NIH Grant P01HL149626 (X.A.).

## Author contributions
W.Y. designed the experiments, conducted the studies, analyzed the data, and wrote the manuscript. P.R., H.R., X.A., and M.G. contributed to data generation and analysis and reviewed and edited the manuscript. T.Y. and J.K. contributed to oxygen consumption rate (OCR) data generation and analysis. C.T.N. contributed to the experimental design and discussion of data and reviewed and edited the manuscript.

## Funding

## Competing interests
The authors declare no competing interests.
