## [Peer Review File · Nature Communications]

Erythropoietin regulates energy metabolism through EPO-EpoR-RUNX1 AxisREVIEWER COMMENTS

Reviewer #1 (Remarks to the Author):

Previous work by the authors demonstrated that the deletion of EpoR in adipose and erythroid tissues induces metabolic disturbances, causing increased fat storage and reduced EE. The present study built upon this and determined that adipose tissue plays a significant and specialized role in the Epo-EpoR axis in regulating metabolism in a beneficial manner due to its high expression of the EPO-specific receptor EpoR. They found that when EPO is overexpressed or given at high concentrations in mice, this led to the reduction of body weight, fat loss and lipogenesis genes while increasing lipolytic genes not only in WAT, but also in BAT, skeletal muscle and liver. Further, it was determined that the Epo-EpoR axis regulates RUNX1 protein expression through the inhibition of the ubiquitin ligase FBXW7. Overall, these data suggest that promoting the Epo-EpoR axis in subcutaneous WAT may improve systemic lipid and glucose metabolism in young lean mice. Despite these promising data, some major points should be addressed prior to publication.

Major comments:

-The rationale for the study purpose could be stronger in the introduction, as well as outlining the big-picture relevance/clinical significance contribution to mitigating obesity in humans in the introduction and abstract. Currently, the introduction seems too long and unfocused.

-While significance is detected in most measurements, mouse numbers relatively low, which may cast some doubt on reproducibility. How many cohorts were run for each experiment?

-Experiments have been conducted in relatively young, lean mice under mild cold stress at room temperature. Why would it be beneficial to further decrease body weight, fat mass, the ability to synthesize lipids in nearly every metabolic organ, etc. in healthy lean mice? With a further loss in these parameters, does this then produce an unhealthy, catabolic phenotype in young mice? Along these lines, are systemic markers of inflammation, such as IL6, TNFa, AST/ALT normal in these mice? Overall, this study is lacking a translational aspect. It would be valuable to determine whether high EPO has the same effects in a diet-induced obesity and/or aged model, perhaps also under thermoneutral conditions to more closely relate to human obesity.

-Why would elevated EPO cause browning? Although this is speculated upon, is browning the main contributor to improved metabolism, or is it due to increased lipolysis and decreased lipogenesis?

Minor Comments:

-The addition of mitochondrial respiration data and/or metabolic caging studies would be interesting to assess the metabolic demand and mitochondrial capacity of mice given EPO or Tg6 mice in addition to gene expression and histology data, if available, as shown in previous papers by the group.

-Please calculate the GTT and ITT AUCs. Especially for the RUNX1 inhibitor study, it seems that there may be significance between the inhibitor and non inhibitor within the WT or Tg groups. Further, the symbols denoting significance (?) within Figure 5K are not indicated in the figure caption.

-Figure 4 (f-j) with experiments in adipose specific EpoR KO, may be better to compare to corresponding EpoR^{-/-} control.

-n sizes are not reported for some experiments. Please report individual data points for all data.

Reviewer #2 (Remarks to the Author):

Comments for 7721279 W Yin/CT Noguchi

The authors studied the role of Erythropoietin (EPO) in energy metabolism in adipose tissues. They found that EPO activation of Epo receptor (EpoR) modulates the expression of lipolytic and lipogenic gene expression in subcutaneous white adipose tissue (scWAT) through transcription factor RUNX1. RUNX1 is rapidly degraded by posttranslational ubiquitylation. EPO disrupts this activity by decreasing FBXW7 mRNA and Fbw7 protein in scWAT. The authors provided the evidence for enhancer activity of the RUNX1 binding motif in the promoter region of lipolytic genes and silencer activity of the RUNX1 binding motif in the promoter region of lipogenic genes,

Thus, the authors describe that FBXW7 would be an important regulator of energy and lipid metabolism.

RUNX1 stimulated activity in WAT by EPO can affect fat metabolism in other tissues and contribute to body weight and fat mass regulation independent of EPO stimulated erythropoietic response.

Comments:

The study was well-done and the results are convincing.

I have the following questions:

1) The core of the RUNX binding site is GxGG. it was not clear to me how they detected the core motif from the data shown in Figure 5 ? It would be helpful if the authors explain in more detail how RUNX binding site was extracted from the data in Figure 5b.

2) RUNX family proteins function together with the partner protein CBF². For RUNX1 to function as a DNA binding transcription factor, it must form a complex with CBF². Then the

question is whether RUNX1 can be ubiquitinated once RUNX1 is complexed with CBF². Please clarify the role of CBF² in the biological system that the authors are studying.

Reviewer #3 (Remarks to the Author):

This work by Noguchi's group builds on their long-standing work on the non-erythroid role of Epo on metabolic regulation. They have generated many genetic tools over the years and reported various findings on the role of Epo signaling in non-erythroid tissues including the brain and in metabolism. The current study is another extension of their work building on the tools that they have generated. They have shown previously that overexpression of Epo leads to lower body and adipose mass. In this paper, using the various overexpressing and selective knockout of EpoR that they have used before, show novel mechanisms of Epo effect on adipose tissue.

They begin by examining Tg6 mice that overexpress epo – with hematocrit of 80%. This is highly unphysiologic and it is not clear what relevance this mouse with such abnormal hematocrit would have on their metabolic outcome.

They show in Tg6 mice that they weigh less and show less adiposity. They don't discuss the expression pattern of EpoR overexpression using PDGF² promoter. Why was this promoter chosen and what is the significance. Furthermore, it is not shown starting at what age their weight separation occur? What happens in response to high fat diet? The lower weight occurring in response to normal diet suggest a catabolic response which is not favored. This is not at all how they interpret the results. Are they born with this abnormal weight? It seems at least in the ages shown that the curve is parallel which could mean that their growth is not changed but follow a parallel trajectory perhaps starting from birth? Also, in fig 1A, the liver looks almost black – perhaps necrotic likely related to exceedingly high hematocrit? This is what could be causing their catabolic state from their sickness or

cytokine response from necrotic tissue. Their low body weight from being sick would naturally be associated with improved glucose and insulin tolerance but this is very different from the metabolic benefit that the authors are referring to in this study. They state that 'These data suggest that EPO plays an important role in regulating fat mass and glucose metabolism' – line 104 – but this could all be because the mice are in a catabolic state due to some generalized sickness from tissue necrosis.

For X-Axis labelling in Fig 1A – what is meant by 'time'? Do they mean age?

For Fig 1f, ITTs are usually shown as percent decline from baseline – in which case the curve would look to not support their statement of improved insulin sensitivity – since they don't seem to decrease in response to insulin as stated by the authors. This also suggest perhaps inflammatory or insulin resistant state. What would be the circulating cytokine levels in these mice? Leptin, adiponectin and insulin, and cytokine levels would be helpful in interpreting their results.

They state that EpoR expression is highly expressed in adipose tissue using the reporter EpoR-tdTomato-Cre mice. This may very well represent the abundant adipose tissue resident macrophages analogous to erythroblastic island macrophages that support erythropoiesis as reported by other groups. Adipose tissue contains many cell types including adipocytes and many immune cell types including macrophages. Adipose tissue-KO using Ap2 mice is a very vague way to describe their finding. They need to specify whether they are referring to effects by adipocytes or other cells in the stromal vascular fraction that is responsible for their findings.

For Fig 2, scale bar is only shown for liver section and is likely markedly different for magnification for adipose tissue. Also, for gene expression is it normalized for cell number? This is not indicated and therefore not interpretable and does not support their statement

Use of Ap2-cre to disrupt EpoR in adipose tissue is problematic given the wide expression of AP2 in many cell types, in particular macrophages. Adiponectin-cre would be preferred if

they want to proposed that the metabolic effects are attributed to adipocytes. Referring to adipose tissue is too vague and non-descript.

Their entire work needs to be separated to delineate whether they are derived from stromal vascular fraction or adipocytes.

Point-by-point response to Reviewers' Comments

Reviewer #1 (Remarks to the Author):

Previous work by the authors demonstrated that the deletion of EpoR in adipose and erythroid tissues induces metabolic disturbances, causing increased fat storage and reduced EE. The present study built upon this and determined that adipose tissue plays a significant and specialized role in the Epo-EpoR axis in regulating metabolism in a beneficial manner due to its high expression of the EPO-specific receptor EpoR. They found that when EPO is overexpressed or given at high concentrations in mice, this led to the reduction of body weight, fat loss and lipogenesis genes while increasing lipolytic genes not only in WAT, but also in BAT, skeletal muscle and liver. Further, it was determined that the Epo-EpoR axis regulates RUNX1 protein expression through the inhibition of the ubiquitin ligase FBXW7. Overall, these data suggest that promoting the Epo-EpoR axis in subcutaneous WAT may improve systemic lipid and glucose metabolism in young lean mice. Despite these promising data, some major points should be addressed prior to publication.

The Reviewer's comments and opportunity to improve our manuscript are greatly appreciated.

Major comments:

-The rationale for the study purpose could be stronger in the introduction, as well as outlining the big-picture relevance/clinical significance contribution to mitigating obesity in humans in the introduction and abstract. Currently, the introduction seems too long and unfocused.

Introduction has been revised to provide a shorter, focused rationale for the study purpose.

-While significance is detected in most measurements, mouse numbers relatively low, which may cast some doubt on reproducibility. How many cohorts were run for each experiment?

We conducted a minimum of three repetitions for each experiment. For specific images, one representative outcome is presented in the manuscript. Additionally, we have included the corresponding 'n' numbers in the legends of each figure.

-Experiments have been conducted in relatively young, lean mice under mild cold stress at room temperature. Why would it be beneficial to further decrease body weight, fat mass, the ability to synthesize lipids in nearly every metabolic organ, etc. in healthy lean mice? With a further loss in these parameters, does this then produce an unhealthy, catabolic phenotype in young mice?

While EPO treatment in male mice can reduce body fat and weight gain, especially during high fat diet feeding, this is not observed in female mice that are not ovariectomized due to the protective effect of estrogen to diet-induced obesity^A. (Now included in the text.) Therefore, we have not seen and do not expect to see that EPO treatment in young healthy mice will produce unhealthy, catabolic phenotype. On the other hand, EPO treatment in sick lean mice can decrease inflammation, decrease disease manifestation, and improve locomotor activity as exemplified in a model for sepsis, without significant change in body weight^B.

A. Zhang, Y., Rogers, H. M., Zhang, X. & Noguchi, C. T. Sex difference in mouse metabolic response to erythropoietin. *FASEB J* **31**, 2661-2673, doi:10.1096/fj.201601223RRR (2017).

B. Vachharajani, V., Vital, S. & Russell, J. Modulation of circulating cell-endothelial cell interaction by erythropoietin in lean and obese mice with cecal ligation and puncture. *Pathophysiology* **17**, 9-18, doi:10.1016/j.pathophys.2009.04.002 (2010).

Along these lines, are systemic markers of inflammation, such as IL6, TNF α , AST/ALT normal in these mice? Overall, this study is lacking a translational aspect. It would be valuable to determine whether high EPO has the same effects in a diet-induced obesity and/or aged model, perhaps also under thermoneutral conditions to more closely relate to human obesity.

We thank the reviewer for the valuable question regarding our study. We have conducted tests on inflammation markers in both Tg6 mice and control mice. The mice with high expression of EPO exhibited reduced leptin, adiponectin, insulin, and serum glucose levels in their serum. However, there was no significant difference observed in the levels of inflammatory cytokines or liver function when compared to control mice from the same litter. These data are now included in the text as New Supplementary Figure s2. Please find further details outlined below:

New Supplementary Figure s2

Fig.s2

Supplementary Figure s2. Measurements of metabolic parameters, serum glucose, inflammatory cytokines and liver function in Tg6 and WT mice. a-k, ELISA assays were used to measure serum levels of leptin (a), adiponectin (b), insulin (c); inflammatory cytokines, IFN- γ (e), IL1- α (f), IL1- β (g), IL-6 (h), and TNF- α (i); liver function, AST (j) and ALT (k) in WT (grey) and Tg6 (red) mice. Glucometer was used to determine serum glucose (d). WT: n = 8, Tg6: n = 8. p-values are indicated.

In terms of translation, it is possible that extended administration of EPO could lead to a decrease in body weight and fat mass. We previously reported on EPO administration in wild type mice that various doses of EPO (1000, 600, 300, and 150 U/kg) resulted in significant reductions in both body weight gain and fat mass. Additionally, EPO doses of 300 U/kg and above demonstrated a significant impact on glucose tolerance. Notably, none of the EPO doses tested displayed any noticeable effects on food intake^A.

- A. Amanda Foskett, Mawadda Alnaeeli, Li Wang, Ruifeng Teng, Constance T Noguchi. The effects of erythropoietin dose titration during high-fat diet-induced obesity. *J Biomed Biotechnol.* 2011; 2011:373781.

We also carried out experiments using high fat diet (HFD) as a model for diet induced obesity. We found that in WT mice on HFD and treated with EPO for three weeks, body weight and fat mass were significantly decreased compared with HFD alone, with no change in lean mass. EPO treatment also significantly improved glucose tolerance. In Tg6 mice and WT littermate control mice on HFD for three weeks, Tg6 mice showed decrease body weight and fat mass with no significant difference in lean mass. Tg6 mice also showed significantly improved glucose tolerance. These data are now included in the text as New Supplementary Figure s3. Please find further details outlined below:

New Supplementary Figure s3

Fig.s3

Supplementary Figure s3. Body composition and glucose tolerance after high fat diet (HFD) feeding of WT mice treated with EPO or saline, and Tg6 mice. a-e, Tg6 mice (red) and WT littermate control mice (black) were fed HFD for three weeks and body weight (a), fat mass (b), and lean mass (c) were monitored, and glucose tolerance (GTT) (d) and

area under the curve (AUC of GTT) (e) were determined. p-values are indicated. f-j, WT mice were fed HFD for three weeks were treated with EPO (red), or phosphate buffered saline (PBS, black) and body weight (f), fat mass (g), and lean mass (h) were monitored, and glucose tolerance (GTT) (i) and area under the curve (AUC of GTT) (j) were determined. p-values are indicated.

-Why would elevated EPO cause browning? Although this is speculated upon, is browning the main contributor to improved metabolism, or is it due to increased lipolysis and decreased lipogenesis?

EPO regulation of increased lipolysis and decreased lipogenesis may be the main contributor to improved metabolism. In mice, inhibiting the lipogenic pathway initiated by adipocyte enzyme fatty acid synthase (FAS) increased thermogenesis and promoted browning subcutaneous WAT. Disruption of lipogenesis in WAT by knockout of FAS decreased adiposity, increased oxygen consumption, glucose tolerance and insulin sensitivity, and promoted the appearance of beige adipocytes including increased UCP1 in subcutaneous WAT^A. In mice, adipocyte knockout of FAS or acetyl-CoA carboxylase 1 (ACC1) required for lipid metabolism inhibits fatty acid synthesis which was recently shown to result in depletion of adipocyte palmitate that triggers WAT adipocyte browning response and increased UCP1 in vivo^B. This is now included in the text.

A. Lodhi, I. J. *et al.* Inhibiting adipose tissue lipogenesis reprograms thermogenesis and PPARgamma activation to decrease diet-induced obesity. *Cell Metab* **16**, 189-201, doi:10.1016/j.cmet.2012.06.013 (2012).

B. Guilherme, A. *et al.* Acetyl-CoA carboxylase 1 is a suppressor of the adipocyte thermogenic program. *Cell Rep* **42**, 112488, doi:10.1016/j.celrep.2023.112488 (2023).

Minor Comments:

-The addition of mitochondrial respiration data and/or metabolic caging studies would be interesting to assess the metabolic demand and mitochondrial capacity of mice given EPO or Tg6 mice in addition to gene expression and histology data, if available, as shown in previous papers by the group.

We have conducted an experiment to assess mitochondrial respiration data and metabolic caging in Tg6 male mice compared to their control counterparts. Our findings indicate that there is no discernible difference between the two groups in terms of mitochondrial respiration data or metabolic caging outcomes. However, it is worth noting that we did observe an increase in maximum mitochondrial respiratory capacity specifically in the Tg6 mice. These data are now included as the new Supplementary Figure s4. Please see the detailed results presented below:

New Supplementary Figure s4

Fig.s4

Supplementary Figure s4. Metabolic measurements by indirect calorimetry did not identify significant differences between WT and Tg6 male mice. a-c, Body weight (a), fat mass (b) and lean mass (c) were determined for Tg6 mice (red) and their littermate control (WT, black). d, Mouse activity with time expressed in intervals of 13 minutes for up to 4 days was recorded for Tg6 (red) and control (black) mice. e, Food intake was also recorded for Tg6 (red) and control (black) mice. f-h, TEE (f) Volume of O₂ (g), RER (h) were monitored for Tg6 (red) and control (black) mice with time expressed in intervals of 13 minutes. i-j, Volume of O₂, TEE, and RER of mice at 22°C (i) and food intake and mouse activity (j) at 22°C in Tg6 (red) and control (black) mice were recorded. a-h, WT, n = 7 mice; Tg6, n = 5 mice. i-j, WT, n = 6 mice; Tg6, n = 5 mice. p-values are indicated.

-Please calculate the GTT and ITT AUCs. Especially for the RUNX1 inhibitor study, it seems that there may be significance between the inhibitor and non inhibitor within the WT or Tg groups. Further, the symbols denoting significance (?) within Figure 5K are not indicated in the figure caption.

We thank the Reviewer for pointing out the omission in the Figure 5 legend. We have now included AUC for GTT in Figure 5k. The legend now also includes the following:

For panel k, \$ indicates comparison between WT+DMSO and Tg6+DMSO, † indicates comparison between WT+Ro5-3335 and Tg6+Ro5-3335, and & indicates comparison between Tg6+DMSO and Tg6+Ro5-3335. Data shown as mean ± SEM. Indicated are * (\$, †, &), $p < 0.05$; ** (\$\$, ††), $p < 0.01$; ***, $p < 0.005$; n.s. not significant.

New Figure 5

Fig.5

Figure 5. Regulation of lipid metabolism-associated gene expression in scWAT of Tg6 mice with high EPO compared with WT assessed by conserved motif identification. a,b, Corresponding promoter sequences (3000 bp) of lipid metabolism-associated genes, *Lpl*, and *Pparg* that exhibit elevated expression and *Acc1*, *Acc2*, *Fas*, *Lipin1*, *Srebf1*, and *Scd1* that exhibit lower expression in scWAT of Tg6 mice were searched for conserved motif by Clustal omega and MEME. Two conserved sequence motifs were identified and their relative locations in the respective promoter sequences are indicated in (a) by boxes: 1) 41 bp conserved RUNX binding motif (red) (b, upper panel); 2) 29 bp conserved Irf binding motif (green) (b, lower panel). Letter code is K (G/T), R (A/G), S (C/G), D (not C), and H (not G). The height of each amino acid code for each position reflects the weighted probability of occurrence at that position (b). c, RUNX1 protein expression was determined by Western blotting with GAPDH as control (upper) and quantified by ImageJ (lower) for scWAT from WT (grey) and Tg6 (red) mice. d, RUNX1 protein expression was determined by Western blotting with GAPDH as control (upper) and quantified by ImageJ (lower) for eWAT, BAT, skeletal muscle, and liver from WT (grey) and Tg6 (red) mice. e,f, RUNX1 protein level by Western blotting (e) and relative gene expression by qPCR (f) were determined for DEpoR_E mice treated with EPO (dark purple) or PBS (light purple) and WT mice treated with EPO (dark grey) or PBS (light grey) for 3 weeks beginning at postnatal 21 days. RUNX1 protein levels with GAPDH control were quantified by ImageJ (e; right). g, WT and Tg6 mice were treated with RUNX1 inhibitor Ro5-3335 (WT, dark grey; Tg6, red) or DMSO vehicle control (WT, light grey; Tg6, pink) and RUNX1 protein levels in scWAT were determined by Western blotting with GAPDH as control (upper) and quantified by ImageJ (lower). DMSO was used as vehicle and mice were injected with Ro5-3335 (5 mg/kg) subcutaneously every other day for six times. h, RUNX1 protein binding to the proposed RUNX1 binding region from the promoters of *Lpl*, *Acc1*, *Acc2*, *Fas*, *Lipin1*, *Srebf1*, *Pparg*, and *Scd1* in WT scWAT was determined by ChIP analysis. Shown are also Input DNA, ChIP-IgG, and RUNX1 binding to b-actin as controls. Original images for ChIP analysis are in Supplementary Fig.s11. i, Luciferase assay for transcription activity of the RUNX1 binding region from the promoters of *Pparg*, *Lpl*, *Acc1*, *Acc2*, *Fas*, *Lipin1*, *Srebf1*, and *Scd1* genes and ZBTB7B control (RUNX1 dependent silencer activity) inserted into a luciferase reporter plasmid and transfected into HEK293T cells (grey, control). Cells transfected with luciferase reporter plasmids were also co-transfected with RUNX1 expression plasmid without (orange; RUNX1) and with (blue; EPO+RUNX1) EPO treatment. In addition, the luciferase reporter plasmids were co-transfected with RUNX1 and EpoR expression plasmids without (purple; RUNX1+EpoR) and with (pink; EPO+RUNX1+Epo) EPO treatment. * indicates comparison with control. j, WT and Tg6 mice were treated with RUNX1 inhibitor Ro5-3335 (squares; WT, dark grey; Tg6, red) or DMSO vehicle control (circles; WT, light grey; Tg6, pink) as described in (e) and gene expression in scWAT for *Pparg*, *Lpl*, *Acc1*, *Acc2*, *Fas*, *Lipin1*, *Srebf1*, and *Scd1* determined by qPCR. k,l, Glucose tolerance test (GTT) (k) and area under the curve (AUC) for GTT (l) were performed on mice treated with RUNX1 inhibitor (Ro5-3335) or DMSO vehicle control as described in (e). Indicated are WT (dark grey) and Tg6 (red) mice treated with Ro5-3335 (dashed line, squares), and WT (light grey) and Tg6 (pink) mice treated with DMSO (solid line, circles). For panel k, \$ indicates comparison between WT+DMSO and Tg6+DMSO, † indicates comparison between WT+Ro5-3335 and Tg6+Ro5-3335, and & indicates comparison between Tg6+DMSO and Tg6+Ro5-3335. Data shown as mean ± SEM. Indicated are * (\$, †, &), p < 0.05; ** (\$\$, ††), p < 0.01; ***, p < 0.005; n.s. not significant.

-Figure 4 (f-j) with experiments in adipose specific EpoR KO, may be better to compare to corresponding EpoR^{-/-} control.

Whole-body ablation of EpoR results in embryonic lethality in mice. Therefore, we employed adipocyte-specific ablation of EpoR to investigate the crucial phenotype. In addition, we utilized Adiponectin-cre/EpoR^{fl/fl} mice to replicate the findings, and we arrived at the same conclusion as observed with Ap2-cre/EpoR^{fl/fl} mice. Data for body weight and fat mass and for lipid metabolism gene expression in ScWAT, skeletal muscle and liver for EpoR^{Adiponectin-KO} mice are now presented in new Figure 4f-j.

Data for lipid metabolism gene expression in eWAT and BAT for EpoR^{Adiponectin-KO} mice are now included as the new Supplementary Figure s6a-b. Data for body weight and fat mass and for lipid metabolism gene expression in ScWAT, skeletal muscle and liver for EpoR^{Ap2KO} mice are now presented in new Supplementary Figure s6c-j.

Please see the detailed results presented below:

New Figure 4

Fig.4

Figure 4. EPO regulation of lipid metabolism gene expression in WT mice is not evident in DEpoR_E mice or mice that lack EPOR in WAT. a,b, Body weight (a) and fat mass (b) were determined for WT mice treated with EPO (dark grey) or PBS (light grey) and DEpoR_E mice treated with EPO (purple) or PBS (light purple) for 3 weeks beginning at postnatal 21 days. **c-e,** WT mice treated with EPO (dark grey) or PBS (light grey) and DEpoR_E mice treated with EPO (purple) or PBS (light purple) for 3 weeks beginning at postnatal 21 days and gene expression determined for *Pparg*, *Lpl*, *Acc1*, *Acc2*, *Fas*, *Lipin1*, *Srebf1*, and *Scd1* by qPCR in the ScWAT (c), skeletal muscle (d), and liver (e) (n = 5 per group). **f,g,** Body weight (f) and fat mass (g) were determined for EpoR^{Adiponectin-KO} mice treated with EPO (blue) or PBS (grey) (n = 5 mice per group). **h-j,** EpoR^{Adiponectin-KO} mice were treated with EPO (blue) or PBS (grey) for three weeks beginning at postnatal 21 days and gene expression for *Pparg*, *Lpl*, *Acc1*, *Acc2*, *Fas*, *Lipin1*, *Srebf1*, and *Scd1* in ScWAT (h), skeletal muscle (i), and liver (j) determined by qPCR. Data are shown as mean ± SEM. p-values are indicated.

New Supplementary Figure s6

Fig.s6

Supplementary Figure s6. Ablation of EpoR in adipocytes diminished the EPO effect on lipid metabolism. a-b, $EpoR^{Adiponectin-KO}$ mice were generated and gene expression for lipid metabolism genes, *Pparg*, *Lpl*, *Acc1*, *Acc2*, *Fas*, *Lipin1*, *Srebf1*, and *Scd1* were quantified by qRT-PCR from eWAT (a) and BAT (b) from PBS (gray) or EPO treated (blue) $EpoR^{Adiponectin-KO}$ mice. **c-j,** $EpoR^{aP2-creKO}$ mice were generated by crossing aP2-Cre mice (Strain #:005069, C57BL/6 background) with $EpoR^{floxp/floxp}$ mice. Body weight (c), fat mass

(d), and lean mass (e) were monitored in PBS (gray) or EPO treated EpoR^{aP2KO} (green) mice. Expression of lipid metabolism genes, *Pparg*, *Lpl*, *Acc1*, *Acc2*, *Fas*, *Lipin1*, *Srebf1*, and *Scd1*, were quantified by qRT-PCR for ScWAT (f), eWAT (g), BAT (h), skeletal muscle (i), and liver (j) from PBS (gray) or EPO treated (green) EpoR^{aP2KO} mice. No significant differences were observed between PBS and EPO treatment in EpoR^{Adiponectin-KO} or EpoR^{aP2KO} mice. p-values are indicated.

-n sizes are not reported for some experiments. Please report individual data points for all data.

"n" is now indicated for all experiments and figures modified to report individual data points.

Reviewer #2 (Remarks to the Author):

Comments for 7721279 W Yin/CT Noguchi

The authors studied the role of Erythropoietin (EPO) in energy metabolism in adipose tissues. They found that EPO activation of Epo receptor (EpoR) modulates the expression of lipolytic and lipogenic gene expression in subcutaneous white adipose tissue (ScWAT) through transcription factor RUNX1. RUNX1 is rapidly degraded by posttranslational ubiquitylation. EPO disrupts this activity by decreasing FBXW7 mRNA and Fbw7 protein in ScWAT. The authors provided the evidence for enhancer activity of the RUNX1 binding motif in the promoter region of lipolytic genes and silencer activity of the RUNX1 binding motif in the promoter region of lipogenic genes, Thus, the authors describe that FBXW7 would be an important regulator of energy and lipid metabolism. RUNX1 stimulated activity in WAT by EPO can affect fat metabolism in other tissues and contribute to body weight and fat mass regulation independent of EPO stimulated erythropoietic response.

Comments:

The study was well-done and the results are convincing.

I have the following questions:

1) The core of the RUNX binding site is GxGG. it was not clear to me how they detected the core motif from the data shown in Figure 5 ? It would be helpful if the authors explain in more detail how RUNX binding site was extracted from the data in Figure 5b.

We thank the reviewer for the encouraging comments. We successfully identified key genes that exhibited upregulation or downregulation in response to EPO in Tg6 mice. Subsequently, we retrieved the complete promoter regions of these genes, encompassing approximately 3000 base pairs upstream of the start codon. Utilizing the MEME suite online software (<https://meme-suite.org/meme/>), we conducted predictive analyses to determine the protein binding patterns within these gene promoters. The MEME software enabled us to pinpoint significant motifs representing binding sites for distinct transcription factors. Notably, our

investigation highlighted the presence of RUNX1 as a prominent candidate, prompting us to delve further into its role and implications.

2) RUNX family proteins function together with the partner protein CBF β . For RUNX1 to function as a DNA binding transcription factor, it must form a complex with CBF β . Then the question is whether RUNX1 can be ubiquitinated once RUNX1 is complexed with CBF β . Please clarify the role of CBF β in the biological system that the authors are studying.

We thank the reviewer for raising the question about the influence of CBF β on RUNX1 stability. We now provide evidence that RUNX1 may contribute to CBF β stability.

RUNX1 binds to DNA and partners with CBF β to enhance DNA binding and regulate gene expression^A. In transient expression in P19 cells, CBF β protects RUNX1 from ubiquitination and extends the RUNX1 half-life, and RUNX1 protein is not detectable in embryonic extracts from CBF β knockout mice while RUNX1 transcripts remain unchanged^B. CBF β is also expressed in fat tissue, consistent with our observation of RUNX1 expression in white adipose tissue especially with EPO treatment, and CBF β is required for adipocyte development and adipogenesis^C. Although whole body knockout of CBF β is embryonic lethal, fat specific knockout of CBF β generates mice with similar body weight as WT mice with low body adiposity and adipokines, and a progressive loss of body fat with hyperglycemia and hyperinsulinemia and worsening of lipodystrophy with increasing age^C.

To demonstrate that RUNX1 can be ubiquitinated once RUNX1 is complexed with CBF β , combinations of RUNX1, CBF β , FBW7 and EpoR expression plasmids were transfected into HEK-293 cells and treated with EPO or PBS, analogous to the transfection studies presented in Figure 6c. We assessed K48-polyubiquitin (K48-Ub) and K63-Ub (proteasome-independent ubiquitylation), and plasmid derived expression of Flag-RUNX1, CBF β , FBW7-Myc, and EpoR. K48-Ub was detected with transfection of the FBW7 expression plasmid alone, with co-transfection of FBW7 and RUNX1, and with co-transfection of FBW7, RUNX1, CBF β and EpoR, but K48-Ub was markedly diminished with activation of the EPO-EpoR axis (co-transfection of EpoR expression plasmid combined with EPO treatment), indicating EPO-EpoR activation decreased FBW7 ubiquitin ligase activity. K63-polyubiquitin was not markedly affected with or without transfection of expression plasmids for FBW7, RUNX1 or EpoR. Assessing RUNX1 stability, we found that co-transfection of expression plasmids for RUNX1 and FBW7 markedly decreased RUNX1 protein level with and without co-transfection with the CBF β expression plasmid suggesting that even with the increased potential for RUNX1 to complex with CBF β , increased FBW7 E3 ubiquitin ligase expression decreased RUNX1 stability and increased RUNX1 degradation. Activation of the EPO-EpoR axis by co-transfection of EpoR combined with EPO treatment abrogated the decrease in RUNX1 protein level by increased FBW7 protein. Interestingly, CBF β protein was detected with increased RUNX1 expression both with and without co-transfection of the CBF β expression plasmid, and appeared to be reduced with decreased RUNX1 stability. While CBF β has been reported to stabilize RUNX1^B, these data suggest that, conversely, RUNX1 contributes to CBF β stability. These data are now included as the new Figure 6c. The original Figure 6c that shows co-transfection combination of RUNX1, FBW7 and EpoR treated with EPO and with PBS (without CBF β expression plasmid) is now included as new Supplementary Figure s10. Please see the detailed results presented below:

New Figure 6c

Figure 6. EPO activation of EPOR signaling in scWAT inhibits ubiquitin ligase FBW7, thereby increasing RUNX1 protein stability.

c, HEK293T cells were transfected with expression plasmids for FBW7 (FBW7-Myc), and/or RUNX1 (Flag-RUNX1), and/or CBF β , and/or EpoR and were treated with EPO (5 U/ml) or PBS for 24 h. The ubiquitylation of K48-Ub and K63-Ub, and protein levels of Flag-RUNX1, FBW7-Myc, CBF β , and EpoR were determined by Western blotting with β -actin as control.

New Supplementary Figure s10

Supplementary Figure s10. Assessment of relative protein stability of RUNX1 by activation of the EPO-EpoR axis without over-expression of CBF β . HEK293T cells were transfected with expression plasmids for FBW7 (FBW7-Myc), and/or RUNX1 (Flag-RUNX1), and/or EpoR and were treated with EPO (5 U/ml) or PBS for 24 h. The ubiquitylation of K48-Ub and K63-Ub, and protein levels of Flag-RUNX1, FBW7-Myc, and EpoR were determined by Western blotting with β -actin as control.

A. Tang, Y. Y. *et al.* Energetic and functional contribution of residues in the core binding factor beta (CBFbeta) subunit to heterodimerization with CBFalpha. *J Biol Chem* **275**, 39579-39588, doi:10.1074/jbc.M007350200 (2000).

B. Huang, G. *et al.* Dimerization with PEBP2beta protects RUNX1/AML1 from ubiquitin-proteasome-mediated degradation. *EMBO J* **20**, 723-733, doi:10.1093/emboj/20.4.723 (2001).

C. Lu, Y. H., Dallner, O. S., Birsoy, K., Fayzikhodjaeva, G. & Friedman, J. M. Nuclear Factor-Y is an adipogenic factor that regulates leptin gene expression. *Mol Metab* **4**, 392-405, doi:10.1016/j.molmet.2015.02.002 (2015).

Reviewer #3 (Remarks to the Author):

This work by Noguchi's group builds on their long-standing work on the non-erythroid role of Epo on metabolic regulation. They have generated many genetic tools over the years and reported various findings on the role of Epo signaling in non-erythroid tissues including the brain and in metabolism. The current study is another extension of their work building on the tools that they have generated. They have shown previously that overexpression of Epo leads to lower body and adipose mass. In this paper, using the various overexpressing and selective knockout of EpoR that they have used before, show novel mechanisms of Epo effect on adipose tissue.

They begin by examining Tg6 mice that overexpress epo – with hematocrit of 80%. This is highly unphysiologic and it is not clear what relevance this mouse with such abnormal hematocrit would have on their metabolic outcome.

We thank the reviewer for the insightful remarks and opportunity to further clarify the role of EPO on metabolic regulation. We use Tg6 mice as a model for chronic high EPO. We found hematocrit in Tg6 mice increases with age beginning after about two weeks of age, becoming at 0.52 versus 0.42 (WT) at postnatal day 21, and then 80% at about 35 days. The Tg6 mice used in the current study were from postnatal 21 days to 100 days, prior to notable organ degeneration. We found that Tg6 mice at postnatal 100 days exhibited no inflammation or liver necrosis. Tg6 mice at 5 months show no signs of degeneration in liver, kidney, skeletal muscle and sciatic nerve^A, suggesting that Tg6 mice may be useful to assess response associated with chronic EPO treatment. In contrast to young Tg6 mice, older Tg6 mice after 7 months exhibit multiple organ degeneration including some liver inflammation, and about half the mice exhibit hindlimb paralysis and myofiber degeneration in hindlimb skeletal muscle^A. Notably, young Tg6 mice displayed lower body weight starting approximately two weeks after birth. Moreover, hematocrit levels were assessed in Tg6 mice and their littermate controls, revealing elevated hematocrit levels in Tg6 mice around the age of two weeks. Additionally, mitochondrial Oxygen Consumption Rate (OCR) was measured in adipose tissue from Tg6 and WT mice. Adipose tissue from Tg6 mice showed a higher respiration rate compared to adipose tissue from control mice. These findings contribute to a deeper understanding of the physiological dynamics and potential implications of the Tg6 genotype and are now included as Supplementary Figure s1. Further research is warranted to elucidate the underlying mechanisms driving these observed effects.

A. Heinicke, K. *et al.* Excessive erythrocytosis in adult mice overexpressing erythropoietin leads to hepatic, renal, neuronal, and muscular degeneration. *Am J Physiol Regul Integr Comp Physiol* **291**, R947-956, doi:10.1152/ajpregu.00152.2006 (2006).

New Supplementary Figure s1

Fig.s1

Supplementary Figure s1. Body weight, hematocrit and oxygen consumption of adipose tissue from young male Tg6 mice. a, Body weight of WT control (black) and Tg6 (red) mice was recorded from day 1 after birth to 40 days old (n = 7). **b,** Hematocrit (percent) of WT control (black) and Tg6 (red) mice was measured from day 0 after birth to 90 days old (n = 7). **c,** Oxygen consumption rate was determined for adipose tissue from WT control (black) and Tg6 (red) mice. p-values are indicated.

Previously, Tg6 mice with transgenic human EPO overexpression provided insight on potential EPO activity, especially in non-hematopoietic tissue such as protection against light induced retinal degeneration^B, cardiac-protection in model of permanent coronary artery ligation^C, sex dependent EPO response on hypoxic ventilation^D, skeletal muscle repair^E, and metabolic regulation of blood glucose and body mass^F, as well as an adverse effect on bone health^{G,H}.

B. Grimm, C. *et al.* HIF-1-induced erythropoietin in the hypoxic retina protects against light-induced retinal degeneration. *Nat Med* **8**, 718-724, doi:10.1038/nm723 (2002).

C. Camici, G. G. *et al.* Constitutively overexpressed erythropoietin reduces infarct size in a mouse model of permanent coronary artery ligation. *Methods Enzymol* **435**, 147-155, doi:10.1016/S0076-6879(07)35008-8 (2007).

D. Soliz, J., Thomsen, J. J., Soulage, C., Lundby, C. & Gassmann, M. Sex-dependent regulation of hypoxic ventilation in mice and humans is mediated by erythropoietin. *Am J Physiol Regul Integr Comp Physiol* **296**, R1837-1846, doi:10.1152/ajpregu.90967.2008 (2009).

E. Jia, Y., Suzuki, N., Yamamoto, M., Gassmann, M. & Noguchi, C. T. Endogenous erythropoietin signaling facilitates skeletal muscle repair and recovery following pharmacologically induced damage. *FASEB J* **26**, 2847-2858, doi:10.1096/fj.11-196618 (2012).

F. Katz, O. *et al.* Erythropoietin treatment leads to reduced blood glucose levels and body mass: insights from murine models. *J Endocrinol* **205**, 87-95, doi:10.1677/JOE-09-0425 (2010).

G. Hiram-Bab, S. *et al.* Erythropoietin directly stimulates osteoclast precursors and induces bone loss. *FASEB J* **29**, 1890-1900, doi:10.1096/fj.14-259085 (2015).

H. Suresh, S., de Castro, L. F., Dey, S., Robey, P. G. & Noguchi, C. T. Erythropoietin modulates bone marrow stromal cell differentiation. *Bone Res* **7**, 21, doi:10.1038/s41413-019-0060-0 (2019).

They show in Tg6 mice that they weigh less and show less adiposity. They don't discuss the expression pattern of EpoR overexpression using PDGF β promoter. Why was this promoter chosen and what is the significance.

Tg6 line of mice was generated using a transgene that was intended to provide EPO overexpression in brain that consisted of human EPO cDNA driven by the human platelet-derived growth factor (PDGF) β -chain promoter to preferentially direct expression to neuronal cells^A. In addition to brain expression, only one transgenic line (Tg6) showed hypoxia independent increased plasma EPO levels to about 10-fold with hematocrit about 80% in heterozygous mice at 3-6 months of age. Tg6 mice provide a model system for chronic erythrocytosis and elevated transgenic human EPO expression in brain and lung. Elevated endothelial nitric oxide production that maintains normotension with no acute cardiovascular complications. However, older Tg6 mice exhibit decreased swim performance by 5 months, multi-organ degeneration (hepatic, renal, neuronal, and muscular) at 7 months, and reduced mean survival of 7.4 months^{B,C}. Therefore, young Tg6 mice were used in the current study prior to 4 months of age. We found that Tg6 male mice display discernible phenotypic characteristics around 14 days post-birth.

A. Ruschitzka, F. T. *et al.* Nitric oxide prevents cardiovascular disease and determines survival in polyglobulic mice overexpressing erythropoietin. *Proc Natl Acad Sci U S A* **97**, 11609-11613, doi:10.1073/pnas.97.21.11609 (2000).

B. Wagner, K. F. *et al.* Chronic inborn erythrocytosis leads to cardiac dysfunction and premature death in mice overexpressing erythropoietin. *Blood* **97**, 536-542, doi:10.1182/blood.v97.2.536 (2001).

C. Heinicke, K. *et al.* Excessive erythrocytosis in adult mice overexpressing erythropoietin leads to hepatic, renal, neuronal, and muscular degeneration. *Am J Physiol Regul Integr Comp Physiol* **291**, R947-956, doi:10.1152/ajpregu.00152.2006 (2006).

Furthermore, it is not shown starting at what age their weight separation occur?

Panel (a) above shows body weight of WT and Tg6 mice from birth to 40 days with weight separation occurring at around 14 days.

What happens in response to high fat diet?

We carried out experiments using high fat diet (HFD) as a model for diet induced obesity. We found that in WT mice on HFD and treated with EPO for three weeks, body weight and fat mass were significantly decreased compared with HFD alone, with no change in lean mass. EPO

treatment also significantly improved glucose tolerance. In Tg6 mice and WT littermate control mice on HFD for three weeks, Tg6 mice showed decrease body weight and fat mass with no significant difference in lean mass. Tg6 mice also showed significantly improved glucose tolerance. These data are now included in the text as New Supplementary Figure s3. Please find further details outlined below:

New Supplementary Figure s3

Fig.s3

Supplementary Figure s3. Body composition and glucose tolerance after high fat diet (HFD) feeding of WT mice treated with EPO or saline, and Tg6 mice. a-e, Tg6 mice (red) and WT littermate control mice (black) were fed HFD for three weeks and body weight (a), fat mass (b), and lean mass (c) were monitored, and glucose tolerance (GTT) (d) and area under the curve (AUC of GTT) (e) were determined. p-values are indicated. **f-j,** WT

mice were fed HFD for three weeks were treated with EPO (red), or phosphate buffered saline (PBS, black) and body weight (f), fat mass (g), and lean mass (h) were monitored, and glucose tolerance (GTT) (i) and area under the curve (AUC of GTT) (j) were determined. p-values are indicated.

The lower weight occurring in response to normal diet suggest a catabolic response which is not favored. This is not at all how they interpret the results. Are they born with this abnormal weight? It seems at least in the ages shown that the curve is parallel which could mean that their growth is not changed but follow a parallel trajectory perhaps starting from birth?

Initially, as shown above, Tg6 mice were born with comparable weights to their respective littermate controls. However, a distinct phenotype emerged around the two-week mark after birth. Tg6 mice displayed notable increased hematocrit level and decreasing body weight compared to control mice.

These observations could potentially be attributed to the influence of elevated EPO levels becoming significant evident after the two-week post-birth threshold. Alternatively, it is plausible that the PDGF β promoter becomes more active after postnatal two weeks, contributing to the observed effects. Further investigation is warranted to elucidate the underlying mechanisms driving these temporal changes.

Also, in fig 1A, the liver looks almost black – perhaps necrotic likely related to exceedingly high hematocrit? This is what could be causing their catabolic state from their sickness or cytokine response from necrotic tissue.

This is a key question. EPO treatment in sick lean mice can decrease inflammation, decrease disease manifestation and improve locomotor activity as exemplified in a model for sepsis, without significant change in body weight^A. The liver tissue in Tg6 mice exhibits a distinctive dark appearance, attributed to a higher concentration of blood red cells when compared to WT control mice. We utilized the H&E staining technique to examine both Tg6 mice and their littermate control counterparts, revealing the presence of intact tissue without signs of necrosis in Tg6 mice. This finding suggests that the overexpression of EPO in mice, leading to elevated hematocrit levels, does not induce tissue necrosis.

Figure A

Fig. A. Non-necrotic cells in Tg6 mice. a, H&E staining of liver sections from control and Tg6 mice. b, Oil red O staining (hematoxylin counterstain) of WT and Tg6 mice liver sections shows no fat accumulation. c, quantification of blood vessels (left) and necrotic cells (right) in the liver sections from WT and Tg6 mice. (n = 8 per group).

Note that EPO treatment in sick lean mice can decrease inflammation, decrease disease manifestation and improve locomotor activity as exemplified in a model for sepsis, without significant change in body weight^A.

A. Vachharajani, V., Vital, S. & Russell, J. Modulation of circulating cell-endothelial cell interaction by erythropoietin in lean and obese mice with cecal ligation and puncture. *Pathophysiology* 17, 9-18, doi:10.1016/j.pathophys.2009.04.002 (2010).

Their low body weight from being sick would naturally be associated with improved glucose and insulin tolerance but this is very different from the metabolic benefit that the authors are referring to in this study. They state that 'These data suggest that EPO plays an important role in regulating fat mass and glucose metabolism' – line 104 – but this could all be because the mice are in a catabolic state due to some generalized sickness from tissue necrosis.

We appreciate the Reviewer's insightful question. We conducted HE staining on liver tissues, which revealed the absence of necrotic tissues (shown in the above, Figure A). Instead, we

observed a notable presence of red cells and blood vessels. As a result, it is evident that the observed catabolic changes are not a consequence of whole-body necrosis. This clarification provides valuable insight into the mechanisms driving the observed effects.

For X-Axis labelling in Fig 1A – what is meant by ‘time’? Do they mean age?

We thank the Reviewer for calling attention to the labelling in figure 1. The figure has been corrected.

For Fig 1f, ITTs are usually shown as percent decline from baseline – in which case the curve would look to not support their statement of improved insulin sensitivity – since they don’t seem to decrease in response to insulin as stated by the authors. This also suggest perhaps inflammatory or insulin resistant state. What would be the circulating cytokine levels in these mice? Leptin, adiponectin and insulin, and cytokine levels would be helpful in interpreting their results.

We thank the reviewer for the valuable question regarding our study. We have conducted tests on inflammation markers in both Tg6 mice and control mice. The mice with high expression of EPO exhibited reduced leptin, adiponectin, insulin and serum glucose levels in their serum. However, there was no significant difference observed in the levels of inflammatory cytokines when compared to control mice from the same litter. These data are now included in the text as New Supplementary Figure s2. Please find further details outlined below:

New Supplementary Figure s2

Fig.s2

Supplementary Figure s2. Measurements of metabolic parameters, serum glucose, inflammatory cytokines and liver function in Tg6 and WT mice. a-k, ELISA assays were used to measure serum levels of leptin (a), adiponectin (b), insulin (c); inflammatory cytokines, IFN- γ (e), IL1- α (f), IL1- β (g), IL-6 (h), and TNF- α (i); liver function, AST (j) and ALT (k) in WT (grey) and Tg6 (red) mice. Glucometer was used to determine serum glucose (d). WT: n = 8, Tg6: n = 8. p-values are indicated.

They state that EpoR expression is highly expressed in adipose tissue using the reporter EpoR-tdTomato-Cre mice. This may very well represent the abundant adipose tissue resident macrophages analogous to erythroblastic island macrophages that support erythropoiesis as reported by other groups. Adipose tissue contains many cell types including adipocytes and many immune cell types including macrophages. Adipose tissue-KO using Ap2 mice is a very vague way to describe their finding. They need to specify whether they are referring to effects by adipocytes or other cells in the stromal vascular fraction that is responsible for their findings.

We thank the Reviewer for this comment and the opportunity to explicitly show EpoR expression on adipocytes. We include here EpoR staining on cells from 3T3-L1 adipocyte cell line revealing a distinct expression of EpoR on the cell membrane. Notably, the use of BODIPY staining confirmed the adipocytic nature of the cells. This crucial outcome reinforces the fact that EpoR expression is localized to the adipocyte cells themselves (shown here) in addition to the stromal vascular components that we previously reported^A. These findings contribute significantly to the understanding of EpoR distribution within the cellular context.

Alnaeeli, M. *et al.* Erythropoietin signaling: a novel regulator of white adipose tissue inflammation during diet-induced obesity. *Diabetes* **63**, 2415-2431, doi:10.2337/db13-0883 (2014).

Figure B

Fig. B. Immunofluorescence stained of 3T3L1 cell that showed the EpoR expressed on the cell membrane. Bodipy was used to stain the lipid droplets. Scale bar: 20 μ m.

For Fig 2, scale bar is only shown for liver section and is likely markedly different for magnification for adipose tissue. Also, for gene expression is it normalized for cell number? This is not indicated and therefore not interpretable and does not support their statement

We have completed a comprehensive revision of all figures, incorporating new scale bars for enhanced clarity. Note that gene expression values were normalized to expression of alpha-tubulin control. This normalization approach underscores the variations in EpoR expression across diverse organs within the mice.

Use of Ap2-cre to disrupt EpoR in adipose tissue is problematic given the wide expression of AP2 in many cell types, in particular macrophages. Adiponectin-cre would be preferred if they want to proposed that the metabolic effects are attributed to adipocytes. Referring to adipose tissue is too vague and non-descript.

We appreciate this important point raised by the Reviewer. To further validate and expand upon our findings using the Ap2-Cre/EpoR^{fl/fl} mice, we conducted parallel experiments using

Adiponectin-Cre/EpoR^{f/f} mice. The findings are in agreement with observations obtained using the Ap2-Cre/EpoR^{f/f} mice. It enhances the robustness of our conclusions by corroborating the observed outcomes across different genetic models. Data for body weight and fat mass and for lipid metabolism gene expression in ScWAT, skeletal muscle and liver for EpoR^{Adiponectin-KO} mice are now presented in new Figure 4f-j. Data for lipid metabolism gene expression in eWAT and BAT for EpoR^{Adiponectin-KO} mice are now included as the new Supplementary Figure s6a-b. Data for body weight and fat mass and for lipid metabolism gene expression in ScWAT, skeletal muscle and liver for EpoR^{ap2KO} mice are now presented in new Supplementary Figure s6c-g.

Please see the detailed results presented below:

New Figure 4

Figure 4. EPO regulation of lipid metabolism gene expression in WT mice is not evident in DEpoR_E mice or mice that lack EPOR in WAT. a,b, Body weight (a) and fat mass (b) were determined for WT mice treated with EPO (dark grey) or PBS (light grey) and DEpoR_E mice treated with EPO (purple) or PBS (light purple) for 3 weeks beginning at postnatal 21 days. **c-e,** WT mice treated with EPO (dark grey) or PBS (light grey) and DEpoR_E mice treated with EPO (purple) or PBS (light purple) for 3 weeks beginning at postnatal 21 days and gene expression determined for *Pparg*, *Lpl*, *Acc1*, *Acc2*, *Fas*, *Lipin1*, *Srebf1*, and *Scd1* by qPCR in the ScWAT (c), skeletal muscle (d), and liver (e) (n = 5 per group). **f,g,** Body weight (f) and fat mass (g) were determined for EpoR^{Adiponectin-KO} mice treated with EPO (blue) or PBS (grey) (n = 5 mice per group). **h-j,** EpoR^{Adiponectin-KO} mice were treated with EPO (blue) or PBS (grey) for three weeks beginning at postnatal 21 days and gene expression for *Pparg*, *Lpl*, *Acc1*, *Acc2*, *Fas*, *Lipin1*, *Srebf1*, and *Scd1* in ScWAT (h), skeletal muscle (i), and liver (j) determined by qPCR. Data are shown as mean ± SEM. p-values are indicated.

New Supplementary Figure s6

Fig.s6

Supplementary Figure s6. Ablation of EpoR in adipocytes diminished the EPO effect on lipid metabolism. a-b, *EpoR^{Adiponectin-KO}* mice were generated and gene expression for lipid metabolism genes, *Pparg*, *Lpl*, *Acc1*, *Acc2*, *Fas*, *Lipin1*, *Srebf1*, and *Scd1* were quantified by qRT-PCR from eWAT (a) and BAT (b) from PBS (gray) or EPO treated (blue)

EpoR^{Adiponectin-KO} mice. **c-j**, EpoR^{aP2-creKO} mice were generated by crossing aP2-Cre mice (Strain #:005069, C57BL/6 background) with EpoR^{flloxP/flloxP} mice. Body weight (**c**), fat mass (**d**), and lean mass (**e**) were monitored in PBS (gray) or EPO treated EpoR^{aP2KO} (green) mice. Expression of lipid metabolism genes, *Pparg*, *Lpl*, *Acc1*, *Acc2*, *Fas*, *Lipin1*, *Srebf1*, and *Scd1*, were quantified by qRT-PCR for ScWAT (**f**), eWAT (**g**), BAT (**h**), skeletal muscle (**i**), and liver (**j**) from PBS (gray) or EPO treated (green) EpoR^{aP2KO} mice. No significant differences were observed between PBS and EPO treatment in EpoR^{Adiponectin-KO} or EpoR^{aP2KO} mice. p-values are indicated.

Their entire work needs to be separated to delineate whether they are derived from stromal vascular fraction or adipocytes.

We thank the Reviewer for the opportunity to clarify this important point. We show above staining of the 3T3-L1 adipocyte cell, revealing distinct EpoR expression on the cell membrane. Additionally, the BODIPY staining provides clear evidence of the adipocytic nature of the cells. This outcome reaffirms that EpoR expression is localized to the adipocyte cells themselves, supporting the notion that EpoR is not only present in the stromal vascular components. This finding significantly advances our understanding of EpoR distribution within the cellular context.

REVIEWER COMMENTS

Reviewer #1 (Remarks to the Author):

Thank you for your detailed response.

Reviewer #2 (Remarks to the Author):

My comments to the original version were primarily on RUNX1 and its partner protein CFBF.

The authors description in the revised manuscript are mostly satisfactory.

The only point that I would like to mention is the following.

365 Although CFBF has been reported to stabilize RUNX160, these data suggest
366 that, conversely, RUNX1 contributes to CFBF stability.

Do the authors suggest that stabilization of RUNX1 by CFBF is wrong and stabilization of CFBF by RUNX1 is actually taking place ?

If so, the authors must show definitive evidence for such mechanism.

This experiments were done by using HEK293T cells transfected with various expression plasmids.

HEK293T cells may be expressing endogenous CFBF without transfecting the expression plasmid.

Reviewer #3 (Remarks to the Author):

This resubmission of a revised manuscript contains much added experiments and substantial revision addressing many of the comments. However, some concerns continue to persist as follows:

The insulin tolerance test shown in Fig 1g does not show increased insulin sensitivity as stated by the authors. There is hardly any glucose lowering effect of insulin during ITT and the reviewer requested showing percentage lowering of glucose (if not in conjunct with absolute glucose levels) which was not done. Also, typically with improved insulin sensitivity, there is an increase in adiponectin levels. The lower adiponectin levels in Tg6 mice is also out of keeping with the improved insulin sensitivity that the authors suggest.

The Tg6 mice used as a basis for all of the metabolic implications of Epo on adipocytes needs to be more rigorously vetted. It is stated in the rebuttal that the original intent of the generation of these mice was to investigate the role of Epo action in the brain. However, the authors did not show reporter expression of the Tg6 mice in the brain, but only showed the predominant expression in the adipose tissue. Other tissues that would be important to include, such as the brain, in particular, which can affect energy homeostasis, as well as the cells in the bone marrow or the spleen, which may affect the interpretation of their metabolic findings in the Tg6 mice would be good to include. Furthermore, to be using mice that have significantly shortened lifespan with serious health issues later in life, to use a narrow window that suitably show phenotypes to focus for this report on metabolism is in some ways highly problematic. Also, this information was not included in the manuscript, only in the rebuttal.

A balanced literature review including absence of metabolic phenotype in absence of EpoR in adipocytes (PMID: 23885016) would be good to include.

REVIEWER COMMENTS

Reviewer #2 (Remarks to the Author):

My comments to the original version were primarily on RUNX1 and its partner protein CBFβ. The authors description in the revised manuscript are mostly satisfactory. The only point that I would like to mention is the following.

365 Although CBFβ has been reported to stabilize RUNX1⁶⁰, these data suggest
366 that, conversely, RUNX1 contributes to CBFβ stability.

Do the authors suggest that stabilization of RUNX1 by CBFβ is wrong and stabilization of CBFβ by RUNX1 is actually taking place ?

If so, the authors must show definitive evidence for such mechanism.

This experiments were done by using HEK293T cells transfected with various expression plasmids.

HEK293T cells may be expressing endogenous CBFβ without transfecting the expression plasmid.

Stabilization of RUNX1 by CBFβ is demonstrated in CBFβ knockout mice and RUNX1 protein is not detectable in embryonic extracts while RUNX1 transcripts remain unchanged¹. As suggested by the reviewer, we conducted a Western Blot analysis on HEK293T cells, revealing the expression of endogenous CBFβ in the HEK293T cells (Now included in the manuscript as Supplementary Fig.s10b) (also previously reported^{2,3,4}) that would account for the stability of transfected Flag-RUNX1 in HEK293T cells without co-transfection of the CBFβ expression plasmid.

New Supplementary Fig.s10b

Fig. The Western blot analysis revealed the endogenous expression of CBFβ in HEK293T cells.

We do not have definitive evidence for stabilization of RUNX1 by CBFβ and have modified the text accordingly. Modified text:

In results:

HEK293T cells express endogenous CBF β (Supplementary Fig.s10b) and transfection of expression plasmid for RUNX1 with and without CBF β expression plasmid did not markedly affect Flag-RUNX1 protein detection, indicating that endogenous CBF β production was sufficient to stabilize Flag-RUNX1. Co-transfection of expression plasmids for RUNX1 and FBW7 markedly decreased RUNX1 protein level with and without co-transfection with the CBF β expression plasmid (Fig.6c and supplementary Fig.s10a). This suggests that even with increased potential for RUNX1 to complex with CBF β , increased FBW7 E3 ubiquitin ligase expression decreased RUNX1 stability and increased RUNX1 degradation and ubiquitylation indicated by increased intensity of K48-Ub. Co-transfection of expression plasmids for RUNX1, FBW7, CBF β and EpoR increased FBW7 protein and decreased RUNX1 stability, while addition of EPO treatment activated the EPO-EpoR axis and increased RUNX1 stability without affecting FBW7 expression (Fig.6c).

In discussion:

Co-transfection assays in HEK293T cells that express endogenous CBF β demonstrated that increased expression and stability of RUNX1 is affected by expression of FBW7 and activation of the EPO-EpoR axis (Fig.6c).

1. Huang, G., et al., *Dimerization with PEBP2beta protects RUNX1/AML1 from ubiquitin-proteasome-mediated degradation*. EMBO J, 2001. **20**(4): p. 723-33.
2. Wee, H.J., et al., *PEBP2-beta/CBF-beta-dependent phosphorylation of RUNX1 and p300 by HIPK2: implications for leukemogenesis*. Blood, 2008. **112**(9): p. 3777-87.
3. Jager, S., et al., *Vif hijacks CBF-beta to degrade APOBEC3G and promote HIV-1 infection*. Nature, 2011. **481**(7381): p. 371-5.
4. Kim, D.Y., et al., *CBFbeta stabilizes HIV Vif to counteract APOBEC3 at the expense of RUNX1 target gene expression*. Mol Cell, 2013. **49**(4): p. 632-44.

Reviewer #3 (Remarks to the Author):

This resubmission of a revised manuscript contains much added experiments and substantial revision addressing many of the comments. However, some concerns continue to persist as follows:

The insulin tolerance test shown in Fig 1g does not show increased insulin sensitivity as stated by the authors. There is hardly any glucose lowering effect of insulin during ITT and the reviewer requested showing percentage lowering of glucose (if not in conjunct with absolute glucose levels) which was not done. Also, typically with improved insulin sensitivity, there is an increase in adiponectin levels. The lower adiponectin levels in Tg6 mice is also out of keeping with the improved insulin sensitivity that the authors suggest.

We thank the reviewers for the opportunity to improve our manuscript. The reviewer raised crucial questions, prompting us to meticulously rework our experiments. Subsequently, we expanded the sample size for both the WT and Tg6 groups. Upon retesting the GTT and ITT, our findings reaffirmed that Tg6 mice exhibit an enhanced

insulin sensitivity effect (Fig.1e-h). We re-tested the adiponectin levels in serum of WT and Tg6 mice due to an error in the previous version. The new data has been included in Fig.s2b. These results are consistent with the decrease in blood glucose level and improvement in GTT in Tg6 mice with elevated EPO at 3 months as previously, and the decrease in insulin level in fed Tg6 mice and increased sensitivity to insulin were noted ⁵.

The Tg6 mice used as a basis for all of the metabolic implications of Epo on adipocytes needs to be more rigorously vetted. It is stated in the rebuttal that the original intent of the generation of these mice was to investigate the role of Epo action in the brain. However, the authors did not show reporter expression of the Tg6 mice in the brain, but only showed the predominant expression in the adipose tissue. Other tissues that would be important to include, such as the brain, in particular, which can affect energy homeostasis, as well as the cells in the bone marrow or the spleen, which may affect the interpretation of their metabolic findings in the Tg6 mice would be good to include. Furthermore, to be using mice that have significantly shortened lifespan with serious health issues later in life, to use a narrow window that suitably show phenotypes to focus for this report on metabolism is in some ways highly problematic. Also, this information was not included in the manuscript, only in the rebuttal.

We're grateful for the valuable question raised by the reviewer. This paper investigates the Tg6 mice, a strain generated by Dr. Max Gassman's lab. Previous findings have suggested that erythropoietin (Epo) plays a protective role in cerebral ischemia. Consequently, Dr. Max Gassman's lab engineered transgenic EPO mice, creating the Tg6 and Tg21 lines using the PDGF β -chain promoter. Notably, the Tg6 mice exhibited higher EPO expression compared to the Tg21 counterparts. In Tg6 mice the PDGFB promoter demonstrated significant expression not only in the brain but also in other tissues. This prompts consideration that the altered metabolism in mice might be influenced by overexpressed EPO in multiple tissues rather than solely in the brain.

We involved the assessment of EpoR expression in various tissues of WT mice, including the brain cerebral cortex, hypothalamus, spleen, and WAT. Interestingly, the spleen showed higher EpoR expression compared to both WAT and the brain. Moreover, our data indicates relatively higher EpoR expression in WAT compared to the brain. While we acknowledge the reviewer's viewpoint, we confirmed substantial EpoR expression in WAT, suggesting a potential interaction between EPO and EPOR to modulate downstream signals. (Now included as new Supplementary Figure Fig.s4k-m).

Fig. Immunofluorescence staining of EpoR performed in the Cerebral cortex, Hypothalamus, Spleen and WAT in mice. (a) Immunofluorescence stained EpoR and Tdtomato, and Hoechst from the cerebral cortex, hypothalamus, and spleen. Scale bar: upper, 50 μ m, lower, 10 μ m. (b) Immunoblot analysis was conducted to assess the expression levels of EpoR across various organs, with β -Actin serving as the loading control. (c) Quantification of figure b. n = 3, One-way ANOVA with Bonferroni's multiple comparison test.

Our primary focus remains on elucidating how EPO-EPOR signaling influences lipid metabolism in adipose tissues. However, we agree on the importance of investigating other organs in Tg6 mice, particularly the brain, to understand their roles in regulating energy homeostasis in mice. This avenue will be explored in future studies.

We expand on description of Tg6 mice in the text:

Compared with wild-type (WT) littermate controls, Tg6-mice exhibit lower body weight and increased hematocrit beginning two weeks after birth and increased adipose tissue mitochondrial oxygen consumption rate (OCR) (Supplementary Fig.s1a-c). Tg6 hematocrit increases after two weeks of age to 0.52 (Tg6) versus 0.42 (WT) at postnatal day 21, and to 80% at 35 days (Supplementary Fig.s1b). Tg6-mice at 100 days exhibited no inflammation or liver necrosis (Supplementary Fig.s1d-f), indicating that overexpression of transgenic EPO in young Tg6-mice does not induce tissue necrosis. Five month Tg6-mice show no signs of degeneration in liver, kidney, skeletal muscle and sciatic nerve⁶, suggesting that Tg6-mice may be useful to assess response associated with chronic EPO treatment. Tg6-mice show EPO protective activity in light induced retinal degeneration⁷, cardiac-protection in permanent coronary artery ligation⁸, skeletal muscle repair⁹, and metabolic regulation of blood glucose and body mass⁵. Of note, EPO treatment in sick lean mice can decrease inflammation and disease manifestation, and improve locomotor activity as exemplified in a model for sepsis, without significant change in body weight¹⁰. However, after 7 months, Tg6-mice exhibit multiple organ degeneration, liver inflammation, and, in about half the mice, hindlimb paralysis and myofiber degeneration in hindlimb skeletal muscle⁶. Tg6-mice mean survival is reduced (7.4 months)¹¹. Therefore, young Tg6-mice prior to 4 months were used in the current study.

We added new Supplementary Figure Fig.s1d-e.

Fig.s1

Supplementary Figure s1. Body weight, hematocrit and oxygen consumption of adipose tissue, and liver sections from young male Tg6 mice.

d, H&E staining of liver sections from control and Tg6 mice reveal intact tissue without signs of necrosis in Tg6 mice. **e**, Oil red O staining (hematoxylin counterstain) of WT and Tg6 mice liver sections shows no fat accumulation. **f**, The notable increase presence of red cells and blood vessels, indicated by quantification of blood vessels (left), without significant increase in necrotic cells (right) are shown for in the liver sections from WT and Tg6 mice. (n = 8)

H&E staining of Tg6 liver sections show increase presence of red cells and blood vessels, and no signs of necrosis (Supplementary Fig.s1d-f), or of increased inflammatory cytokines (Supplementary Figure.s2e-i), providing evidence that the observed catabolic lipid changes are not a consequence of whole-body necrosis.

Tg6 mice fed high fat diet (HFD) for three weeks also showed lower body weight and reduced fat mass and no significant difference in lean mass compared with WT mice on HFD (Supplementary Fig.s3). Glucose tolerance was also significantly increased compared with WT littermate control mice on HFD mice (Supplementary Fig.s3).

Changes in expression of lipid metabolism genes in Tg6 mice were similarly observed in EPO-treated WT mice, indicating the role of elevated EPO in regulating lipid metabolism gene expression. In addition, we found that young Tg6 mice show increases in brown fat associated gene expression in WAT, consistent with the potential for EPO to promote browning of WAT¹²⁻¹⁵, although it is plausible that the PDGFβ promoter driven transgene in the Tg6 mice also contributes to the observed effects¹³⁻¹⁶.

With regards to tissue distribution of increased transgenic EPO in Tg6 mice we add the following:

Tg6-mice have elevated EPO in plasma and brain. Compared with plasma EPO levels in WT-mice, Tg6 EPO levels determined by RIA were 12x in plasma, 5x in brain (about 17 times endogenous WT brain EPO), 2x to 3x in lung and not elevated in heart, kidney or liver^{6, 14}. Interestingly, Tg21 mice that overexpress transgenic EPO only in brain show no change in hematocrit and the metabolic effect of increased brain EPO on fat mass and body weight in male mice was apparent only after high fat diet feeding and was analogous to response to EPO intracerebroventricular pump in WT-mice on high fat diet¹⁵. Differences in expression of lipid metabolism genes observed between Tg6-mice and WT littermate control mice were comparable to differences in WT-mice between EPO and PBS treatment, suggesting that changes in expression of lipid metabolism genes between Tg6-mice and WT-mice may be largely related to the increased circulating EPO levels.

A balanced literature review including absence of metabolic phenotype in absence of EpoR in adipocytes (PMID: 23885016) would be good to include.

We have now expanded on the literature of EPO effects on metabolism and also include the absence of metabolic phenotype in absence of EpoR in adipocytes (PMID: 23885016).

In other studies, EPO administration reduced blood glucose level and improved glucose tolerance in mouse models that include C57BL/6, ob/ob susceptible to diabetes and obesity, BALB/c, and PTP1B^{-/-} associated with resistance to diabetes, and Tg6-mice at 3 months exhibited lower insulin levels and increased insulin sensitivity⁵. Alternatively, generation of EpoR adipocyte knockdown on a mixed background using aP2-Cre-mice did not show altered glucose or energy homeostasis in diet-induced obese mice at 8 months⁵³, suggesting that genetic background influences metabolic response to EPO. In a mouse model of kidney disease, increasing circulating EPO decreased serum triglyceride, and enhanced lipid catabolism and increased JAK2-STAT5 signaling in adipose tissue⁵⁴. In hematologic patients, EPO treatment was also associated with decreased blood glucose levels⁵⁵.

References

- 1 Huang, G. *et al.* Dimerization with PEBP2beta protects RUNX1/AML1 from ubiquitin-proteasome-mediated degradation. *EMBO J* **20**, 723-733 (2001). <https://doi.org:10.1093/emboj/20.4.723>
- 2 Wee, H. J., Voon, D. C., Bae, S. C. & Ito, Y. PEBP2-beta/CBF-beta-dependent phosphorylation of RUNX1 and p300 by HIPK2: implications for leukemogenesis. *Blood* **112**, 3777-3787 (2008). <https://doi.org:10.1182/blood-2008-01-134122>
- 3 Jager, S. *et al.* Vif hijacks CBF-beta to degrade APOBEC3G and promote HIV-1 infection. *Nature* **481**, 371-375 (2011). <https://doi.org:10.1038/nature10693>
- 4 Kim, D. Y. *et al.* CBFbeta stabilizes HIV Vif to counteract APOBEC3 at the expense of RUNX1 target gene expression. *Mol Cell* **49**, 632-644 (2013). <https://doi.org:10.1016/j.molcel.2012.12.012>
- 5 Katz, O. *et al.* Erythropoietin treatment leads to reduced blood glucose levels and body mass: insights from murine models. *J Endocrinol* **205**, 87-95 (2010). <https://doi.org:10.1677/JOE-09-0425>
- 6 Heinicke, K. *et al.* Excessive erythrocytosis in adult mice overexpressing erythropoietin leads to hepatic, renal, neuronal, and muscular degeneration. *Am J Physiol Regul Integr Comp Physiol* **291**, R947-956 (2006). <https://doi.org:10.1152/ajpregu.00152.2006>
- 7 Grimm, C. *et al.* HIF-1-induced erythropoietin in the hypoxic retina protects against light-induced retinal degeneration. *Nat Med* **8**, 718-724 (2002). <https://doi.org:10.1038/nm723>
- 8 Camici, G. G. *et al.* Constitutively overexpressed erythropoietin reduces infarct size in a mouse model of permanent coronary artery ligation. *Methods Enzymol* **435**, 147-155 (2007). [https://doi.org:10.1016/S0076-6879\(07\)35008-8](https://doi.org:10.1016/S0076-6879(07)35008-8)
- 9 Jia, Y., Suzuki, N., Yamamoto, M., Gassmann, M. & Noguchi, C. T. Endogenous erythropoietin signaling facilitates skeletal muscle repair and recovery following pharmacologically induced damage. *FASEB J* **26**, 2847-2858 (2012). <https://doi.org:10.1096/fj.11-196618>
- 10 Vachharajani, V., Vital, S. & Russell, J. Modulation of circulating cell-endothelial cell interaction by erythropoietin in lean and obese mice with cecal ligation and puncture. *Pathophysiology* **17**, 9-18 (2010). <https://doi.org:10.1016/j.pathophys.2009.04.002>
- 11 Wagner, K. F. *et al.* Chronic inborn erythrocytosis leads to cardiac dysfunction and premature death in mice overexpressing erythropoietin. *Blood* **97**, 536-542 (2001). <https://doi.org:10.1182/blood.v97.2.536>
- 12 Wang, L. *et al.* PPARalpha and Sirt1 mediate erythropoietin action in increasing metabolic activity and browning of white adipocytes to protect against obesity and metabolic disorders. *Diabetes* **62**, 4122-4131 (2013). <https://doi.org:10.2337/db13-0518>
- 13 Ruschitzka, F. T. *et al.* Nitric oxide prevents cardiovascular disease and determines survival in polyglobulic mice overexpressing erythropoietin. *Proc Natl Acad Sci U S A* **97**, 11609-11613 (2000). <https://doi.org:10.1073/pnas.97.21.11609>
- 14 Laouafa, S. *et al.* Hypercapnic ventilatory response is decreased in a mouse model of excessive erythrocytosis. *Am J Physiol Regul Integr Comp Physiol* **311**, R940-R947 (2016). <https://doi.org:10.1152/ajpregu.00226.2016>

- 15 Dey, S. *et al.* Sex-specific brain erythropoietin regulation of mouse metabolism and hypothalamic inflammation. *JCI Insight* **5** (2020).
<https://doi.org:10.1172/jci.insight.134061>
- 16 Luk, C. T. *et al.* In vivo knockdown of adipocyte erythropoietin receptor does not alter glucose or energy homeostasis. *Endocrinology* **154**, 3652-3659 (2013).
<https://doi.org:10.1210/en.2013-1113>
- 17 Oster, H. S., Gvili Perelman, M., Kolomansky, A., Neumann, D. & Mittelman, M. Erythropoietin Treatment Is Associated with Decreased Blood Glucose Levels in Hematologic Patients. *Acta Haematol* **144**, 252-258 (2021).
<https://doi.org:10.1159/000507974>
- 18 Li, J. *et al.* Kidney-secreted erythropoietin lowers lipidemia via activating JAK2-STAT5 signaling in adipose tissue. *EBioMedicine* **50**, 317-328 (2019).
<https://doi.org:10.1016/j.ebiom.2019.11.007>
- 19 Tsuma, Y. *et al.* Erythropoietin and long-acting erythropoiesis stimulating agent ameliorate non-alcoholic fatty liver disease by increasing lipolysis and decreasing lipogenesis via EPOR/STAT pathway. *Biochem Biophys Res Commun* **509**, 306-313 (2019).
<https://doi.org:10.1016/j.bbrc.2018.12.131>

REVIEWER COMMENTS

Reviewer #2 (Remarks to the Author):

Comments for NCOMMS-23-25613B:

Weiqin Yin et al describes the role of Erythropoietin (EPO) -Epo receptor (EpoR) system in energy metabolism. They describe that high transgenic EPO in mice increased glucose tolerance and insulin sensitivity and reduced lipogenic-associated genes in WAT (white adipose tissue) expression with increased transcription factor RUNX1 which directly inhibited lipogenic gene expression. EPO treatment decreased the FBXW7 expression and increased RUNX1 stability. They concluded that EPO-EpoR-RUNX1 axis is the mechanism for the EPO regulation of energy metabolism in mice.

This is an interesting study.

I will comment on the RUNX1 stability in relation to CBFβ. I also ask question on the toxicity of DMSO on mice.

In my earlier comments, I suspected that HEK293T cell line would be expressing CBFβ. The authors confirmed that this is indeed the case.

In the data shown in Figure 6c and Supplementary Figure 10a, the authors compared the stability of RUNX1 in the presence or the absence of EPO, EpoR, FBW7, CBFβ (Fig6c).

The authors showed the importance of each of these components by mixing various combination to see the stability of RUNX1 (including CBFβ in Fig 6c, but not including supplementary Figure 10a, due to the presence of CBFβ endogenously). They describe that CBFβ does not show any significant effect on the stability of RUNX1.

I think the authors' conclusion is premature. The experiments shown in Fig 6c and supplementary Figure 10a are, in a way, somewhat similar to the mixing these components in the test tube in vitro.

The authors describe that

283 However, FBXW7 E3 ubiquitin ligase mRNA expression was significantly reduced (Fig.6a)

and

284 FBW7 protein level encoded by the FBXW7 gene was decreased (Fig.6b) in scWAT of Tg6-mice compared with WT-mice.

In Figure 6c, two lanes in the extreme right show the level of detection of FBW7. In the presence of EPO-EpoR (1st lane from the right) and in the absence of them (2nd lane) show almost the same level of FBW7 protein. Nevertheless, RUNX1 is quite significantly reduced (destabilized) in the 2nd lane from the right. If you look at these data alone, you might like to conclude that FBW7 is not involved in the stability of RUNX1.

The authors' argument that they did not find the evidence for the role of CFBF to stabilize the RUNX1 protein may be similarly premature conclusion. When you transfect RUNX1 expression plasmid to the cells, RUNX1 protein would be synthesized and it must meet with endogenous CFBF protein to form RUNX1/CFBF complex. This complex has been shown to protect the degradation of RUNX1 protein as described by the papers that the authors cited. In the authors' experiment, RUNX1 protein, that is synthesized from the incoming plasmid, would also meet with FBW7 immediately, and the complex would proceed with the RUNX1 degradation pathway, although some fractions of RUNX1 protein may be dimerized with endogenous CFBF protein without interacting with FBW7. Therefore, the way how the experiment was performed suggests that RUNX1 is likely to be interacting with and degraded by FBW7 more easily than protected by CFBF by forming heterodimer.

About the toxicity of DMSO to mice:

The authors dissolved RUNX inhibitor, Ro5-3335, by using DMSO and tested the inhibitory activity of Ro5-3335 on RUNX1. It has been known that DMSO is quite toxic to mice. I wonder what was the effect of DMSO alone to mice compared with the mice without treatment with DMSO ?

Reviewer #3 (Remarks to the Author):

This study by Yin et al uses Tg6-mice mice that overexpress Epo to investigate the metabolic effects of excess Epo on metabolism. While these mice show favourable metabolic effects, these same mice also have severely shortened lifespan with premature death at 7.5 months of age. Thus, to investigate these very mice that are known die only a few months after the time of reported study is highly problematic. Hematocrit of 80% is severely unphysiologic, and to study, in depth, these mice will not be helpful given this important confounding factor that has a major effect on the health and lifespan shortly thereafter. The authors posit that all the investigation is done at 4 mo of age, well before they succumb to sickness and death. However, this is a completely unphysiologic model in which these mice succumb to death soon thereafter. Thus, one interpretation of the favourable metabolic effects could be a catabolic effect that ultimately leads to death of the mice. These mice present ultimately with multiple organ degeneration, liver inflammation, and other neuromuscular defects. To find mechanism that are at play during this period in the life of these mice are not at all meaningful.

They show that that favourable lipolytic effects of Epo on adipose tissue may be through RUNX pathway, potentially through a RUNX binding motif. They show mechanistic effects of RUNX in promoting lipolysis, and also modestly rescuing the improved glucose tolerance with RUNX inhibitor. The group proceeds to show multiple series of experiments using in vitro model to detail the mechanistic actions of RUNX in the EPO signaling pathway. Ultimately, the significance of this novel signaling pathway is still tied to the Tg6 mice that ultimately exhibit concerning premature death. What may be helpful is if RUNX that presumably signal downstream of EPO can have a metabolically beneficial effect without the severely high hematocrit that ultimately leads to premature death of the animal.

Reviewer #4 (Remarks to the Author):

In the present paper, Yin and colleagues showed that EPO receptor (EpoR) expression in white adipose tissue (WAT) mediates EPO metabolic activity. Mice lacking EpoR in adipose tissue have increased fat mass and susceptibility to diet induced obesity, while transgenic EPO expression or exogenous administration of EPO increased glucose tolerance and insulin sensitivity. Mechanistically, these effects are mediated through the EPO-EpoR-RUNX1 axis. The paper is interesting and well done. However, there are issues that are worth considering:

- The use of Tg6-mice mice to study the metabolic effects of EPO is not ideal. The experiments with recombinant human EPO and conditional KO EPOR mice concur supporting the authors' working model. However, the use of inducible transgenic systems or blocking EPO/EPOR Ab would further strengthen the authors' conclusions.
- It is somehow surprising that the authors do not mention the major increase normally observed in the spleen size in EPO-Tg6 mice due to increased erythropoiesis.
- The paper do not address whether the EPO metabolic effects are mediated by homodimer (erythropoietic) or heterodimer (not-erythropoietic) EPOR. This is important, as selective heterodimer EPOR agonists exist and they may have important clinical applications, if EPO metabolic effects are mediated by this receptor.

REVIEWER COMMENTS

Reviewer #2 (Remarks to the Author):

Comments for NCOMMS-23-25613B:

Weiqin Yin et al describes the role of Erythropoietin (EPO) -Epo receptor (EpoR) system in energy metabolism. They describe that high transgenic EPO in mice increased glucose tolerance and insulin sensitivity and reduced lipogenic-associated genes in WAT (white adipose tissue) expression with increased transcription factor RUNX1 which directly inhibited lipogenic gene expression. EPO treatment decreased the FBXW7 expression and increased RUNX1 stability. They concluded that EPO-EpoR-RUNX1 axis is the mechanism for the EPO regulation of energy metabolism in mice.

This is an interesting study.

I will comment on the RUNX1 stability in relation to CBFβ. I also ask question on the toxicity of DMSO on mice.

In my earlier comments, I suspected that HEK293T cell line would be expressing CBFβ. The authors confirmed that this is indeed the case.

In the data shown in Figure 6c and Supplementary Figure 10a, the authors compared the stability of RUNX1 in the presence or the absence of EPO, EpoR, FBW7, CBFβ (Fig6c).

The authors showed the importance of each of these components by mixing various combination to see the stability of RUNX1 (including CBFβ in Fig 6c, but not including supplementary Figure 10a, due to the presence of CBFβ endogenously). They describe that CBFβ does not show any significant effect on the stability of RUNX1.

We misspoke that CBFβ does not affect stability of RUNX1. In fact, evidence from the literature stating that “RUNX1 protein is not detectable in embryonic extracts from CBFβ knockout mice while RUNX1 transcripts remain unchanged¹⁷” indicates that CBFβ is required for stability of RUNX1, which we have now clarified in the text. These reported results suggest a potential protective role of CBFβ in adipocytes against RUNX1 degradation. In our in vitro experiments, we utilized the transfection plasmid method to elucidate how RUNX1 may be degraded in fat cells following EPO treatment in mice. Upon retesting samples from the HEK293T cell transfection experiments, we observed a slight decrease of FBW7-myc in the samples with EPO treatment and EpoR co-transfection.

I think the authors' conclusion is premature. The experiments shown in Fig 6c and supplementary Figure 10a are, in a way, somewhat similar to the mixing these components in the test tube in vitro.

The authors describe that

283 However, FBXW7 E3 ubiquitin ligase mRNA expression was significantly reduced (Fig.6a) and

284 FBW7 protein level encoded by the FBXW7 gene was decreased (Fig.6b) in scWAT of Tg6-mice compared with WT-mice.

In Figure 6c, two lanes in the extreme right show the level of detection of FBW7. In the presence of EPO-EpoR (1st lane from the right) and in the absence of them (2nd lane) show almost the same level of FBW7 protein. Nevertheless, RUNX1 is quite significantly reduced (destabilized) in the 2nd lane from the right. If you look at these data alone, you might like to

conclude that FBW7 is not involved in the stability of RUNX1. The authors' argument that they did not find the evidence for the role of CFBF to stabilize the RUNX1 protein may be similarly premature conclusion.

We thank the reviewer for calling attention to our error in describing Figure 6C.

When you transfect RUNX1 expression plasmid to the cells, RUNX1 protein would be synthesized and it must meet with endogenous CFBF protein to form RUNX1/CFBF complex. This complex has been shown to protect the degradation of RUNX1 protein as described by the papers that the authors cited. In the authors' experiment, RUNX1 protein, that is synthesized from the incoming plasmid, would also meet with FBW7 immediately, and the complex would proceed with the RUNX1 degradation pathway, although some fractions of RUNX1 protein may be dimerized with endogenous CFBF protein without interacting with FBW7. Therefore, the way how the experiment was performed suggests that RUNX1 is likely to be interacting with and degraded by FBW7 more easily than protected by CFBF by forming heterodimer.

In our error, we missed the opportunity to correctly describe the last two right lanes. The text has been modified to include the following:

HEK293T cells express endogenous CBF β (Supplementary Fig.s14b) that provides for RUNX1 stability and detection of Flag-RUNX1 with transfection of RUNX1 expression plasmid (Fig.6c). Transfection of Flag-RUNX1 facilitates detection of endogenous CBF β and co-transfection with CBF β expression plasmid further increases Flag-RUNX1 detection (Fig.6c). Co-transfection of RUNX1 expression plasmid with FBW7 expression plasmid activated the degradation pathway for RUNX1 and Flag-RUNX1 detection was decreased and ubiquitylation increased indicated by increased detection of K48-Ub even with co-transfection of CBF β expression plasmid (Fig.6c and supplementary Fig.s14a). This suggests that RUNX1 was likely to be interacting with CBF β and degraded by FBW7 more easily than protection by heterodimer formation with CBF β . Activity of the degradation pathway for RUNX1 by FBW7 expression and increased ubiquitylation and intensity of K48-Ub were also evident in the presence of EPO signaling (EPO treatment with EpoR expression) with only endogenous CBF β expression. However, the combination of increased CBF β by co-transfection with CBF β expression plasmid and EPO signaling (EPO treatment with EpoR expression) increased Flag-RUNX1 stability and decreased ubiquitylation and K48-Ub even with increased expression of FBW7 (Fig.6c), suggesting that EPO signaling promotes RUNX1 interaction and protective heterodimerization with CBF β and a decrease in RUNX1 degradation by FBW7.

About the toxicity of DMSO to mice:

The authors dissolved RUNX inhibitor, Ro5-3335, by using DMSO and tested the inhibitory activity of Ro5-3335 on RUNX1. It has been known that DMSO is quite toxic to mice. I wonder what was the effect of DMSO alone to mice compared with the mice without treatment with DMSO ?

We have provided detailed methods for the subcutaneous injection of Ro5-3335 into mice. Ro5-3335 was initially diluted in DMSO to a final concentration of 10 mM, then further diluted in saline to achieve a final concentration of 75 μ M, resulting in 0.75 μ l of DMSO per 100 μ l of 75 μ M Ro5-3335 solution. These specific details have been incorporated into the Methods section. We also tested the effect of DMSO on mice and found that there

were no differences in the levels of lipid metabolism gene expression. This is now included as Supplementary Figure 13.

Fig. Comparison between saline and DMSO vehicle control. a-b, WT-mice were treated with saline (100 μ l) or DMSO (0.75 μ l DMSO was diluted in 100 μ l saline) for three weeks, and glucose tolerance (GTT) (a) and area under the curve (AUC of GTT) (b) were determined. c-h, Expression of lipid metabolism genes, *Ppar γ* , *Lpl*, *Acc1*, *Acc2*, *Fas*, *Lipin1*, *Srebf1*, and *Scd1*, were quantified by qRT-PCR for ScWAT (c), eWAT (d), BAT (e), liver (f), and skeletal muscle (g) from Saline (black) or DMSO treated (green) in WT mice. No significant differences were observed between saline and DMSO treatment in WT mice. a-b, n=5, c-g, n=4, p-values are indicated (p>0.05). t-test or Two-Way ANOVA with Bonferroni's multiple comparisons test

Reviewer #3 (Remarks to the Author):

This study by Yin et al uses Tg6-mice mice that overexpress Epo to investigate the metabolic effects of excess Epo on metabolism. While these mice show favourable metabolic effects, these same mice also have severely shortened lifespan with premature death at 7.5 months of age. Thus, to investigate these very mice that are known die only a few months after the time of reported study is highly problematic. Hematocrit of 80% is severely unphysiologic, and to study, in depth, these mice will not be helpful given this important confounding factor that has a major effect on the health and lifespan shortly thereafter. The authors posit that all the investigation is done at 4 mo of age, well before they succumb to sickness and death. However, this is a completely unphysiologic model in which these mice succumb to death soon thereafter. Thus, one interpretation of the favourable metabolic effects could be a catabolic effect that ultimately leads to death of the mice. These mice present ultimately with multiple organ degeneration, liver inflammation, and other neuromuscular defects. To find mechanism that are at play during this period in the life of these mice are not at all meaningful.

As indicated by the reviewer, in Tg6 mice, chronic high transgenic EPO leads to high hematocrit and high blood viscosity. The excessive erythrocytosis results in multiple organ degeneration by 7 months that was not evident at 5 months, a motor disorder with unilateral spasms of the hindlimb that is evident at 7 to 8 months, and shortened life expectancy¹. To confirm that the changes in lipid metabolism in Tg6 mice with high transgenic EPO was related specifically to EPO stimulation of EPOR, we treated WT mice with EPO (Figure 4) and have now added treatment with the non-erythropoietic EPOR agonist ARA290 that is proposed to activate a non-erythropoietic EPOR and examined the metabolic response and the effects on lipid metabolism and lipid metabolic gene expression in adipose tissue. We have added the data in Supplementary Figures 8 and 9 and the following to the manuscript:

The EpoR agonist ARA290 is proposed to promote non-erythroid EpoR activity without increasing EPO stimulated erythropoiesis. As observed with EPO treatment, treatment with EPOR agonist ARA290 in WT mice decreased fat mass and body weight and improved glucose tolerance, although, unlike EPO treatment, ARA290 did not increase hematocrit (Supplementary Fig.s8a-g). As observed with EPO treatment, ARA290 treatment in WT mice changed lipid metabolism gene expression, significantly increased expression of lipolysis genes, *Pparγ*, and *Lpl*, and significantly decreased expression of lipogenic genes, *Acc1*, *Acc2*, *Fas*, *Lipin1*, *Srebf1*, and *Scd1* in scWAT, eWAT, BAT, skeletal muscle and liver compared with saline treatment (Supplementary Fig.s9a-e). Conversely, treatment with EpoR antagonist EMP9 decreased lipolysis gene expression and increased lipogenic gene expression compared with saline treatment in scWAT, eWAT, BAT, skeletal muscle and liver (Supplementary Fig.s9a-e). Treatment with EMP9 and the combination of ARA290 and EMP9 in WT mice did not affect hematocrit or show improvement in body weight, fat mass or glucose tolerance (Supplementary Fig.s8a-g), and treatment with the combination of ARA290 and EMP9 did not affect lipid metabolism gene expression compared with Saline treatment (Supplementary Fig.s9a-e). ARA290 treatment in WT mice on high fat diet showed a greater improvement in metabolism with decreased fat mass/body weight and improved glucose tolerance, and protection against diet-induced obesity, also without change in hematocrit (Supplementary Fig.s8h-l), and accentuated the differences in lipid metabolism gene expression (Supplementary Fig.s9f-j). These data provide additional evidence that EPO regulation of metabolism and glucose tolerance is not dependent on EPO stimulated erythropoiesis, result from EpoR response in non-erythroid tissue and can be simulated by activation of the EpoR heterodimer.

Supplementary Figure 8. EpoR antagonist ARA290 treatment in WT mice decreased fat mass and increased glucose tolerance, without increasing EPO stimulated erythropoiesis. Non-erythropoietic EpoR agonist-ARA290, but not EpoR antagonist-EMP9 mimics the EPO effect on fat accumulation in the WT mice. **a**, Experiment set up chart. **b-g**, WT mice were fed normal diet and treated with Saline (black), ARA290 (red), EMP9 (blue), or ARA290+EMP9 (green) for three weeks and body weight (**b**), fat mass (**c**), lean mass (**d**), and hematocrit (**e**) were monitored, and glucose tolerance (GTT) (**f**)

and area under the curve (AUC of GTT) (**g**) were determined. (**h-m**) WT mice were fed on high fat diet (HFD) and treated with Saline (black), ARA290 (red), EMP9 (blue), or ARA290+EMP9 (green) for three weeks and body weight (**h**), fat mass (**i**), lean mass (**j**), and hematocrit (**k**) were monitored, and glucose tolerance (GTT) (**l**) and area under the curve (AUC of GTT) (**m**) were determined. n = 5, p-values are indicated. Not significant p-values are not shown in the figure. One-way ANOVA with Tukey's multiple comparisons test.

Supplementary Figure 9. Non-erythropoietic EpoR agonist ARA290 mimics the EPO effect on lipid metabolism genes expression. a-e, Lipid metabolism genes expression in the ScWAT (a), eWAT (b), BAT (c), Skeletal muscle (d), and Liver (e) was

assayed with qRT-PCR in WT mice feed with normal food and treated with ARA290, EMP9, or ARA290+EMP9, saline treated group was used as control. **f-j**, Lipid metabolism genes expression in the ScWAT (**f**), eWAT (**g**), BAT (**h**), Skeletal muscle (**i**), and Liver (**j**) was assayed with qRT-PCR in WT mice feed with HFD food and treated with ARA290, EMP9, or ARA290+EMP9, saline treated group was used as control. n= 4 mice each group. p-values for are indicated. Two-way ANOVA with Tukey's multiple comparisons test.

They show that that favourable lipolytic effects of Epo on adipose tissue may be through RUNX pathway, potentially through a RUNX binding motif. They show mechanistic effects of RUNX in promoting lipolysis, and also modestly rescuing the improved glucose tolerance with RUNX inhibitor. The group proceeds to show multiple series of experiments using in vitro model to detail the mechanistic actions of RUNX in the EPO signaling pathway. Ultimately, the significance of this novel signaling pathway is still tied to the Tg6 mice that ultimately exhibit concerning premature death. What may be helpful is if RUNX that presumably signal downstream of EPO can have a metabolically beneficial effect without the severely high hematocrit that ultimately leads to premature death of the animal.

We thank the reviewer for providing us the opportunity to improve our studies on EPO control of lipid metabolism through RUNX1. Because of transgenic Tg6 male mice with high EPO have a high hematocrit phenotype, we have now used the EPOR agonist ARA290 that is proposed to bind the non-erythropoietic heterodimer EPO receptor and lacks erythropoietic activity. In WT male mice treated with ARA290, we observed similar results as those obtained for Tg6 male mice and for WT male mice treated with EPO regarding expression of lipid metabolism genes without change in hematocrit. We also observed that treatment with ARA290 inhibited fat accumulation in WT male mice as we had observed in EPO treated WT male mice, suggesting that these metabolic changes including the RUNX1 dependent regulation of lipid metabolism genes are not a consequence of the high hematocrit observed with EPO treatment or in Tg6 mice. These data are now included in Supplementary Figures 8 and 9 as indicated above.

Reviewer #4 (Remarks to the Author):

In the present paper, Yin and colleagues showed that EPO receptor (EpoR) expression in white adipose tissue (WAT) mediates EPO metabolic activity. Mice lacking EpoR in adipose tissue have increased fat mass and susceptibility to diet induced obesity, while transgenic EPO expression or exogenous administration of EPO increased glucose tolerance and insulin sensitivity. Mechanistically, these effects are mediated through the EPO-EpoR-RUNX1 axis. The paper is interesting and well done. However, there are issues that are worth considering:

- The use of Tg6-mice mice to study the metabolic effects of EPO is not ideal. The experiments with recombinant human EPO and conditional KO EPOR mice concur supporting the authors' working model. However, the use of inducible transgenic systems or blocking EPO/EPOR Ab would further strengthen the authors' conclusions.

We thank the reviewer for the positive remarks regarding our studies and for the opportunity to improve our studies on EPO control of lipid metabolism through RUNX1. Because of transgenic Tg6 male mice with high EPO have a high hematocrit phenotype, we have now used the EPOR agonist ARA290 that is proposed to bind the non-erythropoietic heterodimer EPO receptor and lacks erythropoietic activity. In WT male mice treated with ARA290, we observed similar results as those obtained for Tg6 male mice and for WT male mice treated with EPO regarding expression of lipid metabolism genes without change in hematocrit. We also observed that treatment with ARA290 inhibited fat accumulation in WT male mice as we had observed in EPO treated WT male mice, suggesting that these metabolic changes including the RUNX1 dependent regulation of lipid metabolism genes are not a consequence of the high hematocrit observed with EPO treatment or in Tg6 mice. We have added the data in Supplementary Figures 8 and 9 and the following to the manuscript:

The EpoR agonist ARA290 is proposed to promote non-erythroid EpoR activity without increasing EPO stimulated erythropoiesis. As observed with EPO treatment, treatment with EPOR agonist ARA290 in WT mice decreased fat mass and body weight and improved glucose tolerance, although, unlike EPO treatment, ARA290 did not increase hematocrit (Supplementary Fig.s8a-g). As observed with EPO treatment, ARA290 treatment in WT mice changed lipid metabolism gene expression, significantly increased expression of lipolysis genes, *Ppar γ* , and *Lpl*, and significantly decreased expression of lipogenic genes, *Acc1*, *Acc2*, *Fas*, *Lipin1*, *Srebf1*, and *Scd1* in scWAT, eWAT, BAT, skeletal muscle and liver compared with saline treatment (Supplementary Fig.s9a-e). Conversely, treatment with EpoR antagonist EMP9 decreased lipolysis gene expression and increased lipogenic gene expression compared with saline treatment in scWAT, eWAT, BAT, skeletal muscle and liver (Supplementary Fig.s9a-e). Treatment with EMP9 and the combination of ARA290 and EMP9 in WT mice did not affect hematocrit or show improvement in body weight, fat mass or glucose tolerance (Supplementary Fig.s8a-g), and treatment with the combination of ARA290 and EMP9 did not affect lipid metabolism gene expression compared with saline treatment (Supplementary Fig.s9a-e). ARA290 treatment in WT mice on high fat diet showed a greater improvement in metabolism with decreased fat mass/body weight and improved glucose tolerance, and protection against diet-induced obesity, also without change in hematocrit (Supplementary Fig.s8h-l) and accentuated the differences in lipid metabolism gene expression (Supplementary Fig.s9f-j). These data provide additional evidence that EPO regulation of metabolism and glucose tolerance is not dependent on EPO stimulated erythropoiesis, result from EpoR response in non-erythroid tissue and can be simulated by activation of the EpoR heterodimer.

Supplementary Figure 8. EpoR antagonist ARA290 treatment in WT mice decreased fat mass and increased glucose tolerance, without increasing EPO stimulated erythropoiesis. Non-erythropoietic EpoR agonist-ARA290, but not EpoR antagonist-EMP9 mimics the EPO effect on fat accumulation in the WT mice. **a**, Experiment set up chart. **b-g**, WT mice were fed normal diet and treated with Saline (black), ARA290 (red),

EMP9 (blue), or ARA290+EMP9 (green) for three weeks and body weight (**b**), fat mass (**c**), lean mass (**d**), and hematocrit (**e**) were monitored, and glucose tolerance (GTT) (**f**) and area under the curve (AUC of GTT) (**g**) were determined. (**h-m**) WT mice were fed on high fat diet (HFD) and treated with Saline (black), ARA290 (red), EMP9 (blue), or ARA290+EMP9 (green) for three weeks and body weight (**h**), fat mass (**i**), lean mass (**j**), and hematocrit (**k**) were monitored, and glucose tolerance (GTT) (**l**) and area under the curve (AUC of GTT) (**m**) were determined. n = 5, p-values are indicated. Not significant p-values are not shown in the figure. One-way ANOVA with Tukey's multiple comparisons test.

Supplementary Figure 9. Non-erythropoietic EpoR agonist ARA290 mimics the EPO effect on lipid metabolism genes expression. a-e, Lipid metabolism genes expression in the ScWAT (a), eWAT (b), BAT (c), Skeletal muscle (d), and Liver (e) was assayed with qRT-PCR in WT mice feed with normal food and treated with ARA290, EMP9, or ARA290+EMP9, saline treated group was used as control. **f-j**, Lipid

metabolism genes expression in the ScWAT (f), eWAT (g), BAT (h), Skeletal muscle (i), and Liver (j) was assayed with qRT-PCR in WT mice feed with HFD food and treated with ARA290, EMP9, or ARA290+EMP9, saline treated group was used as control. n= 4 mice each group. p-values for are indicated. Two-way ANOVA with Tukey's multiple comparisons test.

- It is somehow surprising that the authors do not mention the major increase normally observed in the spleen size in EPO-Tg6 mice due to increased erythropoiesis.

We thank the reviewer for drawing attention to spleen size in Tg6 mice. As expected, for high transgenic EPO expression, at 2 months of age, spleen mass for Tg6 mice is about four times greater than WT mice. Spleen sizes are now included for male and female Wt and Tg6 mice in Supplementary Figure 2.

Fig. Spleen weight was assessed in 2-month-old WT and Tg6 mice. (male and female: n = 8 each group). p-value are indicated in figure.

- The paper do not address whether the EPO metabolic effects are mediated by homodimer (erythropoietic) or heterodimer (not-erythropoietic) EPOR. This is important, as selective heterodimer EPOR agonists exist and they may have important clinical applications, if EPO metabolic effects are mediated by this receptor.

To address the question about whether activation of the heterodimer (non-erythropoietic) EPOR can contribute to the EPO metabolic effects we observed, we now include data on WT mice treated with the (not-erythropoietic) EPOR agonist ARA290. We obtained results analogous to those observed with EPO treatment, but without the increase in hematocrit due to EPO stimulated erythropoiesis. These new data (Supplementary Figures 8 and 9 indicated above) provide additional evidence that EPO regulation of metabolism and glucose tolerance is not dependent on EPO stimulated erythropoiesis, result from EpoR response in non-erythroid tissue and can be simulated by activation of the EpoR heterodimer.

** See Nature Portfolio's author and referees' website at www.nature.com/authors for

information about policies, services and author benefits.

This email has been sent through the Springer Nature Tracking System NY-610A-NPG&MTS

Confidentiality Statement:

This e-mail is confidential and subject to copyright. Any unauthorised use or disclosure of its contents is prohibited. If you have received this email in error please notify our Manuscript Tracking System Helpdesk team at <http://platformsupport.nature.com> .

Details of the confidentiality and pre-publicity policy may be found here <http://www.nature.com/authors/policies/confidentiality.html>

Privacy Policy | Update Profile

References

- 1 Heinicke, K. *et al.* Excessive erythrocytosis in adult mice overexpressing erythropoietin leads to hepatic, renal, neuronal, and muscular degeneration. *Am J Physiol Regul Integr Comp Physiol* **291**, R947-956, doi:10.1152/ajpregu.00152.2006 (2006).
- 2 Ruschitzka, F. T. *et al.* Nitric oxide prevents cardiovascular disease and determines survival in polyglobulic mice overexpressing erythropoietin. *Proc Natl Acad Sci U S A* **97**, 11609-11613, doi:10.1073/pnas.97.21.11609 (2000).
- 3 Sansbury, B. E. *et al.* Overexpression of endothelial nitric oxide synthase prevents diet-induced obesity and regulates adipocyte phenotype. *Circ Res* **111**, 1176-1189, doi:10.1161/CIRCRESAHA.112.266395 (2012).
- 4 Wang, L. *et al.* PPARalpha and Sirt1 mediate erythropoietin action in increasing metabolic activity and browning of white adipocytes to protect against obesity and metabolic disorders. *Diabetes* **62**, 4122-4131, doi:10.2337/db13-0518 (2013).
- 5 Kodo, K. *et al.* Erythropoietin (EPO) ameliorates obesity and glucose homeostasis by promoting thermogenesis and endocrine function of classical brown adipose tissue (BAT) in diet-induced obese mice. *PLoS One* **12**, e0173661, doi:10.1371/journal.pone.0173661 (2017).
- 6 Lee, J., Walter, M. F., Korach, K. S. & Noguchi, C. T. Erythropoietin reduces fat mass in female mice lacking estrogen receptor alpha. *Mol Metab* **45**, 101142, doi:10.1016/j.molmet.2020.101142 (2021).
- 7 Juul, S. E. Nonerythropoietic roles of erythropoietin in the fetus and neonate. *Clin Perinatol* **27**, 527-541, doi:10.1016/s0095-5108(05)70037-3 (2000).

- 8 Dey, S. *et al.* Sex-specific brain erythropoietin regulation of mouse metabolism and hypothalamic inflammation. *JCI Insight* **5**, doi:10.1172/jci.insight.134061 (2020).
- 9 Zhang, Y., Rogers, H. M., Zhang, X. & Noguchi, C. T. Sex difference in mouse metabolic response to erythropoietin. *FASEB J* **31**, 2661-2673, doi:10.1096/fj.201601223RRR (2017).
- 10 Reinhardt, M. *et al.* Non-hematopoietic effects of endogenous erythropoietin on lean mass and body weight regulation. *Obesity (Silver Spring)* **24**, 1530-1536, doi:10.1002/oby.21537 (2016).
- 11 Suresh, S., Rajvanshi, P. K. & Noguchi, C. T. The Many Facets of Erythropoietin Physiologic and Metabolic Response. *Front Physiol* **10**, 1534, doi:10.3389/fphys.2019.01534 (2019).
- 12 Singh, A. K. *et al.* Daprodustat for the Treatment of Anemia in Patients Not Undergoing Dialysis. *N Engl J Med* **385**, 2313-2324, doi:10.1056/NEJMoa2113380 (2021).
- 13 Singh, A. K. *et al.* Daprodustat for the Treatment of Anemia in Patients Undergoing Dialysis. *N Engl J Med* **385**, 2325-2335, doi:10.1056/NEJMoa2113379 (2021).
- 14 Brines, M. *et al.* ARA 290, a nonerythropoietic peptide engineered from erythropoietin, improves metabolic control and neuropathic symptoms in patients with type 2 diabetes. *Mol Med* **20**, 658-666, doi:10.2119/molmed.2014.00215 (2015).
- 15 Dmytriyeva, O. *et al.* Short erythropoietin-derived peptide enhances memory, improves long-term potentiation, and counteracts amyloid beta-induced pathology. *Neurobiol Aging* **81**, 88-101, doi:10.1016/j.neurobiolaging.2019.05.003 (2019).
- 16 Cho, B. *et al.* Second-generation non-hematopoietic erythropoietin-derived peptide for neuroprotection. *Redox Biol* **49**, 102223, doi:10.1016/j.redox.2021.102223 (2022).
- 17 Huang, G. *et al.* Dimerization with PEBP2beta protects RUNX1/AML1 from ubiquitin-proteasome-mediated degradation. *EMBO J* **20**, 723-733, doi:10.1093/emboj/20.4.723 (2001).

REVIEWER COMMENTS

Reviewer #2 (Remarks to the Author):

My comments were all taken care of with my satisfaction

Reviewer #3 (Remarks to the Author):

In response to my major concern with the erythropoietic effects of Epo overexpression, they have now added treatment with the non-erythropoietic EPOR agonist ARA290 that is proposed to activate a non-erythropoietic EPOR and examined the metabolic response and the effects on lipid metabolism and lipid metabolic gene expression in adipose tissue. Though the response to this agent is rather short, it is an important data that should be moved to the main figures rather than in the supplemental. More importantly, the mechanistic piece need to also hold in response to this model. While the authors have performed numerous experiments measuring metabolic gene expression, they have not measured changes in RUNX1 or C/EBP β upregulation in this model of pharmacologic agonism of EPOR. Without this specific data, the manuscript would not have much significance due to the unphysiologic concerns mentioned in my last review around pathologically high hematocrit.

Reviewer #4 (Remarks to the Author):

I congratulate the authors for the important findings with ARA290 and I have no further comments.

Point-by-point response to Reviewer' Comments
(Nature Communications manuscript NCOMMS-23-25613C)

Reviewer #2 (Remarks to the Author):

My comments were all taken care of with my satisfaction.

Thanks very much for your support in helping us revise the manuscript.

Reviewer #3 (Remarks to the Author):

In response to my major concern with the erythropoietic effects of Epo overexpression, they have now added treatment with the non-erythropoietic EpoR agonist ARA290 that is proposed to activate a non-erythropoietic EpoR and examined the metabolic response and the effects on lipid metabolism and lipid metabolic gene expression in adipose tissue. Though the response to this agent is rather short, it is an important data that should be moved to the main figures rather than in the supplemental.

To address the reviewers concerns about the use of Tg6 transgenic mice with chronic high transgenic EPO and resultant high hematocrit, we included in the last submission metabolic response of WT mice treated with exogenous recombinant human EPO (AMGEN) and with the non-erythropoietic EpoR agonist ARA290 that is proposed to activate a non-erythropoietic EpoR. The results show that both EPO treatment and ARA290 treatment decrease fat mass, improve glucose tolerance, and increase expression of lipolytic genes and decrease expression of lipogenic genes in adipose tissue, skeletal muscle and liver. The increase in hematocrit observed with EPO treatment is not evident with ARA290 treatment. Similar responses were observed in WT mice on high fat diet treated with ARA290. As suggested by the reviewer, the data for metabolic response of WT mice treated with ARA290 including gene expression in scWAT, skeletal muscle and liver are now presented as a new Figure 5 and gene expression analyses for eWAT and BAT are presented in the new supplementary Figure 8. Results for WT mice on high fat diet treated with ARA290 are presented in new supplementary Figure 9.

More importantly, the mechanistic piece need to also hold in response to this model. While the authors have performed numerous experiments measuring metabolic gene expression, they have not measured changes in RUNX1 or CBF β upregulation in this model of pharmacologic agonism of EpoR. Without this specific data, the manuscript would not have much significance due to the unphysiologic concerns mentioned in my last review around pathologically high hematocrit.

To address the potential mechanism of the metabolic response of WT mice treated with the non-erythropoietic EpoR agonist ARA290, we now include analyses of scWAT from mice treated with ARA290, the EpoR antagonist EMP9 and the combination of ARA290 and EMP9 for RUNX1 mRNA expression, RUNX1 protein, CBF β protein, E3 ubiquitin ligase FWB7 protein, and K48-ubiquitin and K63-ubiquitin. These data show the increase in RUNX1 protein and CBF β protein with ARA290 treatment in supplementary Fig.11.

Reviewer #4 (Remarks to the Author):

I congratulate the authors for the important findings with ARA290 and I have no further comments.

We thank the reviewer for their help in revising this study.

REVIEWERS' COMMENTS

Reviewer #3 (Remarks to the Author):

The authors have addressed all my concerns.